# Evaluating Climate Emulation: Fundamental Impulse Testing of Simple Climate Models

Adria K. Schwarber[1], Steven J. Smith[1,2], Corinne A. Hartin[2], Benjamin Aaron Vega-Westhoff[3], Ryan Sriver[3]

[1]Department of Atmospheric and Oceanic Science, University of Maryland, College Park, MD 20742

[2]Joint Global Change Research Institute, 5825 University Research Ct, College Park, MD 20740

[3]Department of Atmospheric Sciences, University of Illinois Urbana-Champaign, Champaign, IL 61820

*Correspondence to*: Adria K. Schwarber (adria.schwarber@gmail.com)

**Abstract.** Simple climate models (SCMs) are numerical representations of the Earth's gas cycles and climate system. SCMs are easy to use and computationally inexpensive, making them an ideal tool in both scientific and decision-making contexts (e.g., complex climate model emulation; parameter estimation experiments; climate metric calculations; and probabilistic analyses). Despite their prolific use, the fundamental responses of SCMs are often not directly characterized. In this study, we use fundamental impulse tests of three chemical species ($CO_2$, $CH_4$, and BC) to understand the fundamental gas cycle and climate system responses of several comprehensive (Hector v2.0, MAGICC 5.3, MAGICC 6.0) and idealized (FAIR v1.0, AR5-IR) SCMs. We find that while idealized SCMs are widely used, they fail to capture the magnitude and timescales of global mean climate responses under emissions perturbations, which can produce biased temperature results. Comprehensive SCMs, which have physically-based non-linear forcing and carbon cycle representations, show improved responses compared to idealized SCMs. Even the comprehensive SCMs, however, fail to capture response timescales to BC emission perturbations seen recently in two general circulation models. Some comprehensive SCMs also generally respond faster than more complex models to a $4xCO_2$ concentration perturbation, although this was not evident for lower perturbation levels. These results suggest where improvements should be made to SCMs. Further, we demonstrate here a set of fundamental tests that we recommend as a standard evaluation suite for any SCM. Fundamental impulse tests allow users to understand differences in model responses and the impact of model selection on results.

## 1 Introduction

Models are one of the primary tools used by interdisciplinary scientists to understand changes in the climate. These models can be classified by their complexity and comprehensiveness, spanning a range from idealized simple climate models (SCMs) to complex coupled Earth System Models (ESMs). While ESMs run on supercomputers and can take several months to simulate 100 years, SCMs can simulate the same period on a personal computer in seconds (van Vuuren et al., 2011a). SCMs have less detailed representations than ESMs, and themselves range in structure from idealized to more comprehensive climate representations (Millar et al., 2017). Comprehensive SCMs are models rooted in physical processes (e.g., energy balance models) and capture the main pathway by which climate forcers

alter the energy budget: emissions to concentrations, top-of-the-atmosphere radiative forcing, and global mean surface air temperature (Geoffroy et al., 2013; Hartin et al., 2015; Meinshausen et al., 2011; Tanaka and Kriegler, 2007). Idealized SCMs use even fewer equations, which do not necessarily correspond to specific physical processes, to parametrically represent the climate system (Millar et al., 2017).

SCMs are widely used in scientific and decision-making contexts largely because of their advantageous features, including their ease of use, transparency, and low computational intensiveness. In particular, SCMs are traditionally used within human-Earth system models that couple the climate system with representations of dynamics within the human system (e.g., energy systems and land-use changes) (Hartin et al., 2015; Ortiz and Markandya, 2009; S.H. Schneider and S.L. Thompson, 2000; Strassmann and Joos, 2018) and are used to assess global forcing or temperature targets (e.g., Representative Concentration Pathways (van Vuuren et al., 2011b), Shared Socioeconomic Pathways (Moss et al., 2010)). Several studies investigated potential sources of human-Earth system model uncertainty by exploring the climate components driving the models (Calel and Stainforth, 2017; Harmsen et al., 2015; van Vuuren et al., 2008, 2011a). Van Vuuren et al. (2011a) concluded that in most cases the results from human-Earth system models and SCMs were similar to the more complex, coupled ESMs. The authors further noted that differences in SCM results can have implications for decision makers informed by such results, illustrating the need for improvements in uncertainty analysis (e.g., carbon cycle feedbacks). Harmsen et al. (2015) extended van Vuuren's analysis to investigate emission reduction scenarios by including non-$CO_2$ radiative forcing. The authors concluded that many models may underestimate forcing differences after applying emission reduction scenarios, due to the omission of important short-lived climate forcers, such as black carbon (BC).

Few studies utilize idealized SCMs in human-Earth system models because of their inability to represent nonlinear forcings, such as air-sea exchanges (Khodayari et al., 2013) or ocean chemistry (Hooss et al., 2001; Tanaka and Kriegler, 2007). With simple extensions of the carbon cycle (e.g., ocean carbonate chemistry), both Hoos et al. (2001) and Tanaka and Kriegler (2007) found improved responses from their respective impulse response models, applicable when coupling to human-Earth system models.

Comprehensive SCMs are also used to simulate the climate or carbon cycle (Friedlingstein et al., 2014; Joos et al., 1999; Knutti et al., 2008), explore responses to anthropogenic perturbations (Geoffroy et al., 2013; Hope, 2006; Meinshausen et al., 2009; Rogelj et al., 2014), or address model spread in the various model intercomparison projects (MIPs) (Knutti and Sedláček, 2012; Monckton et al., 2015; Rogelj et al., 2012). These analyses often include comparisons to more complex models (Meinshausen et al., 2008, 2011). One comprehensive SCM in particular, MAGICC 6.0, is used as a reference in many studies because of its well-documented ability to emulate complex models (e.g., van Vuuren et al., 2011a).

Similarly, individual idealized SCM developers also explore the ability of impulse-response functions to simulate climate or carbon-cycle responses to perturbations (Hooss et al., 2001; Millar et al., 2016; Sausen and Schumann,

2000; Strassmann and Joos, 2018; Thompson and Randerson, 1999), often also comparing to more complex models (Joos and Bruno, 1996). Sand et al. (2016), for example, employed an idealized SCM using sums of exponentials (AR5-IR) to find the Arctic temperature response to regional short-lived climate forcer emissions (e.g., BC) and compared these responses to more complex models.

Climate indicators, such as transient climate response (TCR) (Allen et al., 2018; Millar et al., 2017), can also be informed using SCMs. TCR is the measure of the climate response to a 1% $yr^{-1}$ increase in $CO_2$ concentration until doubling of $CO_2$ relative to pre-industrial level. TCR is useful for understanding the climate response on shorter time scales, as $CO_2$ concentration doubling takes place in 70 years, a time-frame relevant for many planning decisions (Flato et al., 2013; Millar et al., 2015). TCR and the equilibrium climate sensitivity (ECS) can be combined to estimate the realized warming fraction (RWF), the fraction of total warming manifested up to a given time. Millar et al. (2015) investigated TCR and ECS within an impulse-response model to show the implications of these values on future climate projections by specifically looking at the RWF.

Idealized models using sums of exponentials are also commonly used to calculate other climate metrics, such as the global warming potential (GWP) and global temperature potential (GTP) (Aamaas et al., 2013; Berntsen and Fuglestvedt, 2008; Fuglestvedt et al., 2010; Peters et al., 2011; Sarofim and Giordano, 2018). Idealized SCMs, however, often do not account for carbon cycle feedbacks, important for more realistic representations of climate. Both Millar et al. (2017) and Gasser et al. (2017) investigated the effects of adding carbon cycle feedbacks on these metrics produced with idealized SCMs, and found that accounting for feedbacks improved model responses (at least modestly, Gasser et al. 2017).

Despite their importance and wide use, the fundamental responses of SCMs have not been fully characterized (Thompson, 2018). In this paper, we use impulse-response tests to address this gap.

## 2 Methods

**2.1 Fundamental impulse tests.** Impulse-response tests characterize the SCMs' climate and gas-cycle response to a forcing or emission impulse (Good et al., 2011; Joos et al., 2013). Though fundamental impulse tests have been used in the literature (e.g., Joos et al., 2013), we employ these existing techniques to evaluate several SCMs. The U.S. National Academies specifically suggested that SCMs be "assessed on the basis of [the] response to a pulse of emissions," which we do here (2016).

We use three tests to understand the response of the climate system and gas cycles in the models: (a) a concentration impulse of $CO_2$, (b) emissions impulses of BC, $CH_4$, or $CO_2$, (c) a 4×$CO_2$ step increase in $CO_2$ concentration, as described in supplementary section 1 (hereafter, S1). We carry out these experiments by instantaneously increasing emissions or forcing values in 2015 to avoid the model base years of our SCMs (S4).

We note that impulse-response tests can be considered a type of unit test. Unit testing in software refers to a specific method of comparing output from the smallest portion of code, called a unit (i.e., function), to known outputs (Clune

and Rood, 2011). Here, we use this term in a similar way as van Vuuren et al. (2011), where MAGICC 6.0 was used as the reference output to compare several human-Earth system models. We conduct our tests with comparable inputs, which are provided in the Supplementary Materials, and compare model-generated outputs from several SCMs.

The impulse tests result in an impulse response function (IRF) for each model/species combination. IRFs characterize

the dynamics of a linear system (Joos and Bruno, 1996; Ruelle, 2009) and, although climate models exhibit nonlinear responses, even some non-linear systems can be approximated by IRFs for small perturbations (Hooss et al., 2001; Lucarini and Sarno, 2011; Lucarini, 2018). The impulse responses examined here can be considered Green's functions, which form a key component of many simple climate models (Joos et al., 1999; van Vuuren, 2011a; Millar et al., 2015).


**2.2 Background concentrations.** Our impulse response tests are conducted against a time-changing greenhouse gas (GHG) concentration background using emissions from the Representative Concentration Pathway (RCP) 4.5 scenario (Thomson et al., 2011). For each test, therefore, we run a reference scenario in the SCMs, followed by each perturbation case. We report the response, which is obtained by subtracting the reference from the perturbation results

for each model. A changing GHG background concentration is a more realistic scenario overall and also reveals biases not otherwise apparent under constant concentration conditions, for example, in SCMs insensitive to changing background concentrations. Further, for emissions impulses this methodology is more readily implemented as a standard impulse test (see S1), as we recommend below. Conducting tests against a constant concentration background in any but the most idealized SCM requires an inversion calculation to determine the emissions pathway that results

in a constant concentration. This is an unnecessary barrier to conducting routine impulse response tests.

**2.3 Model selection.** Three comprehensive SCMs—Hector v2.0 (Kriegler, 2005; Hartin et al., 2015), MAGICC 5.3 BC-OC (Raper, et al. 1996, Wigley and Raper 2002, Smith and Bond, 2014), and MAGICC 6.0 (Meinshausen et al., 2011)—are used in this study (S2). The models were selected based on their availability, use in the literature, and their

applicability to decision making. We also include two idealized SCMs which employ sums of exponentials to represent the climate or gas-cycle responses, a general approach often used in the literature (Aamaas et al., 2013; Fuglestvedt et al., 2003), referred to as IRFs. A widely used version tested here is the impulse response (IR) model used in the Intergovernmental Panel on Climate Change Fifth Assessment Report (Myhre et al., 2013; See Section 8.7.1.2 - 8.7.1.3; See Section 8.SM.11 for model equations), referred to here as AR5-IR. Additionally, we test version 1.0 of

the Finite Amplitude IR (FAIR) model, an extension of AR5-IR including a representation of carbon cycle feedbacks and non-linear forcing (Millar et al., 2017).

**2.4 Parameter choices.** We are testing the model responses as they would be 'out of the box' and only make modifications if required for the models to run. We note that due to structural differences in the SCMs it is, in general,

not possible to operate the models with identical parameter values (see S2). This reinforces the importance of conducting fundamental impulse response tests to quantify the behavior of the SCMs. However, we have used identical climate sensitivity values, where possible, and discuss in greater detail the specifications used to conduct our tests in each SCM in S1 and S2, including providing input files for each model (S14). Further, a model's ability to emulate an ESM or the multi-model ESM mean is generally explored by the individual SCM development teams, as noted in

the references for the Hector, MAGICC, and FAIR models. While emulation is outside the scope of this paper, we conduct sensitivity tests by relying on parameters derived from ESM emulation experiments using MAGICC 6.0 (see S11).

## 3 Results

In our paper, we evaluated the SCMs by comparing the models to each other and also, in the limited cases where this

is possible, to more complex models (Joos et al., 2013). We compare against the suite of complex model results because it has been shown that the multi-model mean behavior of the complex models replicates well a broad suite of observations (e.g., Figure 9.7, Flato et al. 2013). We highlight differences in model responses to a suite of impulse tests to support an informed model selection (see Table 1).

We begin by testing the fundamental dynamics of the temperature response to a well-mixed greenhouse gas forcing impulse by perturbing $CO_2$ concentrations (Fig. 1), bypassing the carbon cycle (if present). We report both time-series responses (Fig. 1a) and time-integrated responses (Fig. 1b; S9). Integrated responses form the basis of commonly used metrics, such as GWP and GTP (Fuglestvedt et al., 2010).

**3.1 Responses to $CO_2$ Concentration Impulse.** First, we consider the comprehensive SCMs. Both versions of MAGICC show shifted responses in the first few years following the perturbation due to the way this model treats sub-annual integration of forcing (S5 and S6). The shifted responses do not significantly impact integrated results. MAGICC 6.0 initially responds more strongly to the perturbation, with a 6% larger integrated temperature response 20 years after the impulse compared to the comprehensive SCM average (S9). After 30 years, the comprehensive

SCMs are within 2% of each other.

The idealized SCMs show varied responses to a $CO_2$ concentration impulse. Differences in the AR5-IR and FAIR responses are due to a nonlinearity also present in FAIR. According to Equation 8 in Millar et al. (2017) FAIR will have a differential response to changing background $CO_2$ concentrations. By contrast, AR5-IR parameterizes the

climate response to a unit forcing using a sum of exponentials as given by Equation 8.SM.13 in Myhre et al. (2013).

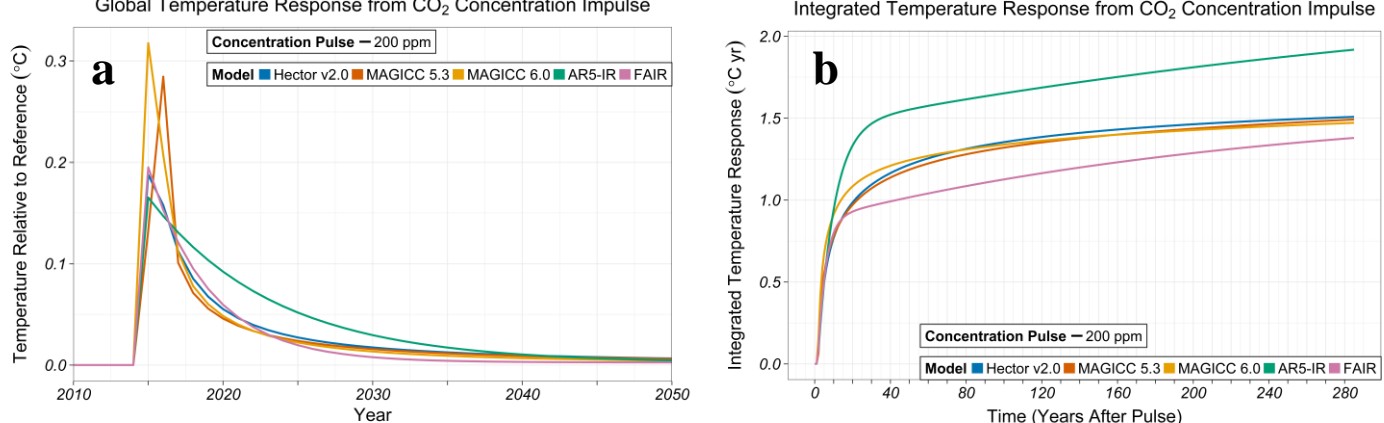

**Figure 1: Global mean temperature response (a) and integrated global mean temperature response (b) from a CO₂ concentration perturbation in SCMs (MAGICC 6.0 – yellow, MAGICC 5.3 BC-OC – red, Hector v2.0 – blue, AR5-IR – green, FAIR –pink). The perturbations are conducted in 2015 against the background of the Representative Concentration Pathway (RCP) 4.5 scenario (see Methods). The time-integrated response, analogous to the Absolute Global Temperature Potential, is reported as 0-285 years after the perturbation (S8).**

AR5-IR has a much stronger response compared to the comprehensive SCMs; the integrated response is 6% larger than the comprehensive SCMs 20 years after the pulse, increasing to 30% by the end of the model runs. This large difference is due to the absence of feedbacks and nonlinearities in the AR-IR model. FAIR contains an approximate representation of these nonlinearities, responding similarly to the comprehensive SCMs in the near-term, but has a 7% weaker integrated response 285 years after the impulse. The approximations used to represent the carbon cycle and non-linear forcing might account for this, but it is unclear from these results.

**3.2 Responses to Emissions Impulses.** We now test the model response to an emissions impulse. Compared to forcing-only experiments, emissions perturbation experiments have additional levels of uncertainty from the conversion of emissions to concentrations, as well as carbon cycle feedbacks. As a diagnostic we examine the forcing response, functionally equivalent to examining the concentration response (S7). The three comprehensive SCMs have small differences (<10%) in the integrated forcing response (Fig. 2b) from CO₂ (dashed) emission impulses for all time horizons. AR5-IR, an idealized SCM, responds 11% stronger than the comprehensive SCMs average 20 years after the pulse, increasing to a 17% difference in the integrated response 285 years after the impulse. FAIR does not calculate concentration or forcing, so cannot be included in these comparisons.

We complete the model response sequence by examining the temperature response from emissions perturbations, which is conceptually the combination of the temperature response from a concentration impulse (Fig. 1) and the forcing response from an emissions impulse (Fig. 2). Similarities in the comprehensive SCM responses in Figs. 1 and 2 are reflected in the <5% difference in the temperature response from a CO₂ emissions perturbation 20 years after the impulse (Fig. 3a). AR5-IR responds 30% stronger and FAIR <10% weaker compared to the comprehensive SCM

average 20 years after the perturbation (Fig. 3a). FAIR introduces a state-dependent carbon cycle representation

(Millar et al., 2017) and is, in general, an improvement over AR5-IR, but shows a systematic difference with the comprehensive SCMs.

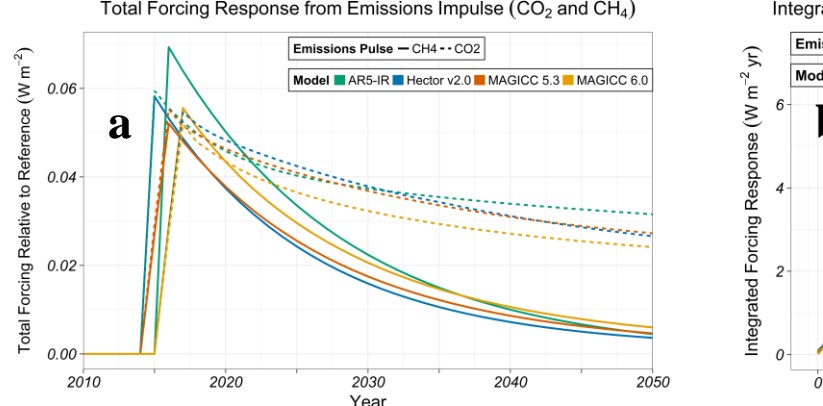 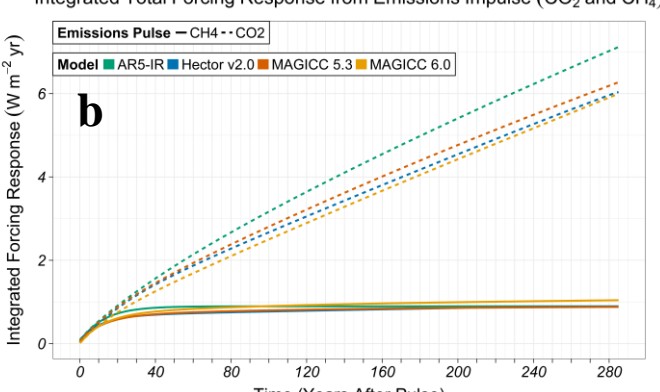

**Figure 2: Total forcing response from CO₂ (dashed) and CH₄ (solid) emissions perturbations in SCMs (MAGICC 6.0 – yellow, MAGICC 5.3 BC-OC – red, Hector v2.0 – blue, AR5-IR – green). FAIR does not report forcing. We report the total forcing response, which has slight differences from the gas-only forcing response. The perturbations are conducted in 2015 against the background of the Representative Concentration Pathway (RCP) 4.5 scenario (see Methods). The time-integrated response, analogous to the Absolute Global Warming Potential, is reported as 0-285 years after the perturbation (S8).**

We indirectly compare the time-integrated airborne fraction in our SCMs to three comprehensive ESMs and seven Earth System Models of Intermediate Complexity (EMICs) using results from the Joos et al. (2013) 100 GtC CO₂ pulse experiment, henceforth referred to as Joos et al. Unlike Joos et al., we conduct this experiment with a changing

background concentration (S12). The airborne fraction is, therefore, higher in our results. Despite the difference in methodology, comparing the MAGICC 6.0 results here and in Joos et al. allows us to use transitive logic to draw broader conclusions about the other comprehensive SCMs. We note that the Joos et al. MAGICC 6.0 ensemble mean airborne fraction is similar to their multi-model mean at each time horizon (Fig. S28). Because Hector and MAGICC 5.3 have a similar response to MAGICC 6.0 in our results, we conclude that the comprehensive SCM carbon cycle

representations generally capture ESM and EMIC responses to the extent this can be evaluated for indirect comparison.

Similarly, we compare the temperature response of the comprehensive SCMs to Joos et al. We find that the comprehensive SCMs capture ESM and EMIC responses in the near-term, with expected differences in response over longer time horizons due to rising background concentrations (S12).


For idealized SCMs, we find that under changing background conditions, FAIR underestimates the airborne fraction compared to the Joos et al. multi-model mean at each time horizon. Without a physical processes-based carbon cycle, AR5-IR is insensitive to pulse size and background concentration (Millar et al., 2017), which results in a similar time-integrated airborne fraction compared to the Joos et al. multi-model mean at each time horizon. The comprehensive

SCMs and to a lesser extent, FAIR, offer an improved response compared to AR5-IR (Millar et al., 2017).

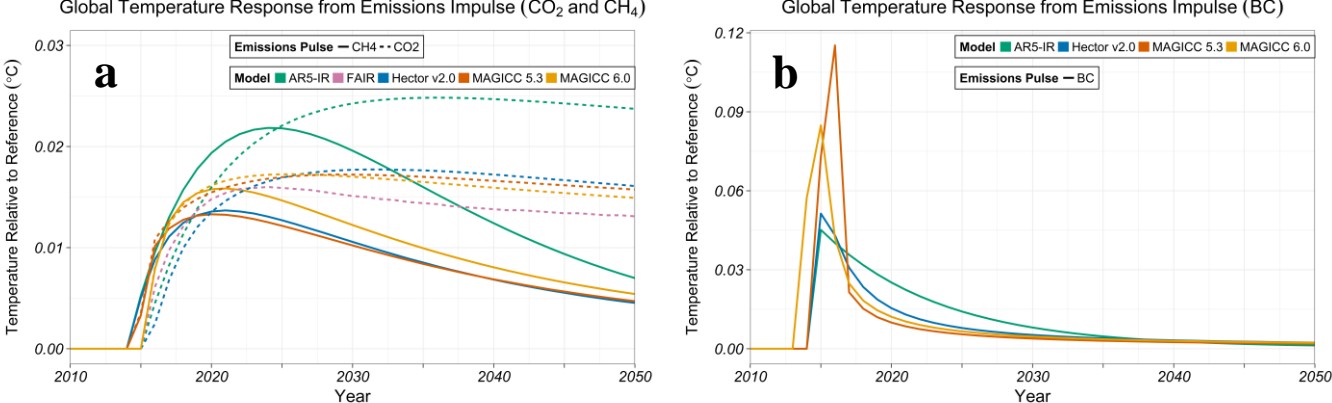

**Figure 3: Global mean temperature response from CO₂ and CH₄ emissions perturbations (a) and BC emissions perturbation (b) in SCMs (MAGICC 6.0 – yellow, MAGICC 5.3 BC-OC – red, Hector v2.0 – blue, AR5-IR – green, FAIR - pink).**

We next consider model responses to methane (CH₄) emissions perturbations, a shorter-lived greenhouse gas with a dynamic atmospheric lifetime (see S1). The integrated forcing responses of Hector and MAGICC 5.3 are similar, as expected (S9.3). The MAGICC 6.0 integrated forcing response difference from the comprehensive SCM average is 9% larger 100 years after the pulse, however (Fig. 2b). As in the CO₂ emissions perturbations, AR5-IR has a much stronger forcing response to a CH₄ emissions perturbation—22% larger 20 years after the pulse—with no meaningful increase 50 years after the pulse (S9).

Finally, we look at the models' temperature responses to aerosols by perturbing black carbon (BC) forcing (Fig. 3b). The BC response increases quickly in both MAGICC models compared to the other SCMs (S9.4). Differences in these responses to a BC perturbation derive from model design. Both versions of MAGICC have differential and faster forcing responses over land, where most BC is located, compared to oceans, termed the geometrical effect (Meinshausen et al., 2011). This results in MAGICC responding faster than Hector v2.0, which does not differentiate forcing over land and ocean. Because AR5-IR represents the aerosol forcing as an exponential decay, the integrated temperature response is 20% stronger 20 years after the pulse compared to the other SCMs.

Due to the geometrical effect, we presume that the faster response in MAGICC is more realistic. However, models vary in the representations of aerosol effects (S2). The greenhouse gas-like representation of aerosols in AR5-IR, for example, results in the unrealistically long response time scale found in this test. We do not explicitly conduct other aerosol perturbations (e.g., sulfate), but we would expect results showing similar responses.

BC has a unique set of atmospheric interactions as an absorbing aerosol, causing warming within the atmosphere, but potentially also surface cooling (Stjern et al., 2017, Yang et al. 2019). The response to a step change in BC emissions in two coupled model experiments has been found to have a flat long-term temperature response (Sand et al., 2016; Yang et al., 2019). In contrast, the comprehensive simple models continue to respond over a much longer time scale (S13). This is an indication that SCM responses to BC, in particular, should be reevaluated.

**3.3 Responses to 4xCO₂ Concentration Step.** Finally, we compare our SCMs with complex models using the abrupt 4xCO₂ concentration experiment from Phase 5 of the Coupled Model Intercomparison Project (CMIP5) (Taylor et al., 2012) (see S1 and S3). We find that Hector, MAGICC 5.3, and FAIR have initially quicker responses to an abrupt
4xCO₂ concentration increase (Fig. 4). This is also reflected in their long term RWF, which is also larger than most of the complex models (see S10). Compared to the other SCMs, AR5-IR has a faster response to an abrupt 4xCO₂ concentration increase and is consistent with the stronger response to a forcing impulse. Differences between the model responses to a finite pulse (Fig. 1) and a large concentration step (Fig. 4) demonstrates the expected bias in AR5-IR under larger perturbations because it lacks the non-linear relationship between concentration and forcing.
This insensitivity of idealized SCMs to changing background concentrations will also bias results if used under realistic future pathways (Millar et al., 2017).

Compared to the other comprehensive SCMs, MAGICC 6.0 initially responds more strongly under a CO₂ concentration impulse (Fig. 1). In the non-linear abrupt 4xCO₂ concentration regime, however, MAGICC 6.0 responds
more slowly, similar to the complex model responses, especially in the first 20 years after the pulse. MAGICC 6.0 appears to respond more reasonably under stronger forcing conditions than the other SCMs.

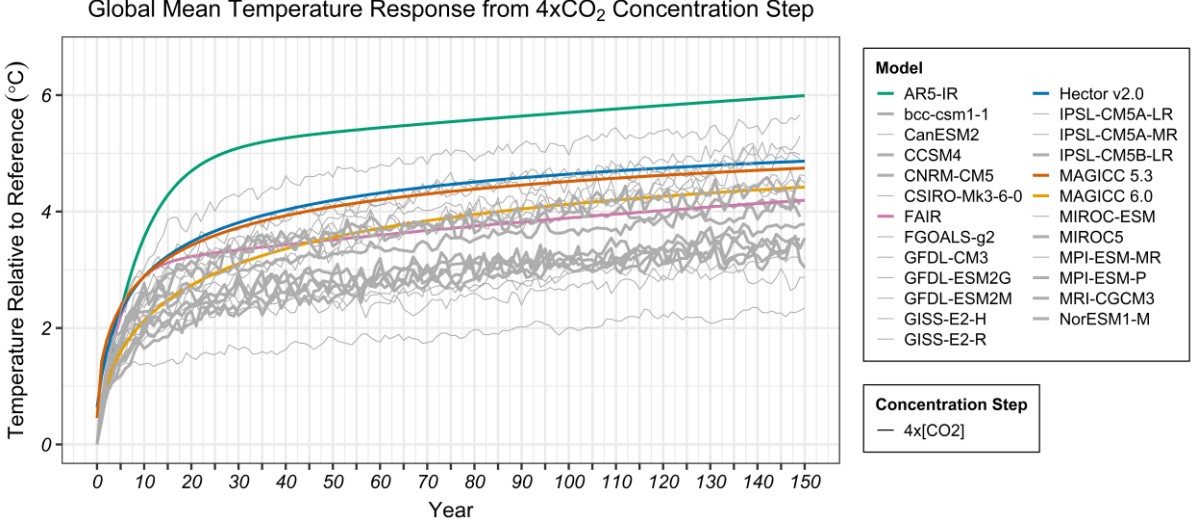

**Figure 4: Global mean temperature response from 4xCO₂ concentration step in CMIP5 models (grey) and SCMs (MAGICC 6.0 – yellow, MAGICC 5.3 BC-OC – red, Hector v2.0 – blue, FAIR – pink, AR5-IR –green). A climate sensitivity value of 3°C was used in the comprehensive SCMs, while in the idealized SCMs the parameter is not adjustable (see S2). The thick lines represent CMIP5 models with an ECS between 2.5 - 3.5 °C.**

**4 Discussion and Conclusion**

The impulse response tests conducted here enable us to uncover differences in model behavior that are not apparent
when running standard, multi-emission scenarios. Indeed, one of the important uses of SCMs is to conduct model experiments where there may be relatively small changes in emissions between two scenarios. Because SCMs do not exhibit internal variability, impulse experiments can be used to quantify such changes. Impulse response tests also

allow us to understand, on a more fundamental level, differences between SCMs that have been found comparing simulations of more conventional scenarios (e.g., van Vuuren et al., 2011a).


By using fundamental impulse tests, we found that idealized SCMs using sums of exponentials often fail to capture the responses of more complex models. SCMs that include some representations of non-linear processes, such as FAIR, show improved responses, though these models still do not perform as well as comprehensive SCMs with physically-based representations. Fundamental tests, such as a $4xCO_2$ concentration step, show that most of the SCMs

used here have a faster warming rate in this strong forcing regime compared to more complex models. However, comprehensive SCM responses are similar to more complex models under smaller, more realistic perturbations (Joos et al., 2013).

It is not possible to compare these fundamental responses with observations, and it is even more difficult to compare

SCMs with the more complex models at decadal time horizons due to internal variability (e.g., Joos et al., 2013, Figure 2a). However, it is common in the climate modeling literature to use the multi-model mean as a base for comparison (e.g., Joos et al., 2013).

For purposes of summarizing our results we compare the individual model responses to the comprehensive SCM

multi-model mean for most of our experiments. We use this both for convenience and because the comprehensive SCMs can generally replicate the long-term results of general circulation models (GCMs; Meinshausen et al., 2011; Joos et al., 2013; Hartin et al., 2015, 2016). This is also, in a general philosophical sense, in line with the finding from GCMs that multi-model means compare better to observations than individual models (Flato et al., 2013), although we note that the Flato et al. finding was not specifically for global temperature. We, therefore, are not

implying that the comprehensive SCM mean is necessarily the most accurate representation of the actual climate system response. It is instead simply a convenient metric for comparison. This metric illustrates both where the comprehensive SCMs are similar or different, and where the more idealized models differ from the comprehensive SCMs. Most of these latter differences are due to simplifications in the idealized models that bias their results, as discussed previously.


We also use the CMIP5 multi-model mean, developed using only those complex models with comparable climate sensitivity values to the SCMs (S10), to compare the SCM responses to a $4xCO_2$ concentration step.

As a summary of our findings, we report the differences in time-integrated temperature response from the relevant

multi-model mean in Table 1 for each of the experiments at selected time horizons. We chose the time horizons to report for each experiment by taking into consideration the atmospheric lifetime of the species and the ability to compare the experiments. For example, to compare the experiments exploring responses to $CO_2$ perturbations, we report the responses at 100 years after the pulse. For $CH_4$ and BC, we report at a time horizon of 20 years after the

pulse reflecting the shorter lifetime of these species. Additional time-integrated temperature responses can be found in S9.

The comprehensive SCMs respond similarly to a $CO_2$ concentration impulse, within 2% of their mean for at 100 years after the pulse (H = 100, Table 1), with a slightly larger difference at 20 years (-4% to 3%, see S9.2). The idealized SCMs, FAIR v1.0 and AR5-IR, have greater differences 100 years after the pulse in opposite directions. The difference in integrated temperature response between the models are only slightly larger for a $CO_2$ emissions pulse.

The comprehensive SCMs show more diverse changes to a $CH_4$ emissions impulse, ranging from -6% to 9% at 20 years after the pulse (H=20, Table 1). AR5-IR overestimates the response by a larger amount likely due to the absence of feedbacks and nonlinearities in the model. It would be useful to evaluate more complex model responses, however, to determine if the simple representation of chemistry in the comprehensive SCMs adequately represents the time evolution of $CH_4$ concentrations in response to a change in emissions.

Under the $4xCO_2$ concentration step experiment, we can compare the SCM responses to more complex models from CMIP5. MAGICC 6.0 appears to respond more reasonably under stronger forcing conditions than the other SCMs 100 years after the pulse, though only marginally better than FAIR. Hector v2.0, MAGICC 5.3, and FAIR have initially quicker responses to an abrupt $4xCO_2$ concentration increase compared to the ESMs (Figure 4). AR5-IR has too strong a response to a $4xCO_2$ concentration increase because it is insensitive to changing background concentrations and, therefore, does not account for the logarithmic dependence of forcing on $CO_2$ concentrations. Because of this dependence forcing from a $4xCO_2$ change is less than twice the forcing from a $2xCO_2$ concentration change.

Finally, we do not have a definitive reference for the time-dependent response to BC forcing perturbations. Instead, we compare the SCMs using the difference from the average of both MAGICC models, which both differentiate aerosol forcing between land and ocean, resulting in a faster overall climate response to aerosols as compared to greenhouse gases (Shindell et al., 2014; Sand et al., 2016; Yang et al. 2019). In the case of BC, we note that all of the SCM responses should be taken critically because none show the fast temporal response to a BC step recently found in more complex models. An experiment using NorESM found a very short temporal response to a global step perturbation in black carbon (BC) with minimal long-term response (Sand et al., 2016) with a similarly short timescale found for BC perturbations in the Arctic and Mid-Latitudes (Yang et al., 2019). A more definitive evaluation of climate system responses to aerosol perturbations in general would be useful. This would require additional complex model simulations of step emission changes for various aerosol species and/or forcing mechanisms.

| Species | Impulse | Time After Pulse | Percent Integrated Temperature Response Differences for each Simple Climate Model (%) | | | | |
|---------|---------|------------------|-----------|-----------|-----------|-----------|--------|
| | | | Hector v2.0 | MAGICC 5.3 | MAGICC 6.0 | FAIR v1.0 | AR5-IR |
| $CO_2$ | 4x Forcing Step | H = 100 yrs | 38% | 35% | 15% | 18% | 73% |
| | Forcing Impulse | H = 100 yrs | 1.0% | -1.2% | 0.25% | -16% | 23% |
| | GHG Emissions | H = 100 yrs | -0.57% | 2.2% | -1.6% | -14% | 31% |
| $CH_4$ | GHG Emissions | H= 20 yrs | -3.1% | -5.6% | 8.7% | -- | 47% |
| BC* | Aerosol Emissions | H = 20 yrs | -9.3% | 1.1% | 8.1% | -- | 19% |

**Table 1: Integrated Temperature Response Differences. The values are the percent difference in time-integrated temperature response compared to the relevant reference (generally comprehensive SCM average, see S9). * For BC specifically, we note that none of the SCMs reflect the temporal response for BC seen in two complex models (Sand et al., 2016; Yang et al., 2019; see S13).**

There are numerous benefits to using simplified models, but the selection of the model should be rooted in a clear understanding of the model responses (see Table 1). Our work illustrates the necessity of using fundamental impulse

 tests to evaluate SCMs and we recommend that modeling communities adopt impulse tests as a standard evaluation suite for any SCM. Given that idealized SCMs are biased in their temporal responses, more comprehensive SCMs could be used for many applications without compromising on accessibility or computational requirements.

**Author contribution.** SJS, CAH, and AKS contributed to experiment design and figure development. AKS performed the experimental simulations and developed the AR5-IR model code in R. BVW and RS developed the ocean model for Hector v2.0 as used in this work. AKS prepared the manuscript with contributions from all co-authors.

**Conflict of Interest.** The authors declare that they have no conflicts of interest.

**Acknowledgements.** This research was supported by the U.S. Department of Energy, Office of Science, as part of research in Multi-Sector Dynamics, Earth and Environmental System Modeling Program, and by the U.S. Environmental Protection Agency. The Pacific Northwest National Laboratory is operated for DOE by Battelle Memorial Institute under contract DE-AC05-76RL01830. The views and opinions expressed in this paper are those of the authors alone.

We acknowledge the World Climate Research Programme's Working Group on Coupled Modelling, which is responsible for CMIP, and we thank the climate modeling groups for producing and making available their model output. For CMIP the U.S. Department of Energy's Program for Climate Model Diagnosis and Intercomparison provides coordinating support and led development of software infrastructure in partnership with the Global Organization for Earth System Science Portals.

**Code Availability**

All model input files generated for our experiments, and the resulting impulse response functions, are provided in the Supplementary Materials or online at https://github.com/akschw04/Fundamental-Impulse-Tests-in-SCMs-Datasets. The authors appreciate that any use of this data be attributed.

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
