# Peer review of "Evaluating Climate Emulation: Fundamental Impulse Testing of Simple Climate Models"

_Earth System Dynamics, 2018_

## Short Comment (SC1) · 1 Oct 2018

The term 'unit test' is widely used in the software engineering community (which I think you are well aware of given your work with Hector on github). Would it possible to refer to your 'unit tests' as something else? Perhaps 'impulse response' tests as used in the paper or 'idealised experiments'? Having the name 'unit test' refer to two completely different things would be an extremely confusing practice to use moving forward (and doesn't seem to be an existing practice as far as I can tell).

---

## Referee Comment (RC1) · 25 Oct 2018

**General comments**

This manuscript presents the responses of a set of climate variables in five different simple climate models (SCMs) to a selected set of impulses. The results of the global temperature response to one of these impulses (a step quadrupling of atmospheric $CO_2$-concentration) is compared to the corresponding responses in an ensemble of CMIP5 Earth System Models (ESMs).

The simple models belong to two categories: the idealized SCMs (AR5-IR and FAIR),

and the comprehensive SCMs (Hector v2.0, MAGICC 5.3, and MAGICC 6.0).

Testing of simple models against more complex ones is interesting and relevant to ESD, but the interpretation of results are difficult, since it is not obvious that a complex model represents specific aspects of reality more correctly than a simple model.

The paper does not seem to present novel concepts, ideas, tools or data. The concept of "unit testing" seems to be a misnomer here, as pointed out in the comment by dr. Nicholls.

The conclusions are not very clear, and the concluding section is very short.

The authors do not present reflections around the assumptions underlying the conclusions.

Model parameters are not given and discussed (not even in the supplement), which has been a source of frustration and confusion for this referee.

Reasonable credit is given to related work.

The title should find another term than "unit testing".

The abstract reflects the content of the paper, apart from the term "unit testing".

The presentation and language is adequate.

**Specific comments**

FAIR is a generalization of AR5-IR to include state dependence of the carbon cycle (Millar et al., 2017). For the experiments shown in Figures 1 and 4 (temperature responses to $CO_2$-forcing), the carbon-cycle module is not active, and from my understanding of the description of FAIR in Millar et al., 2015, the two models should be identical when temperature response to $CO_2$ concentration is simulated. However, in both figures the responses of the two models are very different. If the models are identical in this mode this can only arise from different choices of the time-constant parameters in the simulations of AR5-IR and FAIR. From the figures it looks like the time constants for temperature response in AR5-IR are those used originally by Myhre et al., 2015 (Table 8.SM.11, $d_1 = 8.5$ yr and $d_2 = 409.5$ yr), while in FAIR they look more like the choice of Millar et al. 2017 ($d_1 = 4.1$ yr and $d_2 = 239.0$ yr).

Moreover, if I have got this right, then AR5-IR and FAIR are not only identical models in the simulations shown in Figures 1 and 4, they are also both linear (the nonlinearity in FAIR is in the carbon-cycle module). For a linear response, the time-integrated temperature response shown in Figure 1b and the response to a step forcing shown in Figure 4 are identical, apart from a multiplicative constant depending on the relative strength of the forcings used in Figure 1 and 4. However, in Figure 1b the FAIR response curve is well below the AR5-curve, while in Figure 4 it is well above. For linear, identical models this is possible only if ratio between the climate sensitivities (ECS) of AR5-IR and FAIR is chosen larger in the simulations for Figure 1 than for Figure 4.

In section 3.3 (line 209) the authors write: "Differences between the model responses

to a finite pulse (Fig. 1) and a large concentration step (Fig. 4) demonstrates the expected bias in AR5-IR under larger perturbations." This sentence shows that the authors attribute the different relative response between the two models in Figure 1 and 4 to nonlinear effects in FAIR. While FAIR has a weaker response on decadal time scales than AR5-IR under the the small temperature perturbations in Figure 1, the response is stronger than AR5-IR under the stronger forcing in Figure 4, i.e., if model parameters are unchanged, this amplification must be due to a strong nonlinear feedback. The authors need to clarify the source of this nonlinearity in FAIR.

The total forcing response to $CO_2$ and $CH_4$ emission impulses shown i Figure 2 show quite small spread over the SCMs. Unfortunately the FAIR response is not plotted in that figure, but the AR5-response does not differ drastically from the comprehensive SCMs. This indicates that the carbon-cycle module of the idealized and comprehensive models behave rather similarly. The substantial difference between AR5-IR and the rest appears when the resulting temperature response is displayed in Figure 3a, and also in the temperature response to BC emission in Figure 3b. This is all consistent with Figure 1; the time constant $d_1$ for the temperature response in AR5-IR is too high. Fitting a two-box model to the multimodel mean in the 16 member ESM-ensemble considered by Geoffroy et al., 2013 yields $d_1 = 4.1$ yr, which is about half the e-folding time observed for AR5-IR in Figure 1a and 3b. This supports the assertion that the mismatch between AR5-IR and the other SCMs is just a question of a bad choice of model parameters.

Since no use of observation data is made in this paper, the benchmark to assess the performance of the SCMs are the complex ESMs. The temperature response to a step in BC emission is claimed (in S12) to level off much more slowly in SCMs than in the NorESM model, suggesting that the SCMs do not capture aerosol dynamics correctly, but otherwise the comparison with ESM responses is limited to the ensemble

of $4 \times CO_2$ step forcing simulations. Unfortunately, the spread over the ensemble of ESM responses in Figure 4 is so large that it cannot be used to validate the SCMs. In Figure S22, responses for the three comprehensive ESMs are plotted for two other ECS values, 2.1 and 4.7 degrees. For ECS=2.1, the results are in the mid-range of the ESM-ensemble, while for ECS=4.7 the responses are outside (above) this range. I note, however, that the ESM responses plotted seem to be smaller than typically reported for ESMs. Some of the model runs are also present in the ensemble of Geoffroy et al., 2013, and two of them are possible to recognize in the cloud of response curves. These are the MIROC5 and GISS-E2-R. The MIROC5 run has a characteristic oscillation in the response which is easy to detect in the cloud, and GISS-E2-R is the lower curve in the cloud. For both the temperature values seem to be scaled down by a factor around 0.7 compared with the corresponding curves in Fig. 2 of Geoffroy et al., 2013. The authors should clarify this discrepancy. I notice that if the cloud is adjusted by such a factor, the comprehensive SCM curves (for ECS=3.0 degrees) in Figure 4 will appear much more centered within the range of the ESM cloud.

Table 1 reflects the underlying circular logic in this approach to model testing, a logic that seems to be quite prolific in the modeling community. The performance of the models are ranked according to their deviation from the mean of the three comprehensive SCMs. Is the conclusion that the model closer to this mean is the preferable one?

Finally, I would urge the authors to discuss more explicitly unspoken assumptions underlying their conclusions, and also to make more explicit reference to the results from which these conclusions are drawn. For instance, in the abstract one can read:

Line 17: "While idealized SCMs are widely used, they fail to capture important global

mean climate response features, which can produce biased temperature results."

Since observations are not used in this study, the underlying assumption is that increased model complexity yields more correct results for global response features. This is not obvious. All climate models must be parametrized and constrained against observation. This means parameter fitting, and increased complexity increases the chance of overfitting. Complex models, and ESMs in particular, will to a great extent be parametrized against observations of local processes and not on the global responses. The large spread in the global responses of ESMs is a clear indication that they cannot be used as a substitute for observation of global responses.

Line18: "Comprehensive SCMs, which have non-linear forcing and physically-based carbon-cycle representations, show improved responses compared to idealized SCMs."

Again, a simple model fitted to observation can represent reality better than a more complex model fitted to observation, because overfitting of a complex model may weight real physical processes in an unrealistic manner.

Line 20: "Even some comprehensive SCMs fail to capture response time scales of more complex models under BC or CO2 forcing perturbations."

The BC case may be true, but is based on one single simulation in NorESM. I cannot see where it is shown in the paper that comprehensive SMCs fail to capture response timescales of ESMs to $CO_2$ forcing. This is not apparent in Figure 4.

Line 21: "These results suggest where improvements should be made to SCMs."

It would be very helpful if explicit improvements were suggested.

**Technical comments**

The reference to Chapter 8 in IPCC AR5 WG3 (Myhre et al., 2013) for a description is not very user friendly. It took me a lot of time to identify the relevant part of that chapter and the corresponding Supplement.

In the main manuscript reference to sections, tables, and figures in the supplement are named SI1 etc., while in the supplement itself they are referred to as S1 etc. Be consistent.

On pages 58 and 61 in the supplement is referred to Figure 5 in the main paper. This figure does not exist.

---

## Referee Comment (RC2) · Anonymous Referee #2 · 23 Dec 2018

SCMs are routinely used to emulate state of the art GCMs, and generally display reasonable (though not perfect) agreement when tuned specifically to do so. The authors themselves cite several papers relating to this which discuss strengths and weaknesses of such emulation. While of course SCMs can also be integrated with standard (default) parameter values to provide some guidance as to how the climate system may behave, these simulation will not encapsulate our uncertainty in the best parameter values to use. Furthermore, such simulations will depend greatly on how the default parameter values were chosen, which may differ between SCMs. Given that the GCMs disagree substantially amongst themselves, I do not understand the purpose of this paper in comparing the outputs of standard SCM instances to themselves and GCM output. It is inevitable that these will not match closely when the SCM parame-

ters are set to standard values, and I do not think it is straightforward to attribute such differences to structural limitations of the SCMs without first checking that they cannot be explained by parameter choices. Of course in the simplest of cases one might show that a complex curve output by a sophisticated SCM/GCM simply cannot be explained by a very simple parametric form, but even here it would be appropriate to explore how close a fit could be obtained.

One could reasonably compare SCM responses amongst themselves when tuned to each other or to some common target (either observational or GCM-based). However, this has not been performed here. While in some experiments the sensitivity parameter has been set to a common value of 3, other model parameters appear to differ between the SCMS and were apparently set to standard values which were probably chosen by the SCM authors for a variety of reasons. Thus it is not possible to determine how much of the differences in response are due to model structure, and how much is the result of using different parameter values/tuning strategies.

I would also question whether the relatively unrealistic abrupt tests are a useful diagnostic tool for the model behaviour. While I accept it can be interesting to characterise the response to idealised forcing scenarios, it may be that the differences are much less significant when more realistic scenarios are applied, and the authors acknowledge this point in their conclusions.

Thus, this analysis does not sufficiently advance our understanding of the behaviour of SCMs, and I am sorry to say that I cannot support publication of this manuscript in ESD.

As a minor comment, the "unit testing" terminology seems inappropriate, the test here is rather more comprehensive than such a term usually implies, and furthermore there does not appear to be any clear criteria for success or failure.

---

## Author Comment (AC1) · 7 Feb 2019

**Z. R. Nicholls**
**zebedee.nicholls@climate-energy-college.org**

Dear Mr. Nicholls,

We want to begin by thanking you for taking the time to engage with our manuscript and provide comments. We have replied to your short comment below.

**The term 'unit test' is widely used in the software engineering community (which I think you are well aware of given your work with Hector on github). Would it possible to refer to your 'unit tests' as something else? Perhaps 'impulse response' tests as used in the paper or 'idealised experiments'? Having the name 'unit test' refer to two completely different things would be an extremely confusing practice to use moving forward (and doesn't seem to be an existing practice as far as I can tell).**

We use the phrase "unit testing" with the understanding that this phrase is commonly used in software as we mentioned in the Supplement. Similar to meaning of "unit testing" in software, we are testing the SCM in the simplest way possible, by determining the impulse response of specific model sub-systems such as $CO_2$ and $CH_4$ gas cycles, and the forcing to temperature response of each model. Though we believe our use of the phrase is consistent with its use in software, we will update the language in the manuscript and title to "fundamental impulse tests" to avoid confusion.

---

## Author Comment (AC2) · 7 Feb 2019

**kristoffer.rypdal@uit.no**

Dear Dr. Rypdal,

We want to begin by thanking you for taking the time to read our manuscript and provide comments. We have copied the unedited original comments in bold. Our point-by-point responses are provided in regular font, indented from the original comment for clarity. We will supplement our response with revised text after we have responded to all reviewers and following the ESD process.

**General comments**

**This manuscript presents the responses of a set of climate variables in five different simple climate models (SCMs) to a selected set of impulses. The results of the global temperature response to one of these impulses (a step quadrupling of atmospheric CO2-concentration) is compared to the corresponding responses in an ensemble of CMIP5 Earth System Models (ESMs). The simple models belong to two categories: the idealized SCMs (AR5-IR and FAIR), and the comprehensive SCMs (Hector v2.0, MAGICC 5.3, and MAGICC 6.0).**

We appreciate that you took the time to provide an accurate summary of our work.

**Testing of simple models against more complex ones is interesting and relevant to ESD, but the interpretation of results are difficult, since it is not obvious that a complex model represents specific aspects of reality more correctly than a simple model.**

We appreciate that you agree this work is interesting and relevant to ESD. Comparing simplified models to more complex models is a technique often utilized in the literature (e.g., Joos et al., 2013) and we also employ this technique. We compare the responses of idealized SCMs to comprehensive SCMs and comprehensive SCMs to CMIP5-class models. In our paper, we do not expect individual models to represent reality, but instead rely on the multi-model mean to ground our comparisons. It is well established that the multi-model mean behavior of the complex models replicates well a broad suite of observations (e.g., Figure 9.7, Flato et al. 2013). Our subsequent responses address this comment.

**The paper does not seem to present novel concepts, ideas, tools or data. The concept of "unit testing" seems to be a misnomer here, as pointed out in the comment by dr. Nicholls.**

We strongly believe this paper does present concepts that are new to the literature. Though fundamental impulse tests have been used in the literature, our manuscript employs these existing techniques in a novel way. This is the first study in the literature to rigorously evaluate SCMs using impulse-response tests. SCMs are widely used in the literature and in decision-making context, e.g., within Intergovernmental Panel on Climate Change (IPCC) Reports, coupled with Integrated Assessment Models. In fact, a paper describing a commonly used SCM, MAGICC 6.0, has been cited 371 times in the literature and policy contexts. Another model, the impulse response model used in the IPCC Fifth Assessment Report (AR5-IR), is heavily used by the scientific community to support decision making. Despite their importance, the fundamental responses of SCMs are not fully characterized and we provide a set of tests that we recommend as a standard evaluation suite for any SCM. Further, the U.S. National Academies of

Science (2016) specifically suggested that SCMs be, "assessed on the basis of [the] response to a pulse of emissions," which we do here.

We have added portions of the text above to the revised manuscript introduction to make a more compelling case for our work.

We address the comment about the phrase "unit testing" below.

**The conclusions are not very clear, and the concluding section is very short.**

We will expand the conclusion in the revised manuscript to include a discussion of Table 1, and we copied the revised text into this response below.

**The authors do not present reflections around the assumptions underlying the conclusions.**

We remind the reviewer that we are evaluating the behavior of models and their responses to fundamental impulse-response tests and are not providing information on the underlying mechanisms of the models. The underlying mechanisms are explored by the individual modelling groups in their publications, which we have cited in our manuscript.

**Model parameters are not given and discussed (not even in the supplement), which has been a source of frustration and confusion for this referee.**

We apologize for any confusion in our omission of model parameters. We agree that model parameters are very important for understanding how these models differ. We will add the model parameter files to the supplemental materials so that readers can more easily replicate our results.

**Reasonable credit is given to related work.**

Thank you for the positive comment.

**The title should find another term than "unit testing".**

We use the phrase "unit testing" with the understanding that this phrase is commonly used in software as we mentioned in the Supplement. Similar to meaning of "unit testing" in software, we are testing the SCM in the simplest way possible, by determining the impulse response of specific model sub-systems such as $CO_2$ and $CH_4$ gas cycles, and the forcing to temperature response of each model. Though we believe our use of the phrase is consistent with its use in software, as we replied to the Short Comment, we will update the language in the manuscript and title to "fundamental impulse tests" to avoid confusion.

**The abstract reflects the content of the paper, apart from the term "unit testing".**

Thank you for the comment. We addressed the use of the term "unit testing" in the response above and will instead use the phrase "fundamental impulse tests".

**The presentation and language is adequate.**

Thank you for providing comments on the structure of the paper.

**Specific comments**

**FAIR is a generalization of AR5-IR to include state dependence of the carbon cycle (Millar et al., 2017). For the experiments shown in Figures 1 and 4 (temperature responses to CO2-forcing), the carbon-cycle module is not active, and from my understanding of the description of FAIR in Millar et al., 2015, the two models should be identical when temperature response to CO2 concentration is simulated. However, in both figures the responses of the two models are very different.**

105

We do expect slight differences in the response of FAIR and AR5-IR to a unit forcing. According to Equation 8 in Millar et al., 2017, FAIR will have a differential response to change background $CO_2$ concentrations. By contrast, the AR5-IR response is independent of background concentration.

110

**If the models are identical in this mode this can only arise from different choices of the time-constant parameters in the simulations of AR5-IR and FAIR. From the figures it looks like the time constants for temperature response in AR5-IR are those used originally by Myhre et al., 2015 (Table 8.SM.11, d1 = 8.5 yr and d2 = 409.5 yr), while in FAIR they look more like the choice of Millar et al. 2017 (d1 = 4.1 yr and d2 = 239.0 yr).**

115

As we mentioned above, the FAIR and AR5-IR responses will differ. And we did use the time constant parameters representing the thermal equilibrium of the deep ocean (d2) and the thermal adjustment of the upper ocean (d1) from Myhre et al., 2013 rather than from Millar et al., 2017. We are testing the model responses as they would be 'out of the box' and only make modifications if required for the models to run, as was the case for Hector v1.1 to handle a $4xCO_2$ concentration step.

120

However, to address your comment we have included below additional model responses from the AR5-IR model using parameters from Millar et al., 2017. The parameter choices are available below in Table R1. We will add this information to the Supplement.

125

**Table R1** Parameter values for the simple impulse-response model, AR5-IR

| Parameter (Units) | Value – AR5-IR (from Myhre et al., 2013) | Value – AR5-IR-var (from Millar et al., 2017) | Guiding analogues |
|---|---|---|---|
| $\alpha$ ($Wm^{-2}$) | 5.35 | 5.395 ($\alpha$ = F2x/ln(2); F2x=3.74) | $CO_2$ RF scaling parameter |
| $q_1$ ($KW^{-1}m^2$) | 0.631 | 0.41 | Thermal adjustment of the upper ocean |
| $q_2$ ($KW^{-1}m^2$) | 0.429 | 0.33 | Thermal equilibrium of the deep ocean |
| $d_1$ (year) | 8.4 | 4.1 | Thermal adjustment timescale of the upper ocean |
| $d_2$ (year) | 409.5 | 239.0 | Thermal equilibrium timescale of the deep ocean |

130

Figure R1 shows the temperature response from a $CO_2$ concentration impulse in several SCMs, including the AR5-IR response found using the Millar et al., 2017 time constants, which we refer to as "AR5-IR-Millar-parameters" in this figure. We note that the AR5-IR-parameters response is still not identical to FAIR because FAIR has a differential response to change background $CO_2$ concentrations.

135

[Figure]

**Figure R1** Global mean temperature response (a) and integrated global mean temperature response (b) from a $CO_2$ concentration perturbation in SCMs (MAGICC 6.0 – yellow, MAGICC 5.3 BC-OC – red, Hector v2.0 – blue, AR5-IR – green, FAIR – pink, AR5-IR-Millar-parameters –light blue). The time-integrated response, analogous to the Absolute Global Temperature Potential, is reported as 0-285 years after the perturbation.

**Moreover, if I have got this right, then AR5-IR and FAIR are not only identical models in the simulations shown in Figures 1 and 4, they are also both linear (the nonlinearity in FAIR is in the carbon-cycle module).**

140

The nonlinearity in FAIR is also present in the forcing module based on Millar et al., 2017 Equation 2.

**For a linear response, the time-integrated temperature response shown in Figure 1b and the response to a step forcing shown in Figure 4 are identical, apart from a multiplicative constant depending on the relative**
145 **strength of the forcings used in Figure 1 and 4. However, in Figure 1b the FAIR response curve is well below the AR5-curve, while in Figure 4 it is well above. For linear, identical models this is possible only if ratio between the climate sensitivities (ECS) of AR5-IR and FAIR is chosen larger in the simulations for Figure 1 than for Figure 4.**

150 We used consistent ECS values throughout our experiments, unless otherwise noted, and we do want to thank you for your careful comments. We made an error in applying the 4x$CO_2$ concentration step in the AR5-IR model, which resulted in the response being significantly lower than it should have been. Figure R2 in our response provides the updated results and is consistent with Figure 1b. We have updated the manuscript and supplement to reflect the amended figure, and we note that this change does not impact
155 our overall conclusions that, "Fundamental forcing tests, such as a 4x$CO_2$ concentration step, show that the SCMs used here have a faster warming rate in this strong forcing regime compared to more complex models. However, comprehensive SCM responses are similar to more complex models under smaller, more realistic perturbations (Joos et al., 2013)."

[Figure]

Global Mean Temperature Response from 4xCO₂ Concentration Step

**Figure R2** Global mean temperature response from $4xCO_2$ concentration step in CMIP5 models (grey) and SCMs (MAGICC 6.0 – yellow, MAGICC 5.3 BC-OC – red, Hector v2.0 – blue, FAIR – pink, AR5-IR –green). A climate sensitivity value of 3°C was used in the SCMs and the thick lines represent CMIP5 models with an ECS between 2.5 - 3.5 °C.

**In section 3.3 (line 209) the authors write: "Differences between the model responses to a finite pulse (Fig. 1) and a large concentration step (Fig. 4) demonstrates the expected bias in AR5-IR under larger perturbations." This sentence shows that the authors attribute the different relative response between the two models in Figure 1 and 4 to nonlinear effects in FAIR. While FAIR has a weaker response on decadal time scales than AR5-IR under the the small temperature perturbations in Figure 1, the response is stronger than AR5-IR under the stronger forcing in Figure 4, i.e., if model parameters are unchanged, this amplification must be due to a strong nonlinear feedback. The authors need to clarify the source of this nonlinearity in FAIR.**

We apologize for the confusion, which we believe it is resolved by updating Figure 4 in the manuscript with Figure R2 in this response. The source of the nonlinearity in FAIR is in the forcing component.

**The total forcing response to CO2 and CH4 emission impulses shown i Figure 2 show quite small spread over the SCMs. Unfortunately the FAIR response is not plotted in that figure, but the AR5- response does not differ drastically from the comprehensive SCMs. This indicates that the carbon-cycle module of the idealized and comprehensive models behave rather similarly. The substantial difference between AR5-IR and the rest appears when the resulting temperature response is displayed in Figure 3a, and also in the temperature response to BC emission in Figure 3b. This is all consistent with Figure 1; the time constant d1 for the temperature response in AR5-IR is too high. Fitting a two-box model to the multimodel mean in the 16 member ESM-ensemble considered by Geoffroy et al., 2013 yields d1 = 4.1 yr, which is about half the e-folding time observed for AR5-IR in Figure 1a and 3b. This supports the assertion that the mismatch between AR5-IR and the other SCMs is just a question of a bad choice of model parameters.**

As we mentioned above, we tested these models using their default parameter values unless a change was required for the model to successfully complete an experiment. Though we take the reviewer's point about the importance of parameter choice, we note that the definitions and meanings of each parameter are not consistent across the SCMs used in this manuscript. For example, using the ocean component as an example we find that the vertical diffusivity parameter is not defined in the same way across the comprehensive SCMs, and is completely absent from the idealized SCMs where it is implicitly represented by the parametrized ocean timescale values.

**Since no use of observation data is made in this paper, the benchmark to assess the performance of the SCMs are the complex ESMs. The temperature response to a step in BC emission is claimed (in S12) to level off much more slowly in SCMs than in the NorESM model, suggesting that the SCMs do not capture aerosol**

200 **dynamics correctly, but otherwise the comparison with ESM responses is limited to the ensemble of $4 \times CO2$ step forcing simulations. Unfortunately, the spread over the ensemble of ESM responses in Figure 4 is so large that it cannot be used to validate the SCMs.**

205 We first point out that our primary purpose in this paper is to evaluate the fundamental behavior of the simple climate models. We do this by both comparing them to each other, and also, in the limited cases where this is possible, to more complex models (Joos et al., 2013). We compare against the suite of complex model results because it has been shown that the multi-model mean behavior of the complex models replicates well a broad suite of observations (e.g., Figure 9.7, Flato et al. 2013). Also see the next response, below.

210

**In Figure S22, responses for the three comprehensive ESMs are plotted for two other ECS values, 2.1 and 4.7 degrees. For ECS=2.1, the results are in the mid-range of the ESM-ensemble, while for ECS=4.7 the responses are outside (above) this range.**

215 We changed the ECS values in the SCMs to illustrate the effects of parameter selection on the model responses. We found that spanning the range of complex model ECS values still resulted in stronger SCM responses, which supports the conclusion in our main paper that the SCMs have a faster warming rate under strong forcing regimes compared to more complex models. We revised the supplemental text around Figure S22 to state this as well.

220

**Table 1 reflects the underlying circular logic in this approach to model testing, a logic that seems to be quite prolific in the modeling community. The performance of the models are ranked according to their deviation from the mean of the three comprehensive SCMs. Is the conclusion that the model closer to this mean is the**

225 **preferable one?**

We have moved amended text from the supplement to the main paper to better describe the logic behind our conclusions as represented in Table 1. We do, indeed, find that – at least amongst the simple models examined – the physically based comprehensive SCMs generally respond better than more simplified

230 models such as AR5 or FAIR. As we clarify in the text, this is largely a relative assessment of the responses between the SCMs.

"By using fundamental impulse tests, we found that idealized SCMs using sums of exponentials often fail to capture the responses of more complex models. SCMs that include representations of non-linear processes,

235 such as FAIR, show improved responses, though these models still do not perform as well as comprehensive SCMs with physically-based representations. Fundamental forcing tests, such as a $4xCO_2$ concentration step, show that the SCMs used here have a faster warming rate in this strong forcing regime compared to more complex models. However, comprehensive SCM responses are similar to more complex models under smaller, more realistic perturbations (Joos et al., 2013).

240

It is not possible to compare these fundamental responses with observations, and it is even more difficult to compare SCMs with the more complex models at decadal time horizons due to internal variability (e.g. Joos et al., 2013, Figure 2a). However, it is common in the climate modeling literature to use the multi-model mean as a base comparison. In fact, the CMIP5 multi-model mean has been shown to capture

245 observational trends (among other climate variables) better than any individual complex model (Flato et al. 2013).

Thus, we use the comprehensive SCM multi-model mean to compare to the individual model responses. It is our conclusion that the model response closer to the multi-model mean is more accurately representing that particular response pattern. We illustrate this assumption by using the scale developed for Table 1, which generally uses the time-integrated temperature response percent difference from the comprehensive SCM average. We set the scale based on the range in percent differences found in our analysis: ••• : 0-10% difference, •• : 10-20% difference, and • : 20-30% difference from the comprehensive SCM average (S9).

For example, we assign the comprehensive SCM responses to a $CO_2$ concentration impulse a three (•••) because the responses are within 10% of the comprehensive SCM average. The idealized SCMs, FAIR v1.0 and AR5-IR, have greater differences and are given a two (••) and a one (•), respectively.

Under the 4x$CO_2$ concentration step experiment, we can compare the SCM response to more complex models from CMIP5. We assign MAGICC 6.0 a three (•••) because it appears to respond more reasonably under stronger forcing conditions than the other SCMs. We assign Hector v2.0, MAGICC 5.3, and FAIR a two (••) because these SCMs have initially quicker responses to an abrupt 4xCO2 concentration increase compared to the ESMs. We assign AR5-IR a one (•) because it has a slower response to an abrupt 4xCO2 concentration increase and is insensitive to changing background concentrations.

For $CH_4$ emissions impulses, we use the difference from the comprehensive SCM average to rate the responses. Unlike the 100GtC $CO_2$ and 4x$CO_2$ step experiments, we cannot compare the SCM responses to more complex models, therefore, we are more lenient in our performance assignment against the comprehensive SCM average. $CH_4$ is a well-mixed GHG and, therefore, we expect that the climate system response to $CH_4$ concentration perturbations will be similar to that for $CO_2$. However, it would be useful to evaluate in more complex models if the simple representation of chemistry in the comprehensive SCMs adequately represents the time evolution of $CH_4$ concentrations in response to a change in emissions.

Finally, we assign ratings to the SCM responses to aerosols. We do not explicitly conduct aerosol experiments other than BC because the responses of the SCMs to other aerosols will be similar to their response to BC. We do not have a definitive reference for the time-dependent response to aerosol forcing perturbations. Instead, we rate the SCMs using the difference from the average of both MAGICC models, which both differentiate aerosol forcing between land and ocean, which results in a faster overall climate response to aerosols as compared to greenhouse gases (Shindell et al., 2014).

In the case of BC, we note that all SCM response ratings should be reduced from the values shown because they do not accurately represent the temporal response to a BC step found in an ESM (S12). A more definitive evaluation of climate system responses to aerosol perturbations would be useful. This would require additional GCM simulations to step emission changes for various aerosol species and/or forcing mechanisms. There are currently two studies that have conducted this test, one study specifically investigated NorESM's response to black carbon (BC) perturbations (Sand et al., 2016) and a more recent study that conducted similar BC perturbations in CESM (Yang et al., 2018 *in discussion*).

| Impulse | Species | Model | | | | |
|---------|---------|-----------|-----------|-----------|----------|--------|
| | | Hector v2.0 | MAGICC 5.3 | MAGICC 6.0 | FAIR v1.0 | AR5-IR |
| Forcing | $CO_2$ impulse | ••• | ••• | ••• | •• | • |
| Forcing | $4xCO_2$ step | •• | •• | ••• | •• | • |
| GHG Emissions | $CO_2$ | ••• | ••• | ••• | •• | • |
| GHG Emissions | $CH_4$ | ••• | ••• | ••• | -- | •• |
| Aerosols* | $SO_2$, BC | •• | ••• | ••• | -- | • |

*Table 2: Summary of SCM Performance. The performance scale is generally based on the maximum percent difference in time-integrated temperature response compared to the relevant reference (generally comprehensive SCM average in SI 9). ••• : 0-10%, •• : 10-20%, • : 20-30% difference (SI13). * This ranking refers to aerosol response in general, which do not differ substantially for different aerosol types in these models. For BC specifically, all ratings should be reduced since none of the SCMs accurately represent the temporal response for BC seen in ESMs (Sand et al., 2016) (SI12).*

There are numerous benefits to using simplified models, but the selection of the model should be rooted in a clear understanding of the model responses (see Table 1). Our work illustrates the necessity of using fundamental impulse tests to evaluate SCMs and we recommend that modeling communities adopt them as a standard validation suite for any SCM. Given that idealized SCMs are biased in their response patterns, more comprehensive SCMs could be used for many applications without compromising on accessibility or computational requirements."

**I note, however, that the ESM responses plotted seem to be smaller than typically reported for ESMs. Some of the model runs are also present in the ensemble of Geoffroy et al., 2013, and two of them are possible to recognize in the cloud of response curves. These are the MIROC5 and GISS-E2-R. The MIROC5 run has a characteristic oscillation in the response which is easy to detect in the cloud, and GISS-E2-R is the lower curve in the cloud. For both the temperature values seem to be scaled down by a factor around 0.7 compared with the corresponding curves in Fig. 2 of Geoffroy et al., 2013. The authors should clarify this discrepancy. I notice that if the cloud is adjusted by such a factor, the comprehensive SCM curves (for ECS=3.0 degrees) in Figure 4 will appear much more centered within the range of the ESM cloud.**

Conducting impulse tests with complex models is computationally expensive, illustrated by the few studies employing this technique to understand the responses of models. We cite the Sand et al., 2016 study that specifically investigated NorESMs response to black carbon (BC) perturbations (Sand et al., 2016). We now include another study that conducted similar BC perturbations in CESM (Yang et al., 2018 *in discussion*). Other stylized CMIP5 experiments, such as the 1% $CO_2$ concentration experiment, are not

included in our comparison because we do not consider them to be impulse response tests. It is not possible to cleanly extract the impulse response from the 1% experiments. The CMIP5 $4xCO_2$ concentration step experiment is mathematically related to impulse responses, so are a reasonable comparison, particularly because these are the largest suite of such tests conducted in complex models, which is the reason we highlight these results in the paper.

Geoffroy et al., 2013 reported the $4xCO_2$ concentration step temperature change relative to the 150-year temperature mean from the corresponding pre-industrial control run. For comparison to the simple models, we report the drift corrected (see S3) $4xCO_2$ concentration step temperature change relative to the start of the $4xCO_2$ concentration run. Therefore, there will be a difference in the temperature reported. We included this additional information in the revised supplement to clarify the difference in the way modeled temperature change is reported.

Figure R3 shows the global mean temperature response from the $4xCO_2$ concentration step experiment for the 20 CMIP5 models used in our comparison following the Geoffroy et al. (2013) procedure of reporting the $4xCO_2$ concentration step temperature change relative to the 150-year temperature mean from the corresponding pre-industrial control run. The responses reported in Figure R3 are consistent with Geoffroy et al. (2013). We expanded the number of complex models and updated the supplementary materials accordingly.

[Figure]

**Figure R3** Global mean temperature response from $4xCO_2$ concentration step in 20 CMIP5 models.

**I cannot see where it is shown in the paper that comprehensive SMCs fail to capture response timescales of ESMs to CO2 forcing. This is not apparent in Figure 4.**

To clarify, in Figure 4 of our manuscript the rate of temperature response from the SCMs immediately following the $4xCO_2$ step is generally faster than the rate of temperature response from the ESMs. We also illustrate this in Figure S22 where we will expand the discussion in the revised manuscript, as we mentioned above. From this, we conclude that some SCMs do not capture the response timescales of ESMs.

**Finally, I would urge the authors to discuss more explicitly unspoken assumptions underlying their conclusions, and also to make more explicit reference to the results from which these conclusions are drawn. For instance, in the abstract one can read:**

**Line 17: "While idealized SCMs are widely used, they fail to capture important global mean climate response features, which can produce biased temperature results."**

Our language was vague in the abstract and we provide revised text to more explicitly reference our results.

"We find that while idealized SCMs are widely used, they fail to capture the magnitude and timescales of global mean climate responses under emissions perturbations, which can produce biased temperature results."

**Since observations are not used in this study, the underlying assumption is that increased model complexity yields more correct results for global response features. This is not obvious. All climate models must be parametrized and constrained against observation. This means parameter fitting, and increased complexity increases the chance of overfitting. Complex models, and ESMs in particular, will to a great extent be parametrized against observations of local processes and not on the global responses. The large spread in the global responses of ESMs is a clear indication that they cannot be used as a substitute for observation of global responses.**

We disagree with the reviewer that the large spread in ESM global mean temperature responses means they are not useful. While some climate studies benefit from using observations, we cannot employ observations to compare with impulse response tests, as we mentioned above. As noted previously, ESMs are constrained by more detailed representations of the relevant physics (e.g. energy balance, heat transport, etc.) and the multi-model mean of ESMs does a better job of matching observations than any individual ESM. The suite of ESMs results are, therefore, one of the best (albeit not perfect by any means) tools by which we can compare SCMs.

**Line18: "Comprehensive SCMs, which have non-linear forcing and physically-based carbon-cycle representations, show improved responses compared to idealized SCMs."**

**Again, a simple model fitted to observation can represent reality better than a more complex model fitted to observation, because overfitting of a complex model may weight real physical processes in an unrealistic manner.**

While it is true that a simple model may fit observations better than a more complex model, we do not agree that this is an indication that the fit represents a better representation of reality. This may also mean that, due to a lack of physical constraints in an overly simplified model, a good fit is obtained for the wrong reasons. We again point out the long-standing finding that the multi-model mean for CMIP-class models better represents reality as compared to any individual model. This finding indicates that the physical processes represented in these models (some explicit, some parameterized) are providing meaningful constraints on the behavior of the coupled system.

In our experience, the overall results of these global models, such as global temperature change, are not fitted to observational datasets. Instead, individual components are developed and tested against appropriate observations (e.g., top of atmosphere radiative flux, cloud properties, laboratory measurements, etc.), which provides an emergent, aggregate model behavior (albeit, dependent on the properties of these numerous sub-systems.). Every GCM is wrong, at least in some specific aspects, but the evidence suggest that the behavior of these models taken together is a useful overall constraint on Earth system responses (Flato et al. 2013).

These impulse response tests allow us to determine the underlying dynamics of SCMs so as to better elucidate any potential issues with later analysis using these models. For example, a SCM with a faster overall temperature response to a forcing would return a different implied value of any fitting parameters (such as climate sensitivity) than a model with a slower fundamental response.

**Line 20: "Even some comprehensive SCMs fail to capture response time scales of more complex models under BC or CO2 forcing perturbations."**

**The BC case may be true, but is based on one single simulation in NorESM.**

There are now two studies that have conducted a BC emissions impulse in complex models (i.e., Sand et al., 2016 and Yang et al., 2018) and cited them above. We noted above that the Sand et al., 2016 study specifically investigated NorESMs response to black carbon (BC) perturbations (Sand et al., 2016), while another conducted similar BC perturbations in CESM (Yang et al., 2018 *in discussion*). Further, Shindell et al. (2014) concluded that without accounting for regional warming and feedbacks, simple models could overestimate aerosol impacts, though we note that some models such as MAGICC 5.3 and MAGICC 6.0 do have differential land-ocean and North-South hemisphere forcing.

**Line 21: "These results suggest where improvements should be made to SCMs."**
**It would be very helpful if explicit improvements were suggested.**

We avoided adding explicit suggestions on areas where SCMs could be improved because modeling groups have a variety of reasons for implementing different features and components in their models. We stated in our manuscript that "Given that idealized SCMs are biased in their response patterns, more comprehensive SCMs could be used for many applications without compromising on accessibility or computational requirements." Some modeling groups favor answering certain scientific questions versus flexibility versus computational intensity differently, for instance, and the purpose of our paper is to explore mechanism for assessing those differences to inform users. Nonetheless, we expanded the conclusion in our response to more fully discuss the scale used Table 1 and we believe this expanded discussion suggests areas of improvement.

**Technical comments**

**The reference to Chapter 8 in IPCC AR5 WG3 (Myhre et al., 2013) for a description is not very user friendly. It took me a lot of time to identify the relevant part of that chapter and the corresponding Supplement.**

We have added additional details in our citations of Chapter 8 in the IPCC AR5 for clarity. The manuscript and supplement have been updated.

**In the main manuscript reference to sections, tables, and figures in the supplement are named SI1 etc., while in the supplement itself they are referred to as S1 etc. Be consistent.**

Thank you for identifying this error. We have updated the manuscript to be consistent with the supplement.

**On pages 58 and 61 in the supplement is referred to Figure 5 in the main paper. This figure does not exist.**

Thank you for identifying this error. The reference should be to Figure 4, and we apologize for any confusion this might have caused. The supplement has been updated.

475

Citations

Flato, G., J. Marotzke, B. Abiodun, P. Braconnot, S.C. Chou, W. Collins, P. Cox, F. Driouech, S. Emori, V.
Eyring, C. Forest, P. Gleckler, E. Guilyardi, C. Jakob, V. Kattsov, C. Reason and M. Rummukainen. (2013).
480     Evaluation of Climate Models. In: *Climate Change 2013: The Physical Science Basis. Contribution of
Working Group I to the Fifth Assessment Report of the Intergovernmental Panel on Climate Change*
[Stocker, T.F., D. Qin, G.-K. Plattner, M. Tignor, S.K. Allen, J. Boschung, A. Nauels, Y. Xia, V. Bex and P.M.
Midgley (eds.)]. Cambridge University Press, Cambridge, United Kingdom and New York, NY, USA

485     Geoffroy, Olivier, et al. "Transient climate response in a two-layer energy-balance model. Part I: Analytical
solution and parameter calibration using CMIP5 AOGCM experiments." *Journal of Climate* 26.6 (2013):
1841-1857.

Joos, F., Roth, R., Fuglestvedt, J. S., Peters, G. P., Enting, I. G., Bloh, W. V., ... & Friedrich, T. (2013). Carbon
490     dioxide and climate impulse response functions for the computation of greenhouse gas metrics: a multi-
model analysis. *Atmospheric Chemistry and Physics*, *13*(5), 2793-2825.

Millar, R. J., Nicholls, Z. R., Friedlingstein, P., & Allen, M. R. (2017). A modified impulse-response
representation of the global near-surface air temperature and atmospheric concentration response to
495     carbon dioxide emissions. *Atmospheric Chemistry and Physics*, *17*(11), 7213-7228.

Myhre, G., Shindell, D., Bréon, F. M., Collins, W., Fuglestvedt, J., Huang, J., ... & Nakajima, T. (2013).
Anthropogenic and natural radiative forcing. *Climate change*, *423*, 658-740.

500     Sand, M., Berntsen, T. K., Von Salzen, K., Flanner, M. G., Langner, J., & Victor, D. G. (2016). Response of
Arctic temperature to changes in emissions of short-lived climate forcers. *Nature Climate Change*, *6*(3),
286.

Shindell, D. T. (2014). Inhomogeneous forcing and transient climate sensitivity. *Nature Climate Change*,
505     *4*(4), 274.

Yang, Y., Wang, H., Smith, S. J., Ma, P. L., & Rasch, P. J. (2017). Source attribution of black carbon and its
direct radiative forcing in China. *Atmospheric Chemistry & Physics*, *17*(6).

510

---

## Author Response (AR1)

Dear Editor,

Thank you for inviting us to submit a revised manuscript. Below we have summarized the important changes made to the manuscript in response to the reviewer and editor comments:

1. We have rewritten portions of the manuscript to clarify the purpose of our paper and better articulate the potential impact of our work.
2. We removed references to unit testing throughout the text and in the title to eliminate confusion. We replaced the phrase with "impulse response tests," which is more descriptive and used elsewhere in the literature.
3. We revised the methodology section to include a clearer description of the fundamental impulse tests conducted in our study and the parameters used in the SCMs.
4. We included the input parameter files with the supplementary data files, which are now available on Github.
5. We made an error when applying the 4xCO$_2$ concentration step in the AR5-IR model and, thus, updated Figure 4 in the manuscript and any numerical references to those results in both the main paper and supplement. This change did not impact the overall conclusions of our paper.
6. We revised Table 1 in the manuscript to report the integrated temperature responses of the SCMs to the different perturbations, rather than a scale of performance. We believe this change addresses reviewer concerns about the concluding section.
7. We added large portions of text to the concluding section of the revised manuscript to describe the results reported in the revised Table 1.
8. Several minor text changes were made to improve readability and understanding.
9. We added and updated several citations to both the manuscript and supplement.
10. We added several sections to the supplement based on the reviewer comments and our responses including: S2.4, which explores the differences between AR5-IR and FAIR; S3.1, which explores the CMIP5 data used in the manuscript in greater detail; and S11.2, which explores additional sensitivity experiments in the SCMs using MAGICC6.0-derived parameters.

**Response to Reviewer #1**

**Comment: This manuscript presents the responses of a set of climate variables in five different simple climate models (SCMs) to a selected set of impulses. The results of the global temperature response to one of these impulses (a step quadrupling of atmospheric CO2-concentration) is compared to the corresponding responses in an ensemble of CMIP5 Earth System Models (ESMs). The simple models belong to two categories: the idealized SCMs (AR5-IR and FAIR), and the comprehensive SCMs (Hector v2.0, MAGICC 5.3, and MAGICC 6.0).**

Response: We appreciate that you took the time to provide an accurate summary of our work.

*Changes in Manuscript: None.*

**Comment: Testing of simple models against more complex ones is interesting and relevant to ESD, but the interpretation of results are difficult, since it is not obvious that a complex model represents specific aspects of reality more correctly than a simple model.**

Response: We appreciate that you agree this work is interesting and relevant to ESD. Comparing simplified models to more complex models is a technique often utilized in the literature (e.g., Joos et al., 2013) and we also employ this technique. We compare the responses of idealized SCMs to comprehensive SCMs and comprehensive SCMs to CMIP5-class models. In our paper, we do not necessarily expect individual models to represent reality, but instead rely on the multi-model mean to ground our comparisons. It is well established that the multi-model mean behavior of the complex models replicates a broad suite of observations better than any individual model (e.g., Figure 9.7, Flato et al. 2013). Our subsequent responses will also address this comment.

*Changes in Manuscript: We added text to clarify our comparison of the individual models to the multi-model mean.*

**Comment: The paper does not seem to present novel concepts, ideas, tools or data. The concept of "unit testing" seems to be a misnomer here, as pointed out in the comment by dr. Nicholls.**

Response: We strongly believe this paper does present a specific application of concepts that are new to the literature. Though fundamental impulse tests have been used in the literature, our manuscript employs these existing techniques in a novel way. This is the first study in the literature to rigorously evaluate SCMs using impulse-response tests. SCMs are widely used in the literature and in decision-making context, e.g., within Intergovernmental Panel on Climate Change (IPCC) Reports, coupled with Integrated Assessment Models. In fact, a paper describing a commonly used SCM, MAGICC 6.0, has been cited 371 times in the literature. Another model, the impulse response model used in the IPCC Fifth Assessment Report (AR5-IR), is heavily used by the scientific community to support decision making. Despite their importance, the fundamental responses of SCMs are not fully characterized and we provide a set of tests that we recommend as a standard evaluation suite for any SCM. Further, the U.S. National Academies of Science (2016) specifically suggested that SCMs be, "assessed on the basis of [the] response to a pulse of emissions," which we do here.

We have added portions of the text above to the revised manuscript introduction to make a more compelling case for our work.

We address the comment about the phrase "unit testing" below.

*Changes in Manuscript: We added language to the manuscript that better articulates the potential impact of our work. Some of this language is derived from this response*

**Comment: The conclusions are not very clear, and the concluding section is very short.**

Response: We will expand the conclusion in the revised manuscript to include a discussion of Table 1, and we copied the revised text into this response below.

*Changes in Manuscript: We amended Table 1 in the conclusion section and elaborated on our findings more thoroughly in the revised manuscript.*

**Comment: The authors do not present reflections around the assumptions underlying the conclusions.**

Response: We remind the reviewer that we are evaluating the behavior of models and their responses to fundamental impulse-response tests and are not providing information on the underlying mechanisms of the models. The underlying mechanisms are explored by the individual modelling groups in their publications, which we have cited in our manuscript.

*Changes in Manuscript: We added text to the revised manuscript clarifying that the purpose of our paper to employ impulse response tests to evaluate the behavior of the SCMs.*

**Comment: Model parameters are not given and discussed (not even in the supplement), which has been a source of frustration and confusion for this referee.**

Response: We apologize for any confusion in our omission of model parameters. We agree that model parameters are very important for understanding how these models differ. We will add the model parameter files to the supplemental materials so that readers can more easily replicate our results.

*Changes in Manuscript: We have added the parameter input files for each SCM to the supplementary data.*

**Comment: Reasonable credit is given to related work.**

Response: Thank you for the positive comment.

*Changes in Manuscript: None.*

**Comment: The title should find another term than "unit testing".**

Response: We use the phrase "unit testing" with the understanding that this phrase is commonly used in software as we mentioned in the Supplement. Similar to meaning of "unit testing" in software, we are testing the SCM in the simplest way possible, by determining the impulse response of specific model sub-systems such as $CO_2$ and $CH_4$ gas cycles, and the forcing to temperature response of each model. Though we believe our use of the phrase is consistent with its use in software, as we replied to the Short Comment, we will update the language in the manuscript and title to "fundamental impulse tests" to avoid confusion.

*Changes in Manuscript: Throughout the revised manuscript, including in the title, we now refer to our tests as "fundamental impulse tests."*

**Comment: The abstract reflects the content of the paper, apart from the term "unit testing".**

Response: Thank you for the comment. We addressed the use of the term "unit testing" in the response above and will instead use the phrase "fundamental impulse tests".

*Changes in Manuscript: None that were not already adopted.*

**Comment: The presentation and language is adequate.**

Response: Thank you for providing comments on the structure of the paper.

*Changes in Manuscript: None.*

**Comment: FAIR is a generalization of AR5-IR to include state dependence of the carbon cycle (MiIlar et al., 2017). For the experiments shown in Figures 1 and 4 (temperature responses to CO2-forcing), the carbon-cycle module is not active, and from my understanding of the description of FAIR in Millar et al., 2015, the two models should be identical when temperature response to CO2 concentration is simulated. However, in both figures the responses of the two models are very different.**

Response: We do expect slight differences in the response of FAIR and AR5-IR to a unit forcing. According to Equation 8 in Millar et al., 2017, FAIR will have a differential response to change background $CO_2$ concentrations. By contrast, AR5-IR parameterizes the climate response to a unit forcing, $R_T$, using a sum of exponentials as given by Equation 8.SM.13 in Myhre et al., 2013:

$$R_T(t) = \sum_j \frac{q_j}{d_j} e^{\frac{-t}{d_j}}$$

Values of the $R_T$ input parameters, $q_j$ and $d_j$, are available in Table R1 of this response, where j=1,2 represent the timescales of the fast and slow ocean response. We note all parameters *are* independent of background concentration in AR5.

*Changes in Manuscript: We added specific references to the differences between FAIR and AR5-IR to the manuscript and included the language from our response above in the supplement (see S2.4).*

**Comment: If the models are identical in this mode this can only arise from different choices of the time-constant parameters in the simulations of AR5-IR and FAIR. From the figures it looks like the time constants for temperature response in AR5-IR are those used originally by Myhre et al., 2015 (Table 8.SM.11, d1 = 8.5 yr and d2 = 409.5 yr), while in FAIR they look more like the choice of Millar et al. 2017 (d1 = 4.1 yr and d2 = 239.0 yr).**

Response: As we mentioned above, the FAIR and AR5-IR responses will differ. And we did use the time constant parameters representing the thermal equilibrium of the deep ocean (d2) and the thermal adjustment of the upper ocean (d1) from Myhre et al., 2013 rather than from Millar et al., 2017. We are testing the model responses as they would be 'out of the box' and only make modifications if required for the models to run, as was the case for Hector v1.1 to handle a $4xCO_2$ concentration step.

However, to address your comment we have included below additional model responses from the AR5-IR model using parameters from Millar et al., 2017. The parameter choices are available below in Table R1. We will add this information to the Supplement.

**Table R1** Parameter values for the simple impulse-response model, AR5-IR

| Parameter (Units) | Value – AR5-IR (from Myhre et al., 2013) | Value – AR5-IR-var (from Millar et al., 2017) | Guiding analogues |
|---|---|---|---|
| $\alpha$ (Wm$^{-2}$) | 5.35 | 5.395 ($\alpha$ = F2x/ln(2); F2x=3.74) | $CO_2$ RF scaling parameter |
| $q_1$ (KW$^{-1}$m$^2$) | 0.631 | 0.41 | Thermal adjustment of the upper ocean |
| $q_2$ (KW$^{-1}$m$^2$) | 0.429 | 0.33 | Thermal equilibrium of the deep ocean |
| $d_1$ (year) | 8.4 | 4.1 | Thermal adjustment timescale of the upper ocean |
| $d_2$ (year) | 409.5 | 239.0 | Thermal equilibrium timescale of the deep ocean |

Figure R1 shows the temperature response from a $CO_2$ concentration impulse in several SCMs, including the AR5-IR response found using the Millar et al., 2017 time constants, which we refer to as "AR5-IR-Millar-parameters" in this figure. We note that the AR5-IR-parameters response is still not identical to FAIR because FAIR has a differential response to change background $CO_2$ concentrations.

We note that, while the Millar et al., 2017 parameters in table R1 may provide a better short-term fit, they underestimate the long-term response of the ocean. The long-term ocean thermal time scale, which can only be estimated using multi-century model runs, is known to be longer than 200 years from basic physical principles (as seen in the original literature cited by the AR5 model, which used longer model runs to inform those parameters). While this may be an acceptable tradeoff if this model is only going to be used over a 100-year timescale, this will inevitably lead to bias on longer and longer time-scales. The simple climate models tested in this study are used for a variety of purposes and over a range of time-scales. This illustrates why we use the original parameters of the models as set by their designers.

[Figure]

**Figure R1** Global mean temperature response (a) and integrated global mean temperature response (b) from a $CO_2$ concentration perturbation in SCMs (MAGICC 6.0 – yellow, MAGICC 5.3 BC-OC – red, Hector v2.0 – blue, AR5-IR – green, FAIR – pink, AR5-IR-Millar-parameters –light blue). The time-integrated response, analogous to the Absolute Global Temperature Potential, is reported as 0-285 years after the perturbation.

*Changes in Manuscript: This response was added in its entirety to the revised supplement (see S2.4) to make the differences between AR5-IR and FAIR easier for readers to identify and understand.*

**Comment: Moreover, if I have got this right, then AR5-IR and FAIR are not only identical models in the simulations shown in Figures 1 and 4, they are also both linear (the nonlinearity in FAIR is in the carbon-cycle module).**

Response: A nonlinearity is also present in FAIR based on Millar et al., 2017 Equation 8 as noted above.

*Changes in Manuscript: None that were not already adopted.*

**Comment: For a linear response, the time-integrated temperature response shown in Figure 1b and the response to a step forcing shown in Figure 4 are identical, apart from a multiplicative constant depending on the relative strength of the forcings used in Figure 1 and 4. However, in Figure 1b the FAIR response curve is well below the AR5-curve, while in Figure 4 it is well above. For linear, identical models this is possible only if ratio between the climate sensitivities (ECS) of AR5-IR and FAIR is chosen larger in the simulations for Figure 1 than for Figure 4.**

Response: We used consistent ECS values throughout our experiments, unless otherwise noted, and we do want to thank you for your careful comments. We made an error in applying the 4xCO₂ concentration step in the AR5-IR model, which resulted in the response being significantly lower than it should have been. Figure R2 in our response provides the updated results and is consistent with Figure 1b. We have updated the manuscript and supplement to reflect the amended figure, and we note that this change does not impact our overall conclusions that, "Fundamental forcing tests, such as a 4xCO₂ concentration step, show that the SCMs used here have a faster warming rate in this strong forcing regime compared to more complex models. However, comprehensive SCM responses are similar to more complex models under smaller, more realistic perturbations (Joos et al., 2013)."

[Figure]

**Figure R2** Global mean temperature response from 4xCO$_2$ concentration step in CMIP5 models (grey) and SCMs (MAGICC 6.0 – yellow, MAGICC 5.3 BC-OC – red, Hector v2.0 – blue, FAIR – pink, AR5-IR – green, AR5-IR-Millar-parameters – light blue). A climate sensitivity value of 3°C was used in the SCMs and the thick lines represent CMIP5 models with an ECS between 2.5 - 3.5 °C.

*Changes in Manuscript: Based on this comment, in the revised manuscript we updated Figure 4 after mistakenly applying the 4xCO$_2$ concentration step in AR5-IR, which did not impact our over conclusions. We updated figures and numerical results in the main paper and supplement to reflect this update to Figure 4 and added the language mentioned in the response above.*

**Comment: In section 3.3 (line 209) the authors write: "Differences between the model responses to a finite pulse (Fig. 1) and a large concentration step (Fig. 4) demonstrates the expected bias in AR5-IR under larger perturbations." This sentence shows that the authors attribute the different relative response between the two models in Figure 1 and 4 to nonlinear effects in FAIR. While FAIR has a weaker response on decadal time scales than AR5-IR under the the small temperature perturbations in Figure 1, the response is stronger than AR5-IR under the stronger forcing in Figure 4, i.e., if model parameters are unchanged, this amplification must be due to a strong nonlinear feedback. The authors need to clarify the source of this nonlinearity in FAIR.**

Response: We apologize for the confusion, which we believe it is resolved by updating Figure 4 in the manuscript with Figure R2 in this response. The source of the nonlinearity in FAIR is in the forcing component.

*Changes in Manuscript: None that were not already adopted.*

**Comment: The total forcing response to CO2 and CH4 emission impulses shown i Figure 2 show quite small spread over the SCMs. Unfortunately the FAIR response is not plotted in that figure, but the AR5- response does not differ drastically from the comprehensive SCMs. This indicates that the carbon-cycle module of the idealized and comprehensive models behave rather similarly. The substantial difference between AR5-IR and the rest appears when the resulting temperature response is displayed in Figure 3a, and also in the temperature response to BC emission in Figure 3b. This is all consistent with Figure 1; the time constant d1 for the temperature response in AR5-IR is too high. Fitting a two-box model to the multimodel mean in the 16 member ESM-ensemble considered by Geoffroy et al., 2013 yields d1 = 4.1 yr, which is about half the e-folding time observed for AR5-IR in Figure 1a and 3b. This supports the assertion that the mismatch between AR5-IR and the other SCMs is just a question of a bad choice of model parameters.**

Response: As we mentioned above, we tested these models using their default parameter values unless a change was required for the model to successfully complete an experiment. Though we take the reviewer's point about the importance of parameter choice, we note that the definitions and meanings of each parameter are not consistent across the SCMs used in this manuscript. For example, using the ocean component as an example we find that the vertical diffusivity parameter is not defined in the same way across the comprehensive SCMs, and is completely absent from the idealized SCMs where it is implicitly represented by the parametrized ocean timescale values. This is why we use the models as parameterized "out of the box" in the same manner that users of these models do.

*Changes in Manuscript: In the methodology section of the revised manuscript, we added a paragraph specifically discussing the parameter choices, and added additional explanation in the revised supplement (see S2).*

**Comment: Since no use of observation data is made in this paper, the benchmark to assess the performance of the SCMs are the complex ESMs. The temperature response to a step in BC emission is claimed (in S12) to level off much more slowly in SCMs than in the NorESM model, suggesting that the SCMs do not capture aerosol dynamics correctly, but otherwise the comparison with ESM responses is limited to the ensemble of 4 × CO2 step forcing simulations. Unfortunately, the spread over the ensemble of ESM responses in Figure 4 is so large that it cannot be used to validate the SCMs.**

Response: We first point out that our primary purpose in this paper is to evaluate the fundamental behavior of the simple climate models. We do this by both comparing them to each other, and also, in the limited cases where this is possible, to more complex models (Joos et al., 2013). We compare against the suite of complex model results because it has been shown that the multi-model mean behavior of the complex models replicates a broad suite of observations better than any individual model (e.g., Figure 9.7, Flato et al. 2013). Also see the next response, below.

*Changes in Manuscript: We added some of this response language to the revised manuscript conclusion section.*

**Comment: In Figure S22, responses for the three comprehensive ESMs are plotted for two other ECS values, 2.1 and 4.7 degrees. For ECS=2.1, the results are in the mid-range of the ESM-ensemble, while for ECS=4.7 the responses are outside (above) this range.**

Response: We changed the ECS values in the SCMs to illustrate the effects of parameter selection on the model responses. We found that spanning the range of complex model ECS values still resulted in stronger SCM responses, which supports the conclusion in our main paper that the SCMs have a faster warming rate under strong forcing regimes compared to more complex models. We revised the supplemental text around Figure S22 to state this as well.

In response to Reviewer #2, we have also expanded the revised supplement to include the effects of climate sensitivity and ocean diffusivity parameter selection on the SCM responses.

*Changes in Manuscript: We amended the text around the figures to note that the SCMs have a faster warming rate under strong forcing regimes compared to more complex models. We also added the discussion from Reviewer #2 to the supplement (see S11.2).*

**Comment: Table 1 reflects the underlying circular logic in this approach to model testing, a logic that seems to be quite prolific in the modeling community. The performance of the models are ranked according to their deviation from the mean of the three comprehensive SCMs. Is the conclusion that the model closer to this mean is the preferable one?**

Response: We have moved amended text from the supplement to the main paper to better describe the logic behind our conclusions as represented in Table 1. We do, indeed, find that – at least amongst the simple models examined – the physically based comprehensive SCMs generally respond better than more simplified models such as AR5 or FAIR. As we clarify in the revised conclusion text below, which is updated in our manuscript, this is largely a relative assessment of the responses between the SCMs, but also in comparison to more complex models where this is possible.

By using fundamental impulse tests, we found that idealized SCMs using sums of exponentials often fail to capture the responses of more complex models. SCMs that include representations of non-linear processes, such as FAIR, show improved responses, though these models still do not perform as well as comprehensive SCMs with physically-based representations. Fundamental forcing tests, such as a $4xCO_2$ concentration step, show that the SCMs used here have a faster warming rate in this strong forcing regime compared to more complex models. However, comprehensive SCM responses are similar to more complex models under smaller, more realistic perturbations (Joos et al., 2013).

It is not possible to compare these fundamental responses with observations, and it is even more difficult to compare SCMs with the more complex models at decadal time horizons due to internal variability (e.g. Joos et al., 2013, Figure 2a). However, it is common in the climate modeling literature to use the multi-model mean as a base comparison. In fact, the CMIP5 multi-model mean has been shown to capture observational trends (among other climate variables) better than any individual complex model (Flato et al. 2013).

Thus, we use the comprehensive SCM multi-model mean to compare to the individual model responses for many of our experiments. We use the CMIP5 multi-model mean, developed using only those complex models with comparable climate sensitivity values to the SCMs (S9), to compare the SCM responses from a $4xCO_2$ forcing step. It is our conclusion that the model response closer to the multi-model mean more accurately represent that particular response pattern. We illustrate this assumption by reporting the time-integrated temperature response percent difference from the relevant multi-model mean in Table 1 (S9).

We note that the comprehensive SCM responses to a $CO_2$ concentration impulse are within 2% of the comprehensive SCM average, while the idealized SCMs, FAIR v1.0 and AR5-IR, have greater differences 100 years after the pulse.

Under the $4xCO_2$ concentration step experiment, we can compare the SCM responses to more complex models from CMIP5. MAGICC 6.0 appears to respond more reasonably under stronger forcing conditions than the other SCMs 100 years after those pulse, though only marginally better than FAIR. Hector v2.0 and MAGICC 5.3 have an initially quicker responses to an abrupt 4xCO2 concentration increase compared to the ESMs. AR5-IR has too strong a response to an abrupt 4xCO2 concentration increase and is insensitive to changing background concentrations.

For $CH_4$ emissions impulses, we use the difference from the comprehensive SCM average to rate the responses. $CH_4$ is a well-mixed GHG and, therefore, we expect that the climate system response to $CH_4$ concentration perturbations will be similar to that for $CO_2$. However, it would be useful to evaluate in more complex models to determine if the simple representation of chemistry in the comprehensive SCMs adequately represents the time evolution of $CH_4$ concentrations in response to a change in emissions.

Finally, we do not have a definitive reference for the time-dependent response to BC forcing perturbations. Instead, we compare the SCMs using the difference from the average of both MAGICC models, which both differentiate aerosol forcing between land and ocean, which results in a faster overall climate response to aerosols as compared to greenhouse gases (Shindell et al., 2014; Sand et al., 2016; Yang et al. 2018 *accepted for publication*). And in the case of BC, we note that the SCM responses should be taken critically because they do not accurately represent the temporal response to a BC step found in ESMs. A more definitive evaluation of climate system responses to aerosol perturbations would be useful. This would require additional GCM simulations to step emission changes for various aerosol species and/or forcing mechanisms. There are currently two studies that have conducted this test, one study specifically investigated NorESM's response to black carbon (BC) perturbations (Sand et al., 2016) and a more recent study that conducted similar BC perturbations in CESM (Yang et al., 2018 *accepted for publication*).

| Species | Impulse | Time After Pulse | Percent Integrated Temperature Response Differences for each Simple Climate Model (%) | | | | |
|---|---|---|---|---|---|---|---|
| | | | Hector v2.0 | MAGICC 5.3 | MAGICC 6.0 | FAIR v1.0 | AR5-IR |
| $CO_2$ | 4x Forcing Step | H = 100 yrs | 38% | 35% | 15% | 18% | 73% |
| | Forcing Impulse | H = 100 yrs | 1.0% | -1.2% | 0.25% | -16% | 23% |
| | GHG Emissions | H = 100 yrs | -0.57% | 2.2% | -1.6% | -14% | 31% |
| $CH_4$ | GHG Emissions | H= 20 yrs | -3.1% | -5.6% | 8.7% | -- | 47% |
| BC | Aerosol Emissions* | H = 20 yrs | -9.3% | 1.1% | 8.1% | -- | 19% |

Table 2: Percent Integrated Temperature Response Differences. The values are the percentage difference in time-integrated temperature response (%) compared to the relevant reference (generally comprehensive SCM average in SI 9). * For BC specifically, we note that none of the SCMs accurately represent the temporal response for BC seen in ESMs (Sand et al., 2016) (SI12).

There are numerous benefits to using simplified models, but the selection of the model should be rooted in a clear understanding of the model responses (see Table 1). Our work illustrates the necessity of using fundamental impulse tests to evaluate SCMs and we recommend that modeling communities adopt them as a standard validation suite for any SCM. Given that idealized SCMs are biased in their response patterns, more comprehensive SCMs could be used for many applications without compromising on accessibility or computational requirements.

*Changes in Manuscript: We added the majority of this response to the main paper conclusion, which more clearly explains the differences in the model responses and emphasizes the importance of conducting fundamental impulse test to evaluate SCMs.*

**Comment: I note, however, that the ESM responses plotted seem to be smaller than typically reported for ESMs. Some of the model runs are also present in the ensemble of Geoffry et al., 2013, and two of them are possible to recognize in the cloud of response curves. These are the MIROC5 and GISS-E2-R. The MIROC5**

**run has a characteristic oscillation in the response which is easy to detect in the cloud, and GISS-E2-R is the lower curve in the cloud. For both the temperature values seem to be scaled down by a factor around 0.7 compared with the corresponding curves in Fig. 2 of Geoffroy et al., 2013. The authors should clarify this discrepancy. I notice that if the cloud is adjusted by such a factor, the comprehensive SCM curves (for ECS=3.0 degrees) in Figure 4 will appear much more centered within the range of the ESM cloud.**

Response: Conducting impulse tests with complex models is computationally expensive, illustrated by the few studies employing this technique to understand the responses of models. We cite the Sand et al., 2016 study that specifically investigated NorESMs response to black carbon (BC) perturbations (Sand et al., 2016). We now include another study that conducted similar BC perturbations in CESM (Yang et al., 2018 *accepted for publication*). Other stylized CMIP5 experiments, such as the 1% $CO_2$ concentration experiment, are not included in our comparison because we do not consider them to be impulse response tests. It is not possible to cleanly extract the impulse response from the 1% experiments. The CMIP5 $4xCO_2$ concentration step experiment is mathematically related to impulse responses, so are a reasonable comparison, particularly because these are the largest suite of such tests conducted in complex models, which is the reason we highlight these results in the paper.

Geoffroy et al., 2013 reported the $4xCO_2$ concentration step temperature change relative to the 150-year temperature mean from the corresponding pre-industrial control run. For comparison to the simple models, we report the drift corrected (see S3) $4xCO_2$ concentration step temperature change relative to the start of the $4xCO_2$ concentration run. Therefore, there will be a difference in the temperature reported. We included this additional information in the revised supplement to clarify the difference in the way modeled temperature change is reported.

Figure R3 shows the global mean temperature response from the $4xCO_2$ concentration step experiment for the 20 CMIP5 models used in our comparison following the Geoffroy et al. (2013) procedure of reporting the $4xCO_2$ concentration step temperature change relative to the 150-year temperature mean from the corresponding pre-industrial control run. The responses reported in Figure R3 are consistent with Geoffroy et al. (2013). We expanded the number of complex models and updated the supplementary materials accordingly.

[Figure]

**Figure R3** Global mean temperature response from $4xCO_2$ concentration step in 20 CMIP5 models.

*Changes in Manuscript: We added the majority of this response to the revised supplement (see S3.1).*

**Comment: I cannot see where it is shown in the paper that comprehensive SMCs fail to capture response timescales of ESMs to CO2 forcing. This is not apparent in Figure 4.**

Response: To clarify, in Figure 4 of our manuscript the rate of temperature response from the SCMs immediately following the 4xCO$_2$ step is generally faster than the rate of temperature response from the ESMs. We also illustrate this in Figure S22 where we will expand the discussion in the revised manuscript, as we mentioned above. From this, we conclude that some SCMs do not capture the response timescales of ESMs.

*Changes in Manuscript: None that were not already adopted.*

**Comment: Finally, I would urge the authors to discuss more explicitly unspoken assumptions underlying their conclusions, and also to make more explicit reference to the results from which these conclusions are drawn. For instance, in the abstract one can read:**

**Line 17: "While idealized SCMs are widely used, they fail to capture important global mean climate response features, which can produce biased temperature results."**

Response: Our language was vague in the abstract and we provide revised text to more explicitly reference our results.

"We find that while idealized SCMs are widely used, they fail to capture the magnitude and timescales of global mean climate responses under emissions perturbations, which can produce biased temperature results."

*Changes in Manuscript: We amended the abstract with the language in this response.*

**Comment: Since observations are not used in this study, the underlying assumption is that increased model complexity yields more correct results for global response features. This is not obvious. All climate models must be parametrized and constrained against observation. This means parameter fitting, and increased complexity increases the chance of overfitting. Complex models, and ESMs in particular, will to a great extent be parametrized against observations of local processes and not on the global responses. The large spread in the global responses of ESMs is a clear indication that they cannot be used as a substitute for observation of global responses.**

Response: We disagree with the reviewer that the large spread in ESM global mean temperature responses means they are not useful. While some climate studies benefit from using observations, we cannot employ observations to compare with impulse response tests, as we mentioned above. As noted previously, ESMs are constrained by more detailed representations of the relevant physics (e.g. energy balance, heat transport, etc.) and the multi-model mean of ESMs does a better job of matching observations than any individual ESM. The suite of ESMs results are, therefore, one of the best (albeit not perfect by any means) tools by which we can compare SCMs.

*Changes in Manuscript: None that were not already adopted.*

**Comment: Line18: "Comprehensive SCMs, which have non-linear forcing and physically-based carbon-cycle representations, show improved responses compared to idealized SCMs."**

**Again, a simple model fitted to observation can represent reality better than a more complex model fitted to observation, because overfitting of a complex model may weight real physical processes in an unrealistic manner.**

Response: While it is true that a simple model may fit observations better than a more complex model, we do not agree that this is an indication that the fit represents a better representation of reality. This may also mean that, due to a lack of physical constraints in an overly simplified model, a good fit is obtained for the wrong reasons. We again point out the long-standing finding that the multi-model mean for CMIP-class models better represents reality as compared to any individual model. This finding indicates that the physical processes represented in these models (some explicit, some parameterized) are providing meaningful constraints on the behavior of the coupled system.

In our experience, the overall results of these global models, such as global temperature change, are not fitted to observational datasets. Instead, individual components are developed and tested against appropriate observations (e.g., top of atmosphere radiative flux, cloud properties, laboratory measurements, etc.), which provides an emergent, aggregate model behavior (albeit, dependent on the properties of these numerous sub-systems.). Every GCM is wrong, at least in some specific aspects, but the evidence suggests that the behavior of these models taken together is a useful overall constraint on Earth system responses (Flato et al. 2013).

These impulse response tests allow us to determine the underlying dynamics of SCMs so as to better elucidate any potential issues with later analysis using these models. For example, a SCM with a faster overall temperature response to a forcing would return a different implied value of any fitting parameters (such as climate sensitivity) than a model with a slower fundamental response.

*Changes in Manuscript: None that were not already adopted.*

**Comment: Line 20: "Even some comprehensive SCMs fail to capture response time scales of more complex models under BC or CO2 forcing perturbations."**

**The BC case may be true, but is based on one single simulation in NorESM.**

Response: There are now two studies that have conducted a BC emissions impulse in complex models (i.e., Sand et al., 2016 and Yang et al., 2018) and cited them above. We noted above that the Sand et al., 2016 study specifically investigated NorESMs response to black carbon (BC) perturbations (Sand et al., 2016), while another conducted similar BC perturbations in CESM (Yang et al., 2018). Further, Shindell et al. (2014) concluded that without accounting for regional warming and feedbacks, simple models can mis-represent the timescale for aerosol impacts, though we note that some models such as MAGICC 5.3 and MAGICC 6.0 do have differential land-ocean and North-South hemisphere forcing that better represent the response of the climate system to aerosols (Smith et al., 2014).

*Changes in Manuscript: We amended the manuscript and supplement to include references to the literature exploring model responses to black carbon.*

**Comment: Line 21: "These results suggest where improvements should be made to SCMs."**
**It would be very helpful if explicit improvements were suggested.**

Response: We avoided adding explicit suggestions on areas where SCMs could be improved because modeling groups have a variety of reasons for implementing different features and components in their models. We stated in our manuscript that "Given that idealized SCMs are biased in their response patterns, more comprehensive SCMs could be used for many applications without compromising on accessibility or computational requirements." Some modeling groups favor answering certain scientific questions versus flexibility versus computational intensity differently, for instance, and the purpose of our paper is to explore mechanism for assessing those differences to inform users. Nonetheless, we expanded the conclusion in our response to more fully discuss the scale used Table 1 and we believe this expanded discussion suggests areas of improvement.

*Changes in Manuscript: We amended the conclusion accordingly.*

**Comment: The reference to Chapter 8 in IPCC AR5 WG3 (Myhre et al., 2013) for a description is not very user friendly. It took me a lot of time to identify the relevant part of that chapter and the corresponding Supplement.**

Response: We have added additional details in our citations of Chapter 8 in the IPCC AR5 for clarity. The manuscript and supplement have been updated.

*Changes in Manuscript: We amended the manuscript and supplement accordingly.*

**Comment: In the main manuscript reference to sections, tables, and figures in the supplement are named SI1 etc., while in the supplement itself they are referred to as S1 etc. Be consistent.**

Response: Thank you for identifying this error. We have updated the manuscript to be consistent with the supplement.

*Changes in Manuscript: We amended the manuscript and supplement accordingly.*

**Comment: On pages 58 and 61 in the supplement is referred to Figure 5 in the main paper. This figure does not exist.**

Response: Thank you for identifying this error. The reference should be to Figure 4, and we apologize for any confusion this might have caused. The supplement has been updated.

*Changes in Manuscript: We amended the supplement accordingly.*

Citations from Response to Reviewer #1

Flato, G., Marotzke, J., Abiodun, B., Braconnot, P., Chou, S. C., Collins, W., Cox, P., Driouech, F., Emori, S., Eyring, V., Forest, C., Gleckler, P., Guilyardi, E., Jakob, C., Kattsov, V., Reason, C. and Rummukainen, M.: Evaluation of Climate Models, Clim. Chang. 2013 Phys. Sci. Basis. Contrib. Work. Gr. I to Fifth Assess. Rep. Intergov. Panel Clim. Chang., 741–866, doi:10.1017/CBO9781107415324, 2013.
Friedlingstein, P., Meinshausen, M., Arora, V. K., Jones, C. D., Anav, A., Liddicoat, S. K. and Knutti, R.: Uncertainties in CMIP5 climate projections due to carbon cycle feedbacks, J. Clim., 27(2), 511–526, doi:10.1175/JCLI-D-12-00579.1, 2014.

Geoffroy, O., Saint-martin, D., Olivié, D. J. L., Voldoire, A., Bellon, G. and Tytéca, S.: Transient climate response in a two-layer energy-balance model. Part I: Analytical solution and parameter calibration using CMIP5 AOGCM experiments, J. Clim., 26(6), 1841–1857, doi:10.1175/JCLI-D-12-00195.1, 2013.

Joos, F., Roth, R., Fuglestvedt, J. S., Peters, G. P., Enting, I. G., Von Bloh, W., Brovkin, V., Burke, E. J., Eby, M., Edwards, N. R., Friedrich, T., Frölicher, T. L., Halloran, P. R., Holden, P. B., Jones, C., Kleinen, T., Mackenzie, F. T., Matsumoto, K., Meinshausen, M., Plattner, G. K., Reisinger, A., Segschneider, J., Shaffer, G., Steinacher, M., Strassmann, K., Tanaka, K., Timmermann, A. and Weaver, A. J.: Carbon dioxide and climate impulse response functions for the computation of greenhouse gas metrics: A multi-model analysis, Atmos. Chem. Phys., 13(5), 2793–2825, doi:10.5194/acp-13-2793-2013, 2013.

Millar, J. R., Nicholls, Z. R., Friedlingstein, P. and Allen, M. R.: A modified impulse-response representation of the global near-surface air temperature and atmospheric concentration response to carbon dioxide emissions, Atmos. Chem. Phys., 17(11), 7213–7228, doi:10.5194/acp-17-7213-2017, 2017.

Myhre, G., Shindell, D., Bréon, F.-M., Collins, W., Fuglestvedt, J., Huang, J., Koch, D., Lamarque, J.-F., Lee, D., Mendoza, B., Nakajima, T., Robock, A., Stephens, G., Takemura, T. and Zhang, H.: Anthropogenic and Natural Radiative Forcing, Clim. Chang. 2013 Phys. Sci. Basis. Contrib. Work. Gr. I to Fifth Assess. Rep. Intergov. Panel Clim. Chang., 659–740, doi:10.1017/ CBO9781107415324.018, 2013.

Sand, M., Berntsen, T. K., Von Salzen, K., Flanner, M. G., Langner, J. and Victor, D. G.: Response of Arctic temperature to changes in emissions of short-lived climate forcers, Nat. Clim. Chang., 6(3), 286–289, doi:10.1038/nclimate2880, 2016.

Smith, S. J., Wigley, T. M. L., Meinshausen, M. & Rogelj, J. 2014. Questions of bias in climate models. Nature Clim. Change, **4**, 741-742. doi: 10.1038/nclimate2345

Shindell, Drew T. 2014. "Inhomogeneous Forcing and Transient Climate Sensitivity." *Nature Climate Change* 4(4): 18–21.

Yang, Y., Wang, H., Smith, S. J., Ma, P. L., & Rasch, P. J. 2018. Source attribution of black carbon and its direct radiative forcing in China. *Atmospheric Chemistry & Physics*, *17*(6).

**Reply to Reviewer #2**

**Comment: SCMs are routinely used to emulate state of the art GCMs, and generally display reasonable (though not perfect) agreement when tuned specifically to do so. The authors themselves cite several papers relating to this which discuss strengths and weaknesses of such emulation. While of course SCMs can also be integrated with standard (default) parameter values to provide some guidance as to how the climate system may behave, these simulation will not encapsulate our uncertainty in the best parameter values to use. Furthermore, such simulations will depend greatly on how the default parameter values were chosen, which may differ between SCMs.**

Response: The aim of this paper is not to validate any individual simple climate model (SCM), nor the range of parameters used in the SCMs, which are also explored in the literature cited in our manuscript as you note. Rather, we are evaluating the fundamental behavior of the simple models. However, we do agree that understanding the uncertainty associated with our results is important and based on the comments here and from Reviewer #1, we have now included the parameter files in the supplement so show the default parameters of the models.

In our original supplement we conducted a simple sensitivity test for the $4xCO_2$ concentration step experiment by changing the climate sensitivity values in the three comprehensive SCMs used in this paper. Based on your concerns, we have added some additional tests to the supplement of our paper by exploring a range of climate sensitivity values and ocean diffusivity values in MAGICC 6.0 under a unit pulse of $CO_2$ emissions and a unit pulse of $CO_2$ concentration.

We selected climate sensitivity and ocean diffusivity values from the parameter ranges presented in Table 1B in Meinshausen et al., 2011. The values are the native MAGICC 6.0 parameters required to emulate complex models used in CMIP3 using three calibrated parameters (climate sensitivity, ocean diffusivity, and land/ocean warming). We provided the climate sensitivity and ocean diffusivity value ranges we explored in Table R2 below.

**Table R2** MAGICC 6.0 parameter values from Meinshausen et al., 2011 Table 1B for sensitivity tests

| Scenario | Climate sensitivity (K) | Ocean diffusivity ($cm^2s^{-1}$) |
|---|---|---|
| Base Case | 3.0 | 1.1 |
| High Ocean diffusivity | 3.0 | 3.74 |
| Low Ocean diffusivity | 3.0 | 0.50 |
| High Climate sensitivity | 6.03 | 1.1 |
| Low Climate sensitivity | 1.94 | 1.1 |

Figure R4 shows the global mean temperature response exploring the range of ocean diffusivity (Kz) (a) and global mean temperature response exploring the range of climate sensitivity (CS) (b)

under a $CO_2$ emissions perturbation. Figure R5 shows the same results for under a $CO_2$ concentration pulse. Both figures illustrate that climate sensitivity has the greatest impact on the responses and in our manuscript, we accounted for this and used similar climate sensitivity values in SCMs where possible, unless otherwise noted in the supplemental figures.

[Figure]

**Figure R4** Global mean temperature response exploring the range of ocean diffusivity (Kz) (a) and Global mean temperature response exploring the range of climate sensitivity (CS) (b) from a $CO_2$ emissions perturbation in SCMs (MAGICC 6.0 – yellow, MAGICC 5.3 BC-OC – red, Hector v2.0 – blue, AR5-IR – green, FAIR –pink, AR5-IR-Millar-parameters –light blue). The grey shaded region in each figure shows the range in MAGICC 6.0 responses found using the Table R2 parameters. We note that the range of responses exploring CS (b) are normalized to account for the different climate conditions under difference CS values.

[Figure]

**Figure R5** Global mean temperature response exploring the range of ocean diffusivity (Kz) (a) and Global mean temperature response exploring the range of climate sensitivity (CS) (b) from a $CO_2$ concentration pulse in SCMs (MAGICC 6.0 – yellow, MAGICC 5.3 BC-OC – red, Hector v2.0 – blue, AR5-IR – green, FAIR –pink, AR5-IR-Millar-parameters –light blue). The grey shaded region in each figure shows the range in MAGICC 6.0 responses found using the Table R2 parameters. We note that the range of responses exploring CS (b) are normalized to account for the different climate conditions difference CS values.

We acknowledge, however, that vertical ocean diffusivity has a large impact on ocean heat uptake and we do note that this parameter selection also impacts the responses in the SCMs, particular under a $CO_2$ emissions pulse (Meinshausen et al., 2011). However, the SCMs we compare in our paper either do not have the same definitions of vertical ocean diffusivity, as is the case for the comprehensive SCMs, or ocean diffusivity is not directly represented in the models, as is the case for idealized SCMs. For our purposes, therefore, we kept the ocean diffusivity values at their default values within the comprehensive SCMs. By exploring the uncertainty in ocean diffusivity, we have, in fact, bolstered the main conclusions of our manuscript.

For completeness we also acknowledge that Meinshausen et al., 2011 spanned ranges of land/ocean warming contrast (RLO) in the three-parameter calibration described in Table 1B of their manuscript. And again, the SCMs either use the same values of RLO, as is the case for both versions of MAGICC, or this parameter is not represented in the idealized models. In fact, from our work using impulse response test to characterize SCMs, we concluded that SCMs without differential warming do not correctly capture the response pattern to BC perturbations.

*Changes in Manuscript: We added this response in its entirety to the revised supplement (see S11.2).*

**Comment: Given that the GCMs disagree substantially amongst themselves, I do not understand the purpose of this paper in comparing the outputs of standard SCM instances to themselves and GCM output. It is inevitable that these will not match closely when the SCM parameters are set to standard values, and I do not think it is straightforward to attribute such differences to structural limitations of the SCMs without first checking that they cannot be explained by parameter choices.**

Response: We remind the reviewer that we are not attempting to emulate GCMs in our paper. Instead, we evaluate SCM responses by comparing the models to themselves and also, in the limited cases where this is possible, to more complex models.

One key purpose of this paper is to determine the fundamental response of these models by conducting impulse response tests (as recommended, for example, by a recent report by the NAS). This has not been done before, and this alone provides useful information on the behavior of these models, how this differs between the models, and the magnitude of those differences. Given the extremely widespread use of these models, this is a critical task.

We go beyond just comparing these models to each other, by comparing against the suite of complex model results were this is possible. We do this because it has been shown that the multi-model mean behavior of the complex models replicates well a broad suite of observations (e.g., Figure 9.7, Flato et al. 2013). Further, comparing simple models to complex models is a common technique employed in the literature (e.g., Joos et al., 2013). For example, in the abrupt 4 X CO2 regime, we find that the SCMs as a group initially respond more quickly than the GCMs. We conclude from this that there must be some physical processes represented in the GCMs that buffers the initial response in this regime that is lacking in the SCMs.

From our exploration of the range of ocean diffusivity values and climate sensitivity values above, we have found that the response differences we noted in our conclusions cannot be explained by parameter choice alone. We appreciate that the reviewer brought up this important question, and we will amend our paper discussion based on these results but note that our main conclusions remain. In fact, we believe that illustrating the ranges in responses further bolsters our claim that impulse response tests are needed to fully understand model behavior.

*Changes in Manuscript: We amended the manuscript to clarify the purpose of our paper and better articulates the potential impact of our work, referencing the National Academies for example. We used text from this response in the manuscript.*

**Comment: Of course in the simplest of cases one might show that a complex curve output by a sophisticated SCM/GCM simply cannot be explained by a very simple parametric form, but even here it would be appropriate to explore how close a fit could be obtained.**

Response: Fitting individual simple models to more complex models is generally explored by the individual SCM development teams, and we cite papers from the Hector, MAGICC, and FAIR model development teams which explore their respective model's ability to fit a GCM (or the multi-model mean) with a given set of parameters. While emulation is outside the scope of this paper, but to address the reviewer's comments, above we have expanded our impulse tests to include an uncertainty test which relies on parameters derived from GCM emulation experiments using MAGICC 6.0. We will also add a discussion of fitting SCMs to more complex models to the paper to address the points raised by the reviewer.

*Changes in Manuscript: We addressed the sensitivity experiment in S11.2.*

**Comment: One could reasonably compare SCM responses amongst themselves when tuned to each other or to some common target (either observational or GCM-based). However, this has not been performed here. While in some experiments the sensitivity parameter has been set to a common value of 3, other model parameters appear to differ between the SCMS and were apparently set to standard values which were probably chosen by the SCM authors for a variety of reasons. Thus it is not possible to determine how much of the differences in response are due to model structure, and how much is the result of using different parameter values/tuning strategies.**

Response: We remind the reviewer that the goal of our paper is to evaluate the SCMs, as we mentioned above, and we ultimately suggest a suite of fundamental impulse-response tests using realistic backgrounds for use in SCM development. We make this clearer in our revised manuscript by including some of the text mentioned above, such as:

"In our paper, we evaluated the SCMs by comparing the models to themselves and also, in the limited cases where this is possible, to more complex models. We compare against the suite of complex model results because it has been shown that the multi-model mean behavior of the complex models replicates well a broad suite of observations (e.g., Figure 9.7, Flato et al. 2013)."

The sensitivity tests we have added to the paper do provide useful general information on how parameter choices might influence model responses, which addresses part of the question posed above. (We note also, that in response to reviewer 1, we have also added an example of how the idealized AR5 model response changes with a change in parameters.) Our text will be amended to reflect these results. However, as we noted above, due to structural differences in the SCMs it is, in general, not possible to operate the models with identical parameter values. This reinforces the importance of conducting fundamental impulse response tests to quantify the behavior of the SCMs.

*Changes in Manuscript: We addressed the sensitivity experiment in S11.2 and amended the text as stated.*

**Comment: I would also question whether the relatively unrealistic abrupt tests are a useful diagnostic tool for the model behaviour. While I accept it can be interesting to characterise the response to idealised forcing scenarios, it may be that the differences are much less significant when more realistic scenarios are applied, and the authors acknowledge this point in their conclusions.**

Response: There is a long history of doing just such idealized, abrupt tests to evaluate model behavior. The CMIP5 $4xCO_2$ concentration step experiments are the largest suite of impulse response tests conducted in complex models, for example, which is the reason we highlight these results in the paper.

The impulse response tests conducted enable us to uncover differences in model behavior that are not apparent when running standard, multi-emission scenarios. Indeed, one of the important uses of SCMs is to conduct model experiments were there may be relatively small changes in emissions between two scenarios. Because SCMs do not exhibit internal variability, such experiments can be used to quantify such changes. Impulse response tests also allow us to understand, on a more fundamental level, differences between SCMs that have been found comparing simulations with more conventional scenarios (e.g., van Vuuren et al., 2011).

*Changes in Manuscript: None that were not already adopted.*

**Comment: Thus, this analysis does not sufficiently advance our understanding of the behaviour of SCMs, and I am sorry to say that I cannot support publication of this manuscript in ESD.**

Response: We believe this paper does present novel and useful results that are new to the literature. Though fundamental impulse tests have been used a few times in the literature in this context, our manuscript employs these existing techniques in a new manner. This is the first study in the literature to rigorously evaluate SCMs using impulse-response tests. SCMs are widely used in the literature and in decision-making context, e.g., within Intergovernmental Panel on Climate Change (IPCC) Reports, coupled with Integrated Assessment Models. In fact, a paper describing a commonly used SCM, MAGICC 6.0, has been cited 371 times in the literature and policy contexts. Another model, the impulse response model used in the IPCC Fifth Assessment Report (AR5-IR), is heavily used by the scientific community to support decision making. Despite their importance, the fundamental responses of SCMs are not fully characterized. The U.S. National Academies of Science (2016) specifically suggested that SCMs be, "assessed on the basis of [the] response to a pulse of emissions," which we do here. Additionally, we provide a set of tests that we recommend as a standard evaluation suite for any SCM.

*Changes in Manuscript: We added some text from this response to the amended manuscript, such as citing the National Academies and clarifying that our manuscript employs existing techniques in a new manner.*

**Comment: As a minor comment, the "unit testing" terminology seems inappropriate, the test here is rather more comprehensive than such a term usually implies, and furthermore there does not appear to be any clear criteria for success or failure.**

Response: We received several comments on our use of the phrase "unit testing". Though we believe our use of the phrase is consistent with its use in software, as we replied to the Short Comment and Reviewer #1, we will update the language to "fundamental impulse tests" to avoid confusion.

*Changes in Manuscript: None that were not already adopted based on Reviewer #1.*

Citations from Response to Reviewer #2

[revised manuscript text omitted]

**Contents**

**S1 Supplementary Method**

We conduct perturbations of three contrasting chemical species: carbon dioxide ($CO_2$), methane ($CH_4$), and black carbon (BC). We begin with $CO_2$ because this well-mixed greenhouse gas is the largest contributor to anthropogenic forcing changes (Myhre et al., 2013). Methane is also of interest because it is a shorter-lived greenhouse gas, with chemical interactions with itself and other species (Cicerone, R.J.; Oremland, 1988). Finally, we use BC perturbations to represent aerosols more generally because we are interested in model responses to short-lived climate forcers (Bond et al., 2013; Harmsen et al., 2015). SCM representations of other aerosols species are similar so we do not conduct impulse tests of other species.

The comprehensive SCMs we use are readily comparable because they read in similar emissions. Background trajectory emissions are taken from the published Representative Concentration Pathway (RCP) 4.5 scenario (Thomson et al., 2011) database, which means that all calculations in the main paper are conducted relative to a changing $CO_2$ concentration background unless otherwise noted. SCMs are often used to project global mean temperature over various future scenarios, so this is the most relevant type of background on which to test these models.

Conducting these experiment with a constant $CO_2$ background, as previously used in the literature (Joos et al., 2013), requires inverse modeling of the individual models to produce constant $CO_2$ concentration emissions files. Our methodology is easier to implement as a regular unit test as it only requires the same emissions inputs with no inverse calculations needed. We provide the input emission files used in this paper.

In many SCMs, forcing over historical periods is explicitly calibrated to a model base year, so it is not possible to conduct perturbations during these time periods. Therefore, our perturbations are conducted in 2015 to avoid the model base years of our SCMs. In the main paper, we show some model responses out to 2300, the end of the MAGICC model runs, equal to 285 years after the perturbation. Additional results are in S8. We note that unit testing in software refer to a specific method of comparing output from the smallest portion of code, called a unit (i.e., function), to known outputs (Clune and Rood 2011). Here, we use this term in a similar way as van Vuuren *et al.* (van Vuuren et al. 2011), where MAGICC 6.0 was used as the reference output to compare several human-Earth system models. We conduct our unit test with comparable inputs and compare model-generated outputs from several SCMs.

We conduct perturbations of three contrasting chemical species: carbon dioxide ($CO_2$), methane ($CH_4$), and black carbon (BC). We begin with $CO_2$ because this well-mixed greenhouse gas is the largest contributor to anthropogenic forcing changes (Myhre et al. 2013). Methane is also of interest because it is a shorter-lived greenhouse gas, with chemical interactions with itself and other species (Cicerone, R.J.; Oremland 1988). Finally, we use BC perturbations to represent aerosols more generally because we are interested in model responses to a short-lived climate forcers (Bond et al. 2013; Harmsen et al. 2015). SCM representations of other aerosols species are similar so we do not conduct impulse tests of other species.

The SCMs we use are readily comparable because they read in similar emissions files. Background trajectory emissions are taken from the published Representative Concentration Pathway (RCP) 4.5 scenario (Thomson et al. 2011) database, which means that all calculations in the main paper are conducted relative to a changing $CO_2$ concentration background, unless otherwise noted. SCMs are often used to project global mean temperature over various future

~~scenarios, so this is the most relevant type of background on which to test these models. Conducting these experiment with a constant $CO_2$ background, as previously used in the literature (Joos et al. 2013), requires inverse modeling of the individual models to produce constant $CO_2$ concentration emissions files. Our methodology is easier to implement as a regular unit test. To this end, we provide comparable input emission files used in this paper.~~

~~In many SCMs, forcing over historical periods is explicitly calibrated to a model base year, so it is not possible to conduct perturbations during these time periods. Therefore, our perturbations are conducted in 2015 to avoid the model base years of our SCMs. In the main paper, we show some model responses out to 2300, the end of the MAGICC model runs, equal to 285 years after the perturbation. Additional results are in the Supplement (SI8).~~

We run reference scenarios in the SCMs, followed by each perturbation case described below.
For each experiment (see below) we report the response, which is obtained by subtracting the reference from the perturbation results. For instance, the $CO_2$ concentration response is obtained as follows:

$$CO_2 Concentration_{response}(t) = CO_2 Concentration_{perturbation}(t) - CO_2 Concentration_{reference}(t) \quad (1)$$

We conducted the following impulse tests:

a.  *Concentration impulse ($CO_2$).*

These SCMs can be used in a mode where $CO_2$ concentrations are exogenously specified. We
carry out this experiment by instantaneously increasing $CO_2$ concentration by 200 ppm in 2015. After 2015, $CO_2$ concentrations return to the baseline levels following the published RCP4.5 scenario. Note, we do not conduct separate forcing impulse experiments because this is functionally equivalent to a concentration impulse. In this experiment, we are only interested in the dynamics of the models' temperature response. This experiment eliminates the added
uncertainty in the emissions to concentrations calculation and complicating factors from carbon cycle feedbacks.

*b. Emissions impulse (BC, CH₄, CO₂).*

For this experiment all models were run with an emissions input. We carry out this experiment
by increasing individual emissions (BC, CH$_4$, or CO$_2$) in one year. Following that year, the
emissions return to the RCP4.5 pathway for all subsequent years. In this experiment CO$_2$ and
other GHG concentrations are allowed to vary as determined by each model. We find our
perturbation values by doubling the 2015 value for each chemical species equal to a 9.2 PgC
pulse of CO$_2$, a 329 Tg pulse of CH$_4$, and a 7981 Gg pulse of BC. We also perturb CO$_2$
emissions in 2010, 2020, 2030, 2040, 2050 to understand changes in model responses over time
and see a very small difference in the model response (S4). We compare results from three
comprehensive SCMs to two IR models, AR5-IR and FAIR model (Millar et al., 2017; Myhre et
al., 2013) (S2~~In this experiment CO$_2$ concentrations are allowed to vary as determined by each
model. We find our perturbation values by doubling the 2015 value for each chemical species
equal to a 9.2 PgC pulse of CO$_2$, a 329 Tg pulse of CH$_4$, and a 7981 Gg pulse of BC. We also
perturb CO$_2$ emissions in 2010, 2020, 2030, 2040, 2050 to understand changes in model
responses over time and see very small difference in the model response (SI5). We compare
results from three comprehensive SCMs to two IR models, AR5-IR and FAIR model (Millar et
al. 2017; Myhre et al. 2013) (SI2~~).

We also compared results to several ESMs and EMICs by carrying out a 100 GtC CO$_2$ impulse,
following Joos et al. (Joos et al., 2013) (S12). This is approximately 10x the CO$_2$ perturbation
pulse described above.

Finally, we conduct a 4xBC emissions step experiment. We compare the SCM temperature
responses with the response of a complex climate model used by Sand et al. (2016) (S13).

~~We also compared results to several ESMs and EMICs by carrying out a 100 GtC CO$_2$ impulse,
following Joos *et al.* (Joos et al. 2013) (SI11). This is approximately 10x the CO$_2$ perturbation
pulse described above.~~

c.  *Step increase in $CO_2$ concentration (instantaneous $4 \times CO_2$ concentration experiment).*
Similar to comparison (a), in this experiment, $CO_2$ concentrations are prescribed. We have $CO_2$
concentrations follow a pre-industrial pathway (278.0516 ppmv in 1765) until 2014. The $CO_2$
concentration is quadrupled (4x) in 2015, and maintained at this level until 2300. This follows the experimental protocol used in the CMIP5 experimental design (Taylor et al., 2012)(Taylor, Stouffer, and Meehl 2012).

We compare these results to drift-corrected (Gupta et al., 2013) global mean temperature results
from 20We compare these results to drift-corrected (Gupta et al. 2013) global mean temperature results from 15 complex climate models from the CMIP5 archive. We drift-correct the CMIP5
global mean temperature time series by subtracting the slope of the linear fit from the full-time
series of the corresponding pre-industrial experiment for each individual model (see S3).

We ran Hector v2.0 with few changes to the default configuration file settings. We changed two model time steps in Hector v2.0: (1) the carbon-cycle-solver.cpp time step from dt(0.3) to dt(0.1)
and (2) the ocean_component.hpp OCEAN_MIN_TIMESTEP from 0.3 to 0.01 to allow for the
carbon cycle, in particular, the ocean carbon cycle to accurately integrate across the sharp
gradient introduced by these experiments. In experiments where we constrained the $CO_2$
concentration, these changes significantly increase the model run time for this scenario.

Additionally, we used an equilibrium climate sensitivity (ECS) value of 3°C in the SCMs, with
the exception of the idealized SCMs, FAIR and AR5-IR (see Table S9). In both FAIR and AR5-
IR ECS is an emergent property from the choice of ocean parameters given by,, where these
parameters cannot be set by the user (see Table S9).

$$ECS = F_{2x}(q_1 + q_2) \tag{2}$$

where $F_{2x}$ is the forcing due to $CO_2$ doubling ($F_{2x} = 3.74\ Wm^{-2}$) and both $q_1$ and $q_2$ are the ocean parameters thermal adjustment of the upper ocean and thermal equilibrium of the deep ocean, respectively (Millar et al. 2017; Equation 4).

We conducted additional sensitivity experiments in the SCMs spanning ranges of climate sensitivity and ocean diffusivity and report the results in S11.

**S2 Discussion of Model SpecificationsDesign**

We conduct unit tests within three comprehensive SCMs and two stylized SCMs. The three comprehensive SCMs have structural differences worth noting. Hector v2.0, has explicit ocean carbon chemistry in four boxes, where ocean carbon uptake is a non-linear function of the solubility of carbon. MAGICC 5.3 BC-OC and 6.0 have differential hemispheric forcing over land and ocean, thereby calculating temperature over each box. Important characteristics of the carbon and climate components of each model are shown in Table S1.

**Table S1** Main carbon cycle and climate characteristics of SCMs and IRFs

| Model | Model description | Carbon cycle | Climate component |
|---|---|---|---|
| Hector v2.0 (Hartin et al., 2015, 2016; Kriegler, 2005) | mechanistic climate carbon-cycle model | One-pool atmosphere, three-pool land, and four-pool ocean | Global Energy balance model, with ocean heat diffusion |
| MAGICC 5.3 BC-OC (Raper and Cubasch, 1996; Smith and Bond, 2014; Wigley and Raper, 1992) | mechanistic climate carbon-cycle model | One-pool atmosphere, three-pool land, and one-pool ocean | 4-box Energy balance model, with ocean heat upwelling diffusion |
| MAGICC 6.0 (Meinshausen et al., 2011) | mechanistic climate-carbon cycle model | One-pool atmosphere, three-pool land, and one-pool ocean | 4-box Energy balance model, with ocean heat upwelling diffusion |
| AR5-IR (Myhre et al., 2013) | Impulse-response function | Impulse-response function | Equilibrium temperature as a function of RF |
| FAIR v1.0 (Millar et al., 2017) | Impulse-response function | Four timescale impulse-response function with state- | Equilibrium temperature as a function of RF; IRF with two timescales |

|  |  | dependence of the CO$_2$ airborne fraction |  |

Some SCMs also include representations of aerosol dynamics, though the model representations differ. As mentioned in the main paper, unlike Hector v2.0, both versions of MAGICC have differential hemispheric forcing over land and ocean. AR5-IR represents BC forcing response as a simple exponential, similar to the response from greenhouse gas forcing. FAIR v1.0, used here, represents the relationship between $CO_2$-only emissions, concentrations, and temperature. Other versions of FAIR include non-$CO_2$ forcing, such as BC.

**S2.1 Model Settings**

Here we discuss the model settings used in our experiments, noting any changes made to the default settings. The three comprehensive SCMs were run with the same ECS values unless otherwise noted. We also acknowledge that vertical ocean diffusivity has a large impact on ocean heat uptake and we do note that the SCMs we compare in our paper either do not have the same definitions of vertical ocean diffusivity, as is the case for the comprehensive SCMs, or ocean diffusivity is not directly represented in the models, as is the case for idealized SCMs. For our purposes, therefore, we kept the ocean diffusivity values at their default values within the comprehensive SCMs. Sensitivity experiments exploring the model response to the range of these two parameters derived from MAGICC 6.0 are available in S11.

**S2.2 AR5-IR**

The IPCC AR5 (Myhre et al. 2013; See caption under Figure 8.28) describes the underlying multi-gas impulse response model used to quantify the multi-gas equivalence metric, Absolute Global Temperature Potential (AGTP), to compare temperature changes at a chosen time in response to a unit pulse of emissions *i*. We refer to this model as AR5-IR and describe below how the sums of exponentials are used to find AGTP and the subsequent temperature response. AGTP is found via a convolution of the fraction of the species *i* remaining in the atmosphere after an emissions pulse and the climate response to a unit forcing,

$$R_T(t) = \sum_{j=1}^{M} \frac{c_j}{d_j} \exp(-\frac{t}{d_j}) \text{ (3; See Equation 8.SM.13).}$$

~~The IPCC AR5 (Myhre et al. 2013) describes a multi-gas impulse function using a multi-gas equivalence metric, Absolute Global Temperature Potential (AGTP), to compare temperature changes at a chosen time in response to a unit pulse of emissions $i$. AGTP is found via a convolution of the fraction of the species $i$ remaining in the atmosphere after an emissions pulse and the climate response to a unit forcing $R_T(t) = \sum_{j=1}^{M} \frac{c_j}{d_j} \exp(-\frac{t}{d_j})$ (1).~~

$$AGTP_i(H) = \int_0^H RF_i(t)R_T(H-t)dt \quad \text{(4; See 8.SM.14}$$

$$\text{}$$

and $RF_i(t) = A_i R_i(t)$ (5; See Equation 8.SM.7

$$\text{}$$

where for most species $R_i(t) = \exp(-\frac{t}{\tau_i})$ (6; See Equation 8.SM.8)

$$\text{}$$

and for $CO_2$ $R_{CO_2}(t) = a_0 + \sum_{i=1}^{N} a_i \exp(-\frac{t}{\tau_i})$ (7; See Equation 8.SM.10)

$$\text{}$$

and $A_i$ is the radiative efficiency yielding, the general equation:

$$AGTP_i(H) = A_i \sum_{j=1}^{2} \frac{\tau c_j}{\tau - d_j}\left(\exp\left(\frac{-H}{\tau}\right) - \exp\left(\frac{-H}{d_j}\right)\right) \quad \text{(8; See}$$

Equation 8.SM.14

AGTP can then be used to calculate global mean temperature change from any given emission scenario using,

$$\Delta T = \sum \int_o^t E_i(s)AGTP_i(t-s)ds \quad \text{(9; See Equation 8.1}$$

where $E_i$ are the emissions of a species, $t$ is the time horizon, and $s$ is the time of emissions (Myhre et al. 2013; See 8.7.13 and Equation 8.1). For this paper, AR5-IR was recoded in R and is available for download with the Supplementaty Materials.

**S2.3 FAIR**

The FAIR v1.0 model is a modified version of the AR5-IR carbon cycle component to include the state-dependence of the $CO_2$ airborne fraction to reproduce the relationship between $CO_2$-only emissions, concentrations, and temperature over the historical period. Millar et al. (2017) began with the impulse response functions used for calculation of multi-gas equivalence metrics in IPCC-AR5 (Myhre et al., 2013) and extended the $CO_2$ IRF by coupling the carbon-cycle to the thermal response and to cumulative carbon uptake by terrestrial and marine sinks. FAIR is available for download at

FAIR calculates the global mean temperature response as the sum of the temperature response from the fast and slow timescale components, which represent the upper and deep ocean. The model does not report the internally-calculated forcing response, so this is not included in Figure 2 in the main paper.

Here, we use the first iteration of FAIR, but we note that two new versions have recently been published, FAIR v1.1 and FAIR v1.3. FAIR v1.3 extends the original version to, "calculate non-$CO_2$ greenhouse gas concentrations from emissions, aerosol forcing from aerosol precursor emissions, tropospheric and stratospheric ozone forcing from the emissions of precursors, and forcings from black carbon on snow, stratospheric methane oxidation to water vapour, contrails and land use change (Smith et al., 2018)."

**S2.4 FAIR (without carbon cycle) versus AR5-IR**

We expect slight differences in the response of FAIR and AR5-IR to a unit forcing. According to Equation 8 in Millar et al. (2017), FAIR will have a differential response to changing background $CO_2$ concentrations. By contrast, AR5-IR parameterizes the climate response to a unit forcing, $R_T$, using a sum of exponentials as given by Equation 8.SM.13 in Myhre et al. (2013):

$$R_T(t) = \sum_j \frac{q_j}{d_j} e^{\frac{-t}{d_j}} \quad \text{(10; See Equation 8.SM.13)}$$

Values of the $R_T$ input parameters, $q_j$ and $d_j$, are available in Table S2 of this response, where j=1,2 represent the timescales of the fast and slow ocean response. We note all parameters $are$ independent of background concentration in AR5.

In the main paper, we used the time constant parameters representing the thermal equilibrium of the deep ocean (d2) and the thermal adjustment of the upper ocean (d1) from Myhre et al. (2013), rather than from Millar et al. (2017). We are testing the model responses as they would be 'out of the box' and only make modifications if required for the models to run, as was the case for Hector v1.1 to handle a 4x$CO_2$ concentration step.

Here we included additional model responses from the AR5-IR model using parameters from Millar et al. (2017). The parameter choices are available below in Table S2.

**Table S2** Parameter values for the simple impulse-response model, AR5-IR

| Parameter (Units) | Value – AR5-IR (from Myhre et al., 2013) | Value – AR5-IR-var (from Millar et al., 2017) | Guiding analogues |
|---|---|---|---|
| $\alpha$ (Wm$^{-2}$) | 5.35 | 5.395 ($\alpha$ = F2x/ln(2); F2x=3.74) | $CO_2$ RF scaling parameter |
| $q_1$ (KW$^{-1}$m$^2$) | 0.631 | 0.41 | Thermal adjustment of the upper ocean |
| $q_2$ (KW$^{-1}$m$^2$) | 0.429 | 0.33 | Thermal equilibrium of the deep ocean |
| $d_1$ (year) | 8.4 | 4.1 | Thermal adjustment timescale of the upper ocean |
| $d_2$ (year) | 409.5 | 239.0 | Thermal equilibrium timescale of the deep ocean |

Figure S1 shows the temperature response from a $CO_2$ concentration impulse and Figure S2 shows the temperature response from a $CO_2$ concentration step in several SCMs, including the AR5-IR response found using the Millar et al. (2017) time constants, which we refer to as "AR5-IR-Millar-parameters" in this figure. We note that the AR5-IR-parameters response is still not identical to FAIR because FAIR has a differential response to changing background $CO_2$ concentrations.

[Figure]

*concentration perturbation in SCMs (MAGICC 6.0 – yellow, MAGICC 5.3 BC-OC – red, Hector v2.0 – blue, AR5-IR – green, FAIR –pink, AR5-IR-Millar-parameters –light blue). The time-integrated response, analogous to the Absolute Global Temperature Potential, is reported as 0-285 years after the perturbation.*

[Figure]

**Figure S2** *Global mean temperature response from 4xCO$_2$ concentration step in CMIP5 models (grey) and SCMs (MAGICC 6.0 – yellow, MAGICC 5.3 BC-OC – red, Hector v2.0 – blue, FAIR – pink, AR5-IR – green, AR5-IR-Millar-parameters – light blue). A climate sensitivity value of 3°C was used in the SCMs and the thick lines represent CMIP5 models with an ECS between 2.5 - 3.5 °C.*

We note that, while the Millar et al. (2017) parameters in Table S2 may provide a better short-term fit, they underestimate the long-term response of the ocean. The long-term ocean thermal time scale, which can only be estimated using multi-century model runs, is known to be longer than 200 years from basic physical principles (as seen in the original literature cited by the AR5 model, which used longer model runs to inform those parameters). While this may be an acceptable tradeoff if this model is only going to be used over a 100-year timescale, this will inevitably lead to bias on longer time-scales. The simple climate models tested in this study are used for a variety of purposes and over a range of time-scales. This illustrates why we use the original parameters of the models as set by their designers.

~~Here, we use the first iteration of FAIR, but we note that two new versions have recently been published, FAIR v1.1 and FAIR v1.3. FAIR v1.3 extends the original version to, "calculate non-CO$_2$ greenhouse gas concentrations from emissions, aerosol forcing from aerosol precursor emissions, tropospheric and stratospheric ozone forcing from the emissions of precursors, and forcings from black carbon on snow, stratospheric methane oxidation to water vapour, contrails and land use change (C. J. Smith et al. n.d.)."~~

**S2.4 MAGICC 5.3 BC-OC**

MAGICC 5.3 BC-OC is a version of MAGICC 5.3 developed in conjunction with the Global
Change Assessment Model (GCAM). MAGICC 5.3 used here is available in GCAM version 4.4,
available for download at https://github.com/JGCRI/gcam-core/releases. The major change in
this version of MAGICC was the addition of explicit BC and OC (Smith and Bond, 2014).(S. J.
Smith and Bond 2014). To enable MAGICC 5.3 within GCAM, the climate model must be set to
<Value name = "climate">../input/climate/magicc.xml</Value> within the configuration file. We
ran this model with all its default configuration settings, unless otherwise noted in the text.

**S2.5 MAGICC 6.0**

MAGICC 6.0 was run with all the default settings. For the main experiments, the climate
sensitivity was set to 3.0°C to match the default setting of MAGICC 5.3 BC-OC and Hector
v2.0, unless otherwise noted. The MAGICC 6.0 executable is available for free download here:
http://www.magicc.org/.

**S2.6 Hector v2.0 Settings**

In the version we use here, Hector (v2.0), is coupled to a 1-D diffusive heat and energy balance
model (DOECLIM: Diffusion Ocean Energy balance CLIMate model). We are using the 1-D
diffusive ocean heat component of DOECLIM. DOECLIM is well documented and has been
widely used in climate uncertainty studies (Bakker et al., 2017; Kriegler, 2005; Urban et al.,
2014). Using default Hector parameter values for climate sensitivity and heat diffusivity, we find
that the new coupled model (Hector v2.0) exhibits improved vertical ocean structure and heat
uptake, as well as surface temperature response to radiative forcing, compared to earlier versions
of Hector.

**S3 CMIP5 Model Data**

We use a new version of Hector (v2.0), an open-source, object-oriented, simple global climate carbon cycle model (C. A. Hartin et al. 2015). The model can found at: http://github.com/JGCRI/hector. In the version used here (Hector v2.0), Hector v1.0 is coupled to a 1-D diffusive heat and energy balance model (DOECLIM: Diffusion Ocean Energy balance CLIMate model). DOECLIM is well documented and has been widely used in climate uncertainty studies (Bakker et al. 2017; Kriegler 2005b; Urban et al. 2014). DOECLIM includes three tunable parameters: climate sensitivity, ocean vertical heat diffusivity, and a scaling factor for aerosol forcing (Garner, Reed, and Keller 2016). Using default values for these parameters, we find that the new coupled model (Hector v2.0) exhibits improved vertical ocean structure and heat uptake, as well as surface temperature response to radiative forcing, compared to earlier versions of Hector.

**S3 CMIP5 Model Data**

The CMIP5 model data used to produce Figure 4, Figure S12, and Figure S22 is described here. Raw climateClimate model output from 2015 models was obtained from the CMIP5 data archive (http://cmip-pcmdi.llnl.gov/cmip5/data_portal.html) and the World DataDate Center for Climate site (http://cera-www.dkrz.de/WDCC/ui/Index.jsp). The monthly temperature data is aggregated to the global annual mean level using code developed using CDOs (see CDO 2018: Climate Data Operators.  Available at http://www.mpimet.mpg.de/cdo). The long-term drift is removed from the CMIP5 model data by subtracting the linear trend from the corresponding pre-industrial control run (Gupta et al., 2013)(Gupta et al. 2013). Table S3S2 provides the CMIP5 modeling centre name and the model name from Figure 4.

**Table S3** CMIP5 and SCM model information

| Centre(s) | Model name |
|---|---|
| Beijing Climate Center (BCC)
**China** | BCC-CSM1.1 |
| Canadian Centre for Climate Modelling and Analysis (CCCma)
**Canada** | CanESM2 |
| National Center for Atmospheric Research
**USA** | CCSM4 |
| Centre National de Recherches Météorologiques,
Centre Européen de Recherche et de Formation Avancée en Calcul Scientifique (CNRM-CERFACS)
**France** | CNRM-CM5  |
| Commonwealth Scientific and Industrial Research Organization/Queensland Climate Change Centre of Excellence
**Australia**
 | CSIRO-Mk3-6-0 |
|  |  |
|  |  |
| Institute of Atmospheric Physics, Chinese Academy of Sciences (LASG-CESS)
**China** | FGOALS-g2 |
| Geophysical Fluid Dynamics Laboratory (NCAR; NSF-DOE-NCAR)
**USA** | GFDL-CM3 |
|  | GFDL-ESM2G |
|  | GFDL-ESM2M |
| NASA/GISS (Goddard Institute for Space Studies; NASA-GISS)
**USA** | GISS-E2-H |
|  | GISS-E2-R |
| Institut Pierre Simon Laplace (IPSL)
**France** | IPSL-CM5A-LR |
|  | IPSL-CM5A-MR |
|  | IPSL-CM5B-LR |
| Atmosphere and Ocean Research Institute (The University of Tokyo),
National Institute for Environmental Studies, and
Japan Agency for Marine-Earth Science and Technology (MIROC) | MIROC-ESM |
|  | MIROC5 |

| | |
|---|---|
| **Japan** | |
| Max Planck Institute for Meteorology (MPI-M) | MPI-ESM-MR |
| **Germany** | MPI-ESM-P |
| Meteorological Research Institute  USA | MRI-CGCM3 |
| Norwegian Meteorological Institute Norway | NorESM1-M |
|  |  |

**S3.1 Abrupt 4xCO$_2$ concentration step response from Geoffroy et al. (2013)**

Conducting impulse tests with complex models is computationally expensive, illustrated by the few studies employing this technique to understand the responses of models. We cite the Sand et al. (2016) study that specifically investigated NorESM's response to black carbon (BC) perturbations (Sand et al., 2016). Another study by Yang et al. (2019) conducted similar BC perturbations in CESM (Yang et al., 2019). Other stylized CMIP5 experiments, such as the 1% CO$_2$ concentration experiment, are not included in our comparison because we do not consider them to be impulse response tests. It is not possible to cleanly extract the impulse response from the 1% experiments. The CMIP5 4xCO$_2$ concentration step experiment is mathematically related to impulse responses, so are a reasonable comparison, particularly because these are the largest suite of such tests conducted in complex models, which is the reason we highlight these results in the paper.

Geoffroy et al. (2013) reported the 4xCO$_2$ concentration step temperature change relative to the 150-year temperature mean from the corresponding pre-industrial control run. For comparison to the simple models, we report the drift corrected 4xCO$_2$ concentration step temperature change relative to the start of the 4xCO$_2$ concentration run. Therefore, there will be a difference in the temperature reported.

Figure S3 shows the global mean temperature response from the $4xCO_2$ concentration step
       experiment for the 20 CMIP5 models used in our comparison following the Geoffroy et al.
       (2013) procedure of reporting the $4xCO_2$ concentration step temperature change relative to the
       150-year temperature mean from the corresponding pre-industrial control run. The responses
       reported in Figure S3 are consistent with Geoffroy et al. (2013).

[Figure]

**Figure S3** *Global mean temperature response from $4xCO_2$ concentration step in CMIP5 models following the
Geoffroy et al. methodology.*

**S4 Sensitivity Experiments in MAGICC 5.3**

We conduct two sensitivity experiments to illustrate there is little impact of these choices on the model responses: (1) perturb $CO_2$ emissions in different years and (2) perturb $CO_2$ emissions at different levels in 2015.

**S4.1 Impact of Changes to the Years of Emission Impulses**

We test $CO_2$ emissions perturbations in different years from the default 2015 used in the main text. Figure S4 shows the global mean temperature response normalized by the 2010 global mean temperature response from a $CO_2$

emissions pulse in MAGICC 5.3. We found a maximum of 0.028°C/PgC difference in the response in MAGICC 5.3 and, therefore, carried out the remainder of the experiment in 2015, avoiding model base years.

[Figure]

**Figure S4** *Normalized global mean temperature response from $CO_2$ emissions impulses in MAGICC 5.3 carried out in different years.*

[Figure]

**Figure S1** *Normalized global mean temperature response from $CO_2$ emissions impulses in MAGICC 5.3 carried out in different years.*

**S4.2 Impact of Emissions Pulses Size on Temperature Response**

In the main text, we carried out annual emissions perturbations equivalent to doubling the value in 2015 to avoid model base years.   shows the global mean temperature response normalized by the perturbation size for different $CO_2$ perturbation sizes in 2015 in MAGICC 5.3. We found there was a maximum difference of 0.0015°C/PgC, and thus we continued our experiments using only one perturbation value.

[Figure]

**Figure S5** *Normalized global mean temperature response from different sized $CO_2$ emissions impulses in MAGICC 5.3 in 2015.*

[Figure]

**Figure S2** *Normalized global mean temperature response from different sized CO$_2$ emissions impulses in MAGICC 5.3 in 2015.*

**S5 Adjusted Total Forcing Response**

We found that MAGICC 5.3, MAGICC 6.0, and Hector v2.0 respond similarly to a $CO_2$
concentration impulse, with differences in the forcing and temperature responses arising from the
treatment of time within each model. Hector v2.0, for example, reads in annual average  emissions and carries out calculations  using that
same classification of time. MAGICC 5.3 and MAGICC 6.0 read in annual emissions
and interpolate to obtain mid-year and end-of-year values and uses those internally to calculate
concentration, forcing, and temperature at mid-year values, and successively reports temperature
at the end-of--year. This change in the timing affects the impulse response by distributing
the pulse over more time periods. Here, we demonstrate the impact of the adjustment for
the forcing response to a $CO_2$ concentration impulse.

[Figure]

[Figure]

**Figure S6** *Total forcing response from a $CO_2$ concentration impulse in SCMs.* All three *SCMs have a collinear response (MAGICC 6.0 – yellow, MAGICC 5.3 BC-OC – red, Hector v2.0 – blue*). The responses are co-linear past 2016.*).*

Due to the differences in model treatment of time, we offer a correction to the forcing in two of
the SCMs. MAGICC 5.3 and MAGICC 6.0 calculate forcing in mid-year, while Hector v2.0
reports forcing at the end of a year. The result is a broadened impulse response peak in both
versions of MAGICC, compared to Hector v2.0. The total forcing response from both version of
MAGICC, however, can be adjusted with the following equation;

$$F_i = (2xf_i) - f_{i-1} \qquad (118)$$

where $F_i$ is the adjusted forcing, $f_i$ is the unadjusted forcing at the current time step, and $f_{i-1}$ is
the unadjusted forcing at the previous time step.

[Figure]

**Figure S7** *Total forcing response from a $CO_2$ concentration impulse in SCMs. All three SCMs have a collinear response (MAGICC 6.0 – yellow, MAGICC 5.3 BC-OC – red, Hector v2.0 – blue, AR5-IR – green, FAIR –pink).*

shows the total forcing response adjusted from mid-year reporting, to end-year reporting using
equation (SI Eqn. S118).  We can also apply this adjustment to the BC impulse, however, the

MAGICC 6.0 distribution is larger in this case because MAGICC 6.0 annual emissions are interpolated to produce end-of-year and intermediate values. An annual emissions pulse is
effectively spread over two model years. In the main paper, we report the integrated response because over these periods, the timing of the internal model calculations has minimal impact on the model results. Additional integrated model responses are in S9.

[Figure]

**Figure S4** *Total forcing response from a CO₂ concentration impulse in SCMs. All three SCMs have a collinear response (MAGICC 6.0 – yellow, MAGICC 5.3 BC-OC – red, Hector v2.0 – blue, AR5-IR – green, FAIR – pink).*

**S6 Total Forcing Response from BC Emissions Impulse**

We see in Figure S8 that the model responses to a pulse of BC have similar patterns of instantaneous behavior seen in Figure 1 from the $CO_2$ concentration pulse. In general, the models behave similarly in response to a BC pulse; Hector v2.0 and AR5-IR have a collinear response, while MAGICC 6.0 distributes the BC emissions pulse over 3 years.

[Figure]

[Figure]

**Figure S8** *Total forcing response from a BC emissions perturbation in SCMs (MAGICC 6.0 – yellow, MAGICC 5.3 BC-OC – red, Hector v2.0 – blue, AR5-IR-green). AR5-IR and Hector v2.0 are collinear.*

**S7 CO₂ Concentration Responses from Emissions Impulses**

Figure S9 shows the $CO_2$ concentration responses from a BC and $CH_4$ emissions pulse. Every model response shows an eventual $CO_2$ concentration increases from a BC impulse; a feedback from the temp increase impact on the carbon cycle. From a $CH_4$ and BC emissions pulse, the $CO_2$ concentration response is stronger in MAGICC 6.0, followed by MAGICC 5.3 and Hector v2.0. MAGICC 6.0, however, shows an initial decrease in $CO_2$ concentration response from the BC pulse.

[Figure]

**Figure S9** *CO$_2$ concentration response from CH$_4$ and BC emissions perturbation (B) in SCMs (MAGICC 6.0 – yellow, MAGICC 5.3 BC-OC – red, Hector v2.0 – blue) illustrating the carbon-cycle feedbacks present in each model.*

Figure S9 also shows that $CH_4$ emission perturbations impact $CO_2$ concentration within both versions of MAGICC. The discrepancy between the MAGICC and Hector responses is partly due to $CH_4$ oxidation in MAGICC 5.3. The MAGICC 6.0 response is larger in Figure S9 presumably due to feedback effects in the model, however the general shape of the response is similar to the other two SCMs.

AR5-IR is notably absent from Figure S9 because, in this IRF model, the $CO_2$ concentrations are not affected by changing temperature (Millar et al., 2017). Rising temps in general and including temp changes due to $CH_4$ and BC emissions perturbations. FAIR v1.0 model (Millar et al., 2017) is absent from  Figure S9 because the model does not report out the internally-calculated forcing response. The $CO_2$ concentration response to a $CO_2$ emissions impulse in FAIR can be seen in .

The $CH_4$ chemistry components in Hector v2.0 and MAGICC 5.3 BC-OC are nearly identical, accounting for the similarities between these two SCMs responses (Hartin et al., 2015). MAGICC 5.3, however, includes $CH_4$ oxidation to $CO_2$, which might account for this response difference. To test this, Figure S10 shows the $CO_2$ concentration response from emissions impulse in SCMs. MAGICC 5.3 is shown with and without $CH_4$ oxidation included for a clearer comparison of the Hector v2.0 response. With the $CH_4$ oxidation turned off, the MAGICC 5.3 BC-OC response is similar to Hector v2.0 with only a slight difference after 2025.

[Figure]

**Figure S10** *CO$_2$ concentration response from emissions impulse in SCMs. MAGICC 5.3 is shown with and without CH$_4$ oxidation included (MAGICC 6.0 – yellow, MAGICC 5.3 BC-OC – red, Hector v2.0 – blue)..*

**S8 Model Responses out to 2300**

 -  show the $CO_2$ concentration response, total forcing response, and global mean temperature response from an emissions impulse, respectively, to the end of the model period equal to 2300.

**S8.1 $CO_2$ Concentration Response to a $CO_2$ Emissions Pulse**

 shows the $CO_2$ concentration response from a $CO_2$ emissions pulse in the SCMs out to 2300. We see that the SCMs respond similarly to this perturbation, with the exception of the stylized SCM, FAIR, which has a weaker response.

[Figure]

**Figure S11** *Carbon dioxide concentration response from a $CO_2$ emissions pulse in SCMs (MAGICC 6.0 – yellow, MAGICC 5.3 BC-OC – red, Hector v2.0 – blue, AR5-IR – green, FAIR - pink).*

[Figure]

**Figure S8** *Carbon dioxide concentration response from a CO₂ emissions pulse in SCMs (MAGICC 6.0 — yellow, MAGICC 5.3 BC-OC — red, Hector v2.0 — blue, AR5-IR — green, FAIR - pink).*

**S8.2 CH$_4$ Concentration Response from CH$_4$ Emissions Pulse**

 shows the CH$_4$ concentration response from a CH$_4$ emissions pulse in the comprehensive SCMs out to 2300. The stylized SCMs do not report CH$_4$ concentrations.  We see that the comprehensive SCMs behave similarly in their response to this perturbation, especially after 2050 when the response tends towards 0 ppb.

[Figure]

**Figure S12** *Methane concentration response from a CH$_4$ emissions pulse in SCMs out to 2300 (MAGICC 6.0 – yellow, MAGICC 5.3 BC-OC – red, Hector v2.0 – blue).*

[Figure]

**Figure S9** *Methane concentration response from a CH$_4$ emissions pulse in SCMs out to 2300 (MAGICC 6.0 – yellow, MAGICC 5.3 BC-OC – red, Hector v2.0 – blue).*

**S8.3 CO$_2$ Concentration Response to a BC or CH$_4$ Emissions Pulse**

shows the CO$_2$ concentration response from a CH$_4$ and BC emissions perturbations in the SCMs out to 2300. We see that the SCMs behave differently across the entire time series. Hector v2.0 appears to change state after 2225.

[Figure]

**Figure S13** *CO$_2$ concentration response from emissions perturbations in SCMs out to 2300 (MAGICC 6.0 – yellow, MAGICC 5.3 BC-OC – red, Hector v2.0 – blue).*

[Figure]

**Figure S10** *CO₂ concentration response from emissions perturbations in SCMs out to 2300 (MAGICC 6.0 — yellow, MAGICC 5.3 BC-OC — red, Hector v2.0 — blue).*

**S8.4 Total Forcing Response to a CO$_2$ or CH$_4$ Emissions Pulse**

We report the total forcing response from the models, rather than the individual species' forcing responses for comparability. Additionally, the total forcing is similar to the individual forcing responses because forcing is dominated by the forcing from the perturbed species.

    shows the total forcing response from a CH$_4$ and CO$_2$ emissions perturbations in the SCMs out to 2300. FAIR does not report total forcing.

[Figure]

**Figure S14** *Total forcing response from emissions perturbations in SCMs out to 2300 (MAGICC 6.0 – yellow, MAGICC 5.3 BC-OC – red, Hector v2.0 – blue, AR5-IR – green).*

[Figure]

**Figure S11** *Total forcing response from emissions perturbations in SCMs out to 2300 (MAGICC 6.0 – yellow, MAGICC 5.3 BC-OC – red, Hector v2.0 – blue, AR5-IR – green).*

**S8.5 Global Mean Temperature Response to a CH$_4$ or CO$_2$ Emissions Pulse**

shows the temperature response from a CH$_4$ and CO$_2$ emissions perturbations in the SCMs out

 to 2300. We see that most of the SCM responses differ slightly immediately following the perturbation, but converge over time. AR5-IR has a stronger response than the other SCMs immediately following the perturbation. More details are included in the main paper.

[Figure]

**Figure S15** *Global mean temperature response from emissions perturbations in SCMs out to 2300 (MAGICC 6.0 – yellow, MAGICC 5.3 BC-OC – red, Hector v2.0 – blue, AR5-IR – green, FAIR - pink).*

[Figure]

**Figure S12** *Global mean temperature response from emissions perturbations in SCMs out to 2300 (MAGICC 6.0 – yellow, MAGICC 5.3 BC-OC – red, Hector v2.0 – blue, AR5-IR – green, FAIR – pink).*

**S9 Time Integrated Responses**

Figure S16S13 – Figure S21S18 shows the integrated forcing and temperature response for the full suite of experiments to the end of the model period. The data tables in this section provide numerical data (rounded to three significant figures) supporting the integrated forcing or temperature response figures. The data tables also include percent differences found using the following formula:

$$Percent\ Difference_{i,t} = \left(\frac{Model\ response_{i,t} - Average\ Comprehensive\ Model\ Response_t}{Average\ Comprehensive\ Model\ Response_t}\right) \times 100 \qquad (129)$$

where *t* is the time horizon and *i* is the individual model. A positive percent difference indicates that the model response is stronger than the average comprehensive model response, while a negative value indicates the model response was weaker than the average comprehensive model response.

**S9.1 Time Integrated Responses from a CO$_2$ Concentration Impulse**

Figure S16S13 shows the time-integrated total forcing response from a CO$_2$ concentration impulse.

[Figure]

**Figure S16** *Time-integrated forcing response from a $CO_2$ concentration impulse for the SCMs to the end of the model period (MAGICC 6.0 – yellow, MAGICC 5.3 BC-OC – red, Hector v2.0 – blue, AR5-IR – green, FAIR - pink).*

Figure S17S14 shows the time-integrated global mean temperature response from a $CO_2$

concentration impulse to the end of the model period. We see that the comprehensive SCMs respond similarly, while AR5-IR has a stronger response and FAIR, a slightly weaker response.

The associated values time integrated temperature responses are in Table S4S3.

[Figure]

**Figure S17** *Time‑‑integrated temperature response from a CO$_2$ concentration impulse for the SCMs to the end of the model period (MAGICC 6.0 – yellow, MAGICC 5.3 BC-OC – red, Hector v2.0 – blue, AR5-IR – green, FAIR - pink).*

**Table S4** Integrated Temperature Responses from a $CO_2$ Concentration Impulse in the SCMs

| Time After Pulse | Integrated Temperature Response (°Cyr) | | | | | | Percent Difference from Comprehensive SCMs Average (%) | | | | |
|---|---|---|---|---|---|---|---|---|---|---|---|
| | MAGICC 5.3 BC-OC | MAGICC 6.0 | Hector v2.0 | FAIR | AR5-IR | Average of Comprehensive SCMs | MAGICC 5.3 BC-OC | MAGICC 6.0 | Hector v2.0 | FAIR | AR5-IR |
| 10 | 0.85 | 0.98 | 0.85 | 0.87 | 0.24 | 0.90 | -4.79 | 9.59 | -4.79 | -3.57 | -73.5 |
| 20 | 1.00 | 1.11 | 1.02 | 0.94 | 1.10 | 1.04 | -4.25 | 6.29 | -2.04 | -9.80 | 5.33 |
| 50 | 1.20 | 1.25 | 1.22 | 1.02 | 1.39 | 1.22 | -2.07 | 2.26 | -0.19 | -16.6 | 13.7 |
| 100 | 1.32 | 1.34 | 1.35 | 1.13 | 1.65 | 1.34 | -1.25 | 0.25 | 1.00 | -15.5 | 23.4 |
| 150 | 1.39 | 1.39 | 1.42 | 1.13 | 1.74 | 1.40 | -0.71 | -0.71 | 1.43 | -19.3 | 24.3 |
| 285 | 1.46 | 1.47 | 1.51 | 1.38 | 1.92 | 1.48 | -1.31 | -0.63 | 1.94 | -6.71 | 29.8 |

**S9.2 Time Integrated Responses from a CO$_2$ Emissions Impulse**

Figure S18S15 and Figure S19 showS16 shows the integrated forcing (Table S5S4) and temperature response (Table S6S5) for the CO$_2$ emissions impulse experiment to the end of the model period, respectively. The numerical data is shownshows in the Table S5S4 and Table

S6.S5.

[Figure]

**Figure S18** *Time—integrated total forcing response from a CO₂ emissions impulse for the SCMs to the end of the model period (MAGICC 6.0 – yellow, MAGICC 5.3 BC-OC – red, Hector v2.0 – blue, AR5-IR – green).*

[Figure]

[Figure]

**Figure S19** *Time‑integrated temperature response from a CO₂ emissions impulse for the SCMs to the end of the*

*model period (MAGICC 6.0 – yellow, MAGICC 5.3 BC-OC – red, Hector v2.0 – blue, AR5-IR – green, FAIR - pink).*

**Table S5** Integrated Forcing Responses from a $CO_2$ Emissions Impulse in the SCMs

| Time After Pulse | Integrated Forcing Response (Wm$^{-2}$yr) | | | | | Percent Difference from Comprehensive SCMs Average (%) | | | |
|---|---|---|---|---|---|---|---|---|---|
| | MAGICC 5.3 BC-OC | MAGICC 6.0 | Hector v2.0 | AR5-IR | Average of Comprehensive SCMs | MAGICC 5.3 BC-OC | MAGICC 6.0 | Hector v2.0 | AR5-IR |
| 10 | 0.51 | 0.43 | 0.48 | 0.54 | 0.47 | 8.67 | -9.51 | 0.85 | 14.38 |
| 20 | 0.88 | 0.75 | 0.85 | 0.91 | 0.82 | 6.39 | -9.38 | 2.99 | 10.63 |
| 50 | 1.70 | 1.48 | 1.65 | 1.86 | 1.61 | 5.63 | -8.28 | 2.65 | 15.38 |
| 100 | 2.81 | 2.50 | 2.67 | 3.17 | 2.66 | 5.52 | -5.96 | 0.44 | 19.13 |
| 150 | 3.82 | 3.47 | 3.62 | 4.32 | 3.63 | 4.97 | -4.52 | -0.45 | 18.87 |
| 285 | 6.26 | 5.97 | 6.03 | 7.12 | 6.09 | 2.79 | -1.89 | -0.90 | 16.98 |

**Table S6** Integrated Temperature Responses from a $CO_2$ Emissions Impulse in the SCMs

| Time After Pulse | Integrated Temperature Response (°Cyr) | | | | | | Percent Difference from Comprehensive SCMs Average (%) | | | | |
|---|---|---|---|---|---|---|---|---|---|---|---|
| | MAGICC 5.3 BC-OC | MAGICC 6.0 | Hector v2.0 | FAIR | AR5-IR | Average of Comprehensive SCMs | MAGICC 5.3 BC-OC | MAGICC 6.0 | Hector v2.0 | FAIR | AR5-IR |
| 10 | 0.16 | 0.16 | 0.13 | 0.14 | 0.18 | 0.15 | 6.81 | 5.49 | -12.31 | -5.71 | 20.66 |
| 20 | 0.33 | 0.33 | 0.31 | 0.29 | 0.42 | 0.32 | 2.99 | 1.44 | -4.43 | -9.38 | 29.90 |
| 50 | 0.81 | 0.78 | 0.79 | 0.69 | 1.00 | 0.79 | 1.86 | -1.69 | -0.17 | -13.07 | 26.15 |
| 100 | 1.50 | 1.45 | 1.46 | 1.27 | 1.93 | 1.47 | 2.16 | -1.59 | -0.57 | -13.51 | 31.44 |
| 150 | 2.17 | 2.10 | 2.10 | 1.85 | 2.86 | 2.12 | 2.20 | -1.10 | -1.10 | -12.87 | 34.69 |
| 285 | 3.87 | 3.85 | 3.80 | 3.44 | 5.87 | 3.84 | 0.83 | 0.17 | -1.00 | -10.38 | 52.93 |

**S9.3 Time Integrated Responses from a CH$_4$ Emissions Impulse**

Figure S17 and Figure S18 showshows the integrated forcing (Table S6) and temperature response (Table S7) for the CH$_4$ emissions impulse experiment to the end of the model period.

The numerical data in Table S6 and Table S7.

[Figure]

**Figure S20** *Time-integrated total forcing response from a CH₄ emissions impulse for the SCMs to the end of the*
*model period (MAGICC 6.0 – yellow, MAGICC 5.3 BC-OC – red, Hector v2.0 – blue, AR5-IR – green).*

[Figure]

**Figure S21** *Time -integrated temperature response from a CO₂ emissions impulse for the SCMs to the end of the*

*model period (MAGICC 6.0 – yellow, MAGICC 5.3 BC-OC – red, Hector v2.0 – blue, AR5-IR – green, FAIR - pink).*

**Table S7** Integrated Forcing Responses from a CH$_4$ Emissions Impulse in the SCMs

| Time After Pulse | Integrated Forcing Response (Wm$^{-2}$yr) | | | | | Percent Difference from Comprehensive SCMs Average (%) | | | |
|---|---|---|---|---|---|---|---|---|---|
| | MAGICC 5.3 BC-OC | MAGICC 6.0 | Hector v2.0 | AR5-IR | Average of Comprehensive SCMs | MAGICC 5.3 BC-OC | MAGICC 6.0 | Hector v2.0 | AR5-IR |
| 10 | 0.41 | 0.41 | 0.44 | 0.51 | 0.42 | -2.14 | -2.14 | 4.28 | 21.1 |
| 20 | 0.58 | 0.61 | 0.59 | 0.72 | 0.59 | -2.31 | 2.76 | -0.45 | 21.9 |
| 50 | 0.73 | 0.81 | 0.72 | 0.88 | 0.75 | -2.44 | 7.28 | -4.84 | 16.9 |
| 100 | 0.80 | 0.90 | 0.77 | 0.89 | 0.82 | -3.04 | 9.36 | -6.32 | 8.63 |
| 150 | 0.83 | 0.95 | 0.82 | 0.89 | 0.87 | -3.88 | 9.95 | -6.07 | 3.03 |
| 285 | 0.88 | 1.04 | 0.89 | 0.89 | 0.94 | -6.01 | 10.9 | -4.94 | -4.62 |

**Table S8** Integrated Temperature Responses from a $CH_4$ Emissions Impulse in the SCMs

| Time After Pulse | Integrated Temperature Response (°Cyr) | | | | | Percent Difference from Comprehensive SCMs Average (%) | | | |
|---|---|---|---|---|---|---|---|---|---|
| | MAGICC 5.3 BC-OC | MAGICC 6.0 | Hector v2.0 | AR5-IR | Average of Comprehensive SCMs | MAGICC 5.3 BC-OC | MAGICC 6.0 | Hector v2.0 | AR5-IR |
| 10 | 0.13 | 0.15 | 0.14 | 0.16 | 0.14 | -4.26 | 7.10 | -2.83 | 17.2 |
| 20 | 0.23 | 0.27 | 0.24 | 0.36 | 0.25 | -5.56 | 8.68 | -3.12 | 47.4 |
| 50 | 0.38 | 0.44 | 0.38 | 0.45 | 0.40 | -5.03 | 9.55 | -4.52 | 12.3 |
| 100 | 0.47 | 0.54 | 0.47 | 0.58 | 0.49 | -4.54 | 9.88 | -5.35 | 17.4 |
| 150 | 0.52 | 0.60 | 0.52 | 0.70 | 0.55 | -4.99 | 10.2 | -5.17 | 28.1 |
| 285 | 0.60 | 0.70 | 0.61 | 0.85 | 0.64 | -6.20 | 10.5 | -4.31 | 33.5 |

**S9.4 Time Integrated Responses from a BC Emissions Impulse**

Figure S19 and Figure S20 shows the integrated forcing and temperature response for the
BC emissions impulse experiment to the end of the model period, respectively. FAIR is not in
this figure because we used FAIR v1.0, which only represented the response from $CO_2$
emissions. An updated version, FAIR v1.3, was recently released and includes non-$CO_2$ forcing.
 Table S8 shows the integrated temperature response data.

[Figure]

**Figure S22** *Time-integrated total forcing response from a BC emissions impulse for the SCMs to the end of the model period (MAGICC 6.0 – yellow, MAGICC 5.3 BC-OC – red, Hector v2.0 – blue, AR5-IR – green).*

[Figure]

**Figure S23** *Time-‐integrated temperature response from a BC emissions impulse for the SCMs to the end of the*
*model period (MAGICC 6.0 – yellow, MAGICC 5.3 BC-OC – red, Hector v2.0 – blue, AR5-IR – green, FAIR - pink).*

We see that Hector v2.0, which does not differentiate BC forcing over land and ocean and has a

9% weaker response 20 years after the pulse. MAGICC 6.0 diverges from the MAGICC 5.3

temperature response 20 years after the pulse. AR5-IR represents the temperature response from a BC perturbation as a simple exponential decay analogous to the greenhouse gas IRF, leading to a much stronger integrated temperature response (20%) 20 years after the pulse.

**Table S9** Integrated Temperature Responses from a BC Emissions Impulse in the SCMs

| Time After Pulse | Integrated Temperature Response (°Cyr) | | | | | Percent Difference from Comprehensive SCMs Average (%) | | | |
|---|---|---|---|---|---|---|---|---|---|
| | MAGICC 5.3 BC-OC | MAGICC 6.0 | Hector v2.0 | AR5-IR | Average of Comprehensive SCMs | MAGICC 5.3 BC-OC | MAGICC 6.0 | Hector v2.0 | AR5-IR |
| 10 | 0.28 | 0.30 | 0.24 | 0.30 | 0.27 | 3.91 | 9.22 | -13.1 | 11.0 |
| 20 | 0.32 | 0.34 | 0.29 | 0.38 | 0.32 | 1.13 | 8.12 | -9.25 | 19.3 |
| 50 | 0.38 | 0.41 | 0.36 | 0.43 | 0.38 | -1.22 | 7.43 | -6.21 | 10.7 |
| 100 | 0.43 | 0.47 | 0.42 | 0.45 | 0.44 | -2.68 | 7.12 | -4.44 | 2.22 |
| 150 | 0.46 | 0.51 | 0.47 | 0.48 | 0.48 | -3.80 | 6.76 | -2.96 | -0.92 |
| 285 | 0.51 | 0.57 | 0.54 | 0.53 | 0.54 | -5.90 | 5.73 | 0.17 | -2.56 |

**S10 Temporal Response of SCMs Compared to 4xCO2 Concentration Step Experiment**
**from CMIP5**

Here we compare the 20-year moving average at time t=30, t= 50, t=70, t=100, and t=130 in the
CMIP5 models and SCMs to show the temporal response of temperature. Hector v2.0 and
MAGICC 5.3 have a faster response than the other SCMs and the majority of the complex
models to an abrupt $4xCO_2$ concentration step.

[Figure]

[Figure]

Figure S24 *20-Year moving average centered at year shown of the global mean temperature response from 4xCO$_2$ concentration step in CMIP5 models (grey) and SCMs (MAGICC 6.0 – yellow, MAGICC 5.3 BC-OC – red, Hector v2.0 – blue, FAIR – pink, AR5-IR –green).*

Table S9 shows the ECS values and the realized warming fraction (RWF) for the CMIP5 data and SCMs used to produce Figure 45. The RWF reveals that the SCMs used in this study generally warm faster than the more complex models in CMIP5.

**Table S10** *CMIP5 and SCM model information with ECS and RWF*

| Centre(s) | Model name | ESC (°C) | RWF (%)
$\sqrt{2}\cancel{LN(2)}\,x\,\dfrac{Average\ of\ the\ last\ 40\ years}{ECS}$ |
|---|---|---|---|
| Beijing Climate Center (BCC)
**China** | BCC-CSM1.1 | 2.8 | 58 |
| Canadian Centre for Climate Modelling and Analysis (CCCma)
**Canada** | CanESM2 | 3.7 | 54 |
| National Center for Atmospheric Research
**USA** | CCSM4 | 2.9 | 51 |
| Centre National de Recherches Météorologiques,
Centre Européen de Recherche et de Formation Avancée en Calcul Scientifique (CNRM-CERFACS)
**France** | CNRM-CM5 | 3.3 | 51 |
| Commonwealth Scientific and Industrial Research Organization/Queensland Climate Change Centre of Excellence
**Australia** | CSIRO-Mk3-6-0 | 4.1 | 47 |
| Institut Pierre Simon Laplace (IPSL)
**France** | IPSL-CM5A-LR | 4.1 | 49 |
|  | IPSL-CM5A-MR | NA | NA |
|  | IPSL-CM5B-LR | 2.6 | 54 |

| | | | |
|---|---|---|---|
| Institute of Atmospheric Physics, Chinese Academy of Sciences (LASG-CESS) **China** | FGOALS-g2 | NA | --NA |
| Atmosphere and Ocean Research Institute (The University of Tokyo), National Institute for Environmental Studies, and Japan Agency for Marine-Earth Science and Technology (MIROC) **Japan** | MIROC-ESM | 4.7 | 4865.1 |
| | MIROC5 | 2.7 | 4968.6 |
| Max Planck Institute for Meteorology (MPI-M) **Germany** | MPI-ESM-MR | NA | --NA |
| | MPI-ESM-P | 3.5 | 5171.7 |
| Meteorological Research Institute **Japan** | MRI-CGCM3 | 2.6 | 55 |
| Norwegian Meteorological Institute **Norway** | NorESM1-M | 2.8 | 48 |
| NASA/GISS (Goddard Institute for Space Studies; NASA-GISS) **USA** | GISS-E2-H | 2.3 | 4970.2 |
| | GISS-E2-R | 2.1 | 4462.3 |
| Geophysical Fluid Dynamics Laboratory (NCAR; NSF-DOE-NCAR) **USA** | GFDL-CM3 | 4.0 | 5270.5 |
| | GFDL-ESM2G | 2.4 | 5782.9 |
| | GFDL-ESM2M | 2.4 | 65 |
| Raper et al., 1996; Wigley and Raper 2002; Smith and Bond 2014 | MAGICC 5.3 BC-OC | 3.0* | 6482.0 |
| Meinshausen et al., 2011 | MAGICC 6.0 | 3.0* | 5383.8 |
| Hartin et al., 2015 | Hector v2.0 | 3.0* | 6390.3 |

| | | | |
|---|---|---|---|
| Hartin et al., 2016 | | | |
| Millar et al., 2017 | FAIR | 2.75 | 6186.2 |
| Myhre et al., 2013 | AR5-IR | 2.7 | 8866.8 |

*Unless otherwise noted.

Note: NA denotes models that have not reported an ESC value from Table 9.5 in IPCC

AR5(Flato et al., 2013).

**S11 Simple Sensitivity Tests in SCMs**

Here we discuss the SCM responses under a range of climate sensitivity and ocean diffusivity values.

S11Note: NA denotes models that have not reported an ESC value from Table 9.5 in IPCC

AR5(Flato et al. 2013).

**S10.1 Changing Equilibrium Climate Sensitivity Values in SCMs with Comparison to CMIP5**

We changed the ECS values in the SCMs to illustrate the effects of parameter selection on the model responses. We reproduce Figure 4 from the main paper using different ECS values in Hector v2.0, MAGCC 5.3, and MAGIC 6.0. We run each of these SCMs with a climate sensitivity values of 2.1°C, the same as GISS-E2-R, and 4.7°C, the same as MIROC-ESM. These two model values were selected because they represent the largest range of climate sensitivity values in the model data used here.

shows the global mean temperature response from 4xCO$_2$ concentration step in CMIP5 models and SCMs. The SCMs were run with two different ECS values. a shows the SCM response with an ECS value of 2.1°C and b shows the SCM responses with an ECS value of 4.7°C. We found that spanning the range of complex model ECS values still resulted in stronger SCM responses, which supports the conclusion in our main paper that the SCMs have a faster warming rate under strong forcing regimes compared to more complex models.

[Figure]

[Figure]

**Figure S25** *Global mean temperature response from 4xCO2 concentration step in CMIP5 models and SCMs, as in*

*Figure 5, with the SCMs run with two different ECS values. Figure 22a shows the SCM response with an ECS*
*value of 2.1°C, and Figure 22b shows the SCM responses with an ECS value of 4.7°C (MAGICC 6.0 – yellow,*

*MAGICC 5.3 BC-OC – red, Hector v2.0 – blue)*

**S11.2 Additional Sensitivity Experiment in SCMs Using MAGICC6.0 Parameters**

The aim of this paper is not to validate any individual simple climate model (SCM), nor the range of parameters used in the SCMs, which are also explored in the literature cited in our manuscript. Rather, we are evaluating the fundamental behavior of the simple models. However, understanding the uncertainty associated with our results is important.

 In S11.1 we conducted a simple sensitivity test for the $4xCO_2$ concentration step experiment by changing the climate sensitivity values in the three comprehensive SCMs used in this paper. Below, we have added some additional tests by exploring a range of climate sensitivity values and ocean diffusivity values in MAGICC 6.0 under a unit pulse of $CO_2$ emissions and a unit pulse of $CO_2$ concentration.

We selected climate sensitivity and ocean diffusivity values from the parameter ranges presented in Table 1B in Meinshausen et al. (2011). The values are the native MAGICC 6.0 parameters required to emulate complex models used in CMIP3 using three calibrated parameters (climate sensitivity, ocean diffusivity, and land/ocean warming). We provided the climate sensitivity and ocean diffusivity value ranges we explored in Table 11 below.

**Table S11** MAGICC 6.0 parameter values from Meinshausen et al., 2011 Table 1B for sensitivity tests

| Scenario | Climate sensitivity (K) | Ocean diffusivity ($cm^2s^{-1}$) |
| --- | --- | --- |
| Base Case | 3.0 | 1.1 |
| High Ocean diffusivity | 3.0 | 3.74 |
| Low Ocean diffusivity | 3.0 | 0.50 |
| High Climate sensitivity | 6.03 | 1.1 |
| Low Climate sensitivity | 1.94 | 1.1 |

Figure 26 shows the global mean temperature response exploring the range of ocean diffusivity
(Kz) (a) and global mean temperature response exploring the range of climate sensitivity (CS) (b)
under a $CO_2$ emissions perturbation. Figure 27 shows the same results for under a $CO_2$
concentration pulse. Both figures illustrate that climate sensitivity has the greatest impact on the
responses and in our manuscript, we accounted for this and used similar climate sensitivity values
in SCMs where possible, unless otherwise noted in the supplemental figures.

[Figure]

**Figure 26** *Global mean temperature response exploring the range of ocean diffusivity (Kz) (a) and Global mean*
*temperature response exploring the range of climate sensitivity (CS) (b) from a $CO_2$ emissions perturbation in SCMs. The*
*grey shaded region in each figure shows the range in MAGICC 6.0 responses found using the Table S11 parameters. We*
*note that the range of responses exploring CS (b) are normalized to account for the different climate conditions under*
*difference CS values. (MAGICC 6.0 – yellow, MAGICC 5.3 BC-OC – red, Hector v2.0 – blue, AR5-IR – green, FAIR –*
*pink, AR5-IR-Millar-parameters –light blue)*

[Figure]

**Figure 27** *Temperature response exploring the range of climate sensitivity (CS) (b) from a $CO_2$ concentration pulse in SCMs*
*(MAGICC 6.0 – yellow, MAGICC 5.3 BC-OC – red, Hector v2.0 – blue, AR5-IR – green, FAIR –pink, AR5-IR-Millar-parameters*
*–light blue). The grey shaded region in each figure shows the range in MAGICC 6.0 responses found using the Table R2*
*parameters. We note that the range of responses exploring CS (b) are normalized to account for the different climate conditions*
*difference CS values.*

We acknowledge, however, that vertical ocean diffusivity has a large impact on ocean heat
uptake and we do note that this parameter selection also impacts the responses in the SCMs,
particular under a $CO_2$ emissions pulse (Meinshausen et al., 2011). However, the SCMs we
compare in our paper either do not have the same definitions of vertical ocean diffusivity, as
is the case for the comprehensive SCMs, or ocean diffusivity is not directly represented in the
models, as is the case for idealized SCMs. For our purposes, therefore, we kept the ocean
diffusivity values at their default values within the comprehensive SCMs.

For completeness, we also acknowledge that Meinshausen et al. (2011) spanned ranges of
land/ocean warming contrast (RLO) in the three-parameter calibration described in Table 1B
of their manuscript. And again, the SCMs either use the same values of RLO, as is the case
for both versions of MAGICC, or this parameter is not represented in the idealized models. In
fact, from our work using impulse response test to characterize SCMs, we concluded that
SCMs without differential warming do not correctly capture the response pattern to BC
perturbations.

**S12 Comparison to Previous Impulse Responses Work by Joos et al. (2013)**

We conducted the same perturbation experiment done by Joos et al. (2013) with our three
comprehensive SCMs and two stylized SCMs, however, we do not conduct this against a
constant $CO_2$ concentration background. Instead, we use the RCP 4.5 scenario and add a 100GtC
$CO_2$ pulse in 2015. The versions used in each study differ slightly. Joos et al. used MAGICC
model version 6.3 run in 171 different parameter settings that emulate 19 AOGCMs and 9
coupled climate-carbon cycle models. MAGICC 6.0 used in this study was set at the default
setting using the AOGCM multi-model mean.

Table S12

Global Mean Temperature Response from 4xCO$_2$ Concentration Step With an ECS of 2.1°C

[Figure]

Global Mean Temperature Response from 4xCO$_2$ Concentration Step With an ECS of 4.7°C

[Figure]

**Figure S22** *Global mean temperature response from 4xCO2 concentration step in CMIP5 models and SCMs, as in Fig. 5, with the SCMs run with two different ECS values. Fig. 22a shows the SCM response with an ECS value of 2.1°C, and Fig. 22b shows the SCM responses with an ECS value of 4.7°C (MAGICC 6.0 — yellow, MAGICC 5.3 BC-OC — red, Hector v2.0 — blue)*

shows the time-integrated airborne fraction at chosen time horizons from the 100 GtC
pulse of $CO_2$ emissions. The Table S10 results are graphically represented in Figure S28
. These results are largely discussed in the main paper.

**Table S12** Time-integrated Airborne Fraction from a 100 GtC $CO_2$ Emissions Impulse in SCMs Compared to Results from Table 4 in Joos et al. (2013)

| Time Horizon | 20 yr | 50 yr | 100 yr |
|---|---|---|---|
| NCAR CSM1.4 | 13.8 | 27.8 | 46.6 |
| HadGEM2-ES | 14.7 | 30.9 | 53.3 |
| MPI-ESM | 14.5 | 29.2 | 48.8 |
| Bern3D-LPJ (reference) | 15.4 | 34.3 | 61.9 |
| Bern3D-LPJ ensemble | 15.1 (14.0-16.0) | 32.7 (28.9-36.0) | 57.6 (48.9-65.6) |
| Bern2.5D-LPJ | 13.9 | 29.7 | 51.1 |
| CLIMBER2-LPJ | 13.0 | 26.8 | 49.2 |
| DCESS | 14.6 | 31.8 | 56.3 |
| GENIE ensemble | 13.6 (10.9-17.6) | 28.9 (21.7-41.4) | 50.5 (38.3-77.9) |
| LOVECLIM | 13.5 | 27.9 | 45.3 |
| MESMO | 15.1 | 33.6 | 61.1 |
| UVic2.9 | 13.7 | 29.5 | 53.0 |
| ACC2 | 13.7 | 27.9 | 46.5 |
| Bern-SAR | 14.0 | 29.0 | 48.9 |
| TOTEM2 | 16.9 | 38.3 | 66.6 |
| MAGICC 6.0 ensemble | 14.0 (12.0-16.1) | 29.6 (23.6-35.7) | 51.8 (40.0-64.2) |
| Multi-model mean | 14.3 ± 1.8 | 30.2 ± 5.7 | 52.4 ± 11.3 |
| | | | |
| Hector v2.0 | 16.2 | 34.0 | 58.3 |
| MAGICC 5.3 | 16.0 | 33.4 | 58.3 |
| MAGICC 6.0 | 15.3 | 32.2 | 57.9 |
| AR5-IR | 15.0 | 31.0 | 53.1 |
| FAIR | 14.6 | 32.6 | 61.6 |

[Figure]

**Figure S28** *Time-integrated airborne fraction from a 100GtC $CO_2$ emissions impulse in SCMs compared to Joos et al. This is not a direct comparison because we did not perform this experiment with a constant $CO_2$ concentration background, as done by Joos et al. The colored points represent the time-integrated airborne fraction in the SCMs used in this study, following Joos et al., and the Joos et al. MAGICC 6.0 ensemble mean. The black point is the Joos et al. multi-model mean and the vertical black line represents the range of the Joos et al. model results. (Joos et al. MAGICC 6.0 ensemble mean –grey, MAGICC 6.0 – yellow, MAGICC 5.3 BC-OC – red, Hector v2.0 – blue, AR5-IR – green, FAIR –pink).*

We also indirectly compare the temperature response of the comprehensive SCMs and more complex models in Joos et al. MAGICC 6.0 was used both here and by Joos et al., and we find similar responses with ≤ 1 °C yr difference from Joos et al. at each reported period. Though the other two comprehensive SCMs were not used by Joos et al., their similar responses to our

MAGICC 6.0 allow us to make a larger conclusion, as done in the main paper. Using this logic, we are able to validate our SCM responses from a finite pulse., without conducting this experiment in ESMs or EMICs, directly. We find that the comprehensive SCM responses are generally less varied, close to the Joos et al. ensemble mean 20 years after the pulse, and below most Joos et al. model responses 50 and 100 years after the pulse (see Figure S29S24).

**Table S13** Time-integrated temperature response from a 100 GtC $CO_2$ Emissions Impulse in SCMs Compared to Results from Table 7 in Joos et al. (2013).

| Time Horizon | 20 yr | 50 yr | 100 yr |
|---|---|---|---|
| NCAR CSM1.4 | 2.53 | 7.36 | 10.6 |
| HadGEM2-ES | 4.24 | 12.4 | 30.3 |
| MPI-ESM | 3.83 | 8.84 | 19.1 |
| Bern3D-LPJ (reference) | 4.11 | 12.1 | 24.5 |
| Bern3D-LPJ ensemble | 3.20 (2.1-4.6) | 8.61 (5.1-13.5) | 17.3 (9.5-29.3) |
| Bern2.5D-LPJ | 3.15 | 8.40 | 17.1 |
| CLIMBER2-LPJ | 3.05 | 7.96 | 16.5 |
| DCESS | 3.38 | 9.96 | 20.6 |
| GENIE ensemble | 3.77 | 10.54 | 21.6 |
| LOVECLIM | 0.22 | 3.46 | 7.83 |
| MESMO | 4.41 | 12.5 | 26.0 |
| UVic2.9 | 3.40 | 9.17 | 18.5 |
| ACC2 | 3.99 | 10.55 | 20.0 |
| Bern-SAR | n/a | n/a | n/a |
| TOTEM2 | n/a | n/a | n/a |
| MAGICC 6.0 ensemble | 3.64 (2.7-4.7) | 8.96 (6.6-12.7) | 17.2 (12-26) |
| Multi-model mean | 3.29 ± 2.03 | 9.13 ± 4.45 | 18.7 ± 11.1 |
| | | | |
| Hector v2.0 | 3.05 | 8.20 | 15.54 |
| MAGICC 5.3 | 3.13 | 8.19 | 15.73 |
| MAGICC 6.0 | 3.39 | 8.28 | 15.54 |

[Figure]

**Figure S29** *Time-integrated temperature response from a 100GtC $CO_2$ emissions impulse in SCMs compared to Joos et al. This is not a direct comparison because we did not perform this experiment with a constant $CO_2$ concentration background, as done by Joos et al. The colored points represent the time-integrated temperature response in the SCMs used in this study, following Joos et al., and the Joos et al. MAGICC 6.0 ensemble mean. The black point is the Joos et al. multi-model mean and the vertical black line represents the range of the Joos et al. model results. (Joos et al. MAGICC 6.0 ensemble mean –grey, MAGICC 6.0 – yellow, MAGICC 5.3 BC-OC – red, Hector v2.0 – blue).*

We compare the comprehensive SCM responses from the 100GtC $CO_2$ pulse to our earlier experiment using a ~10GtC $CO_2$ pulse. We find that the relative behavior of the comprehensive

SCMs in the 100 GtC $CO_2$ impulse is similar to the response pattern from the smaller pulse experiment (see Figure 3a and Figure S29). The MAGICC 6.0 temperature response pattern is consistent with our prior experiments, where we see an initially stronger response (10 years following the perturbation) compared to the other comprehensive SCMs. Due to the initial oscillatory behavior in complex model responses (see Figure 2a in Joos et al. (2013)), it is difficult to compare SCM responses to complex models on these short time scale.

[Figure]

**Figure S30** *Total forcing response (a) and global mean temperature response (b) from a 100GtC CO2 emissions impulse in the SCMs (MAGICC 6.0 – yellow, MAGICC 5.3 BC-OC – red, Hector v2.0 – blue).*

 **S13 Investigating Temperature Response from BC Step Experiment**

We investigate SCM responses to a black carbon (BC) emissions step by quadrupling (4x) the
values in 2015. We choose two of the SCMs, Hector v2.0 and MAGICC 5.3, as examples and
compare the temperature response to Figure 1 in Sand et al. (2016).  Sand et al. finds that after
applying a 25x BC emissions step to NorESM1-M, a complex climate model, the temperature
response levels off after less than 10 years. We find that temperature in both of these SCMs
continue to increase over a century time-scale after the BC perturbation. The SCMs, therefore,
fail to capture the temporal response to BC as seen in Sand et al. (2016), also seen in Yang et al.
(2019).

[Figure]

**Figure S31** *Global mean temperature response from a 4xBC emissions step in the SCMs (MAGICC 5.3 BC-OC – red, Hector v2.0 – blue).*

~~experiment using a ~10GtC $CO_2$ pulse. We find that the relative behavior of the comprehensive~~

[Figure]

[Figure]

**Figure S25** *Total forcing response (a) and global mean temperature response (b) from a 100GtC CO2 emissions impulse in the SCMs (MAGICC 6.0 – yellow, MAGICC 5.3 BC-OC – red, Hector v2.0 – blue).*

**S12 Investigating Temperature Response from BC Step Experiment**

We investigate SCM responses to a black carbon (BC) emissions step by quadrupling (4x) the
values in 2015. We choose two of the SCMs, Hector v2.0 and MAGICC 5.3, as examples and
compare the temperature response to Figure 1 in Sand *et al* (Sand et al. 2016).  Sand *et al.* finds
that after applying a 25x BC emissions step to NorESM1-M, a complex climate model, the
temperature response levels off after it reaches 1.2K after less than 10 years. Sand *et al.* applies a
large BC step to increase the signal in the complex model, while we apply a smaller step in the
SCMs. We find that the SCM responses to a BC emissions step continue to increase 10 years
after the perturbation, suggesting that the SCMs fail to capture aerosol dynamics.

[Figure]

**Figure S26** *Global mean temperature response from a 4xBC emissions step in the SCMs (MAGICC 5.3 BC-OC – red, Hector v2.0 – blue).*

**S14  Summary of SCM Performance in Table 1**

Here we describe the choice of reporting  in Table 1 in the main paper. We justified  the

 use of the integrated

 response percent difference in the main paper, and report additional  percent differences in S9. In Table 1, we report the integrated response percent  difference for each of the experiments conducted in the SCMs at selected time horizons. We chose to report the time horizons for each experiment by taking into consideration the atmospheric lifetime of the species and the ability to compare the experiments.

For example, to

Under the 4xCO₂ concentration step experiment, we can compare the experiments exploring responses to CO2SCM response to more complex models from CMIP5. We assign MAGICC

6.0 a three (•••) because it appears to respond more reasonably under stronger forcing conditions than the other SCMs. We assign Hector v2.0, MAGICC 5.3, and FAIR a two (••) because these

SCMs have initially quicker responses to an abrupt 4xCO₂ concentration increase compared to the ESMs. We assign AR5-IR a one (•) because it has a slower response to an abrupt 4xCO₂

concentration increase and is insensitive to changing background concentrations.

For CH₄ emissions impulses, we use the difference from the comprehensive SCM average to rate the responses. Unlike the 100GtC CO₂ and 4xCO₂ step experiments, we cannot compare the

SCM responses to more complex models, therefore, we are more lenient in our performance assignment against the comprehensive SCM average. CH₄ is a well-mixed GHG and, therefore, we expect that the climate system response to CH₄ concentration perturbations, we report the responses at 100 years after the pulse. For CH4 and will be similar to that for CO₂. However, it would be useful to evaluate in more complex models if the simple representation of chemistry in the comprehensive SCMs adequately represents the time evolution of CH₄ concentrations in response to a change in emissions.

Finally, we assign ratings to the SCM responses to aerosols. We do not explicitly conduct aerosol experiments other than BC, we report at a shorter time horizon of 20 years after the pulse. because the responses of the SCMs to other aerosols will be similar to their response to

BC. We do not have a definitive reference for the time-dependent response to aerosol forcing perturbations. Instead, we rate the SCMs using the difference from the average of both MAGICC

models, which both differentiate aerosol forcing between land and ocean, which results in a faster overall climate response to aerosols as compared to greenhouse gases (Shindell 2014). In the case of BC, we note that all SCM response ratings should be reduced from the values shown because they do not accurately represent the temporal response to a BC step found in an ESM

(see S12). A more definitive evaluation of climate system responses to aerosol perturbations would be useful. This would require additional GCM simulations to step emission changes for various aerosol species and/or forcing mechanisms.

**S15 Supplementary Data**

Other supplementary materials for this manuscript can be found at https://github.com/akschw04/Fundamental-Impulse-Tests-in-SCMs-Datasets and include the following:

**Dataset S1 (separate file)**

Simple climate model responses from 4xBC emissions step.

**Dataset S2 (separate file)**

Simple climate model responses from $4xCO_2$ concentration step with 2.3 ocean diffusion and an ECS = 3 °C.

**Dataset S3 (separate file)**

Simple climate model responses from a 100PgC $CO_2$ emissions impulse experiment.

**Dataset S4 (separate file)**

Simple climate model responses from a $CH_4$ emissions impulse experiment.

**Dataset S5 (separate file)**

Simple climate model responses from a BC emissions impulse experiment.

**Dataset S6 (separate file)**

Simple climate model responses from $CO_2$ concentration impulse experiment.

**Dataset S7 (separate file)**

Simple climate model responses from $CO_2$ emissions impulse experiment.

**Dataset S8 (separate file)**

AR5-IR code to produce responses to BC emissions impulse.

**Dataset S9 (separate file)**

AR5-IR code to produce responses to $CH_4$ emissions impulse.

**Dataset S10 (separate file)**

AR5-IR code to produce responses to $CO_2$ emissions impulse.

**Dataset S11 (separate file)**

AR5-IR code to produce responses to 100PgC $CO_2$ emissions impulse for comparison to Joos *et*
*al.* (2013)

**Dataset S12 (separate file)**

AR5-IR code to produce responses to $CO_2$ concentration step.

**Dataset S13 (separate file)**

FAIR $CO_2$ concentration impulse experiment input file.

**Dataset S14 (separate file)**

FAIR 4x$CO_2$ concentration step experiment input file.

**Dataset S15 (separate file)**

FAIR $CO_2$ emissions impulse experiment input file.

**Dataset S16 (separate file)**

FAIR 100Pg $CO_2$ emissions impulse experiment input file.

**Dataset S17 (separate file)**

FAIR $CO_2$ emissions impulse experiment reference input file.

**Dataset S18 (separate file)**

Hector v2.0 $CO_2$ concentration impulse experiment input file.

**Dataset S19 (separate file)**

Hector v2.0 $CO_2$ concentration impulse experiment reference input file.

**Dataset S20 (separate file)**

Hector v2.0 $4xCO_2$ concentration step experiment reference input file.

**Dataset S21 (separate file)**

Hector v2.0 $4xCO_2$ concentration step experiment input file.

**Dataset S22 (separate file)**

Hector v2.0 BC emissions impulse experiment input file.

**Dataset S23 (separate file)**

Hector v2.0 BC emissions step experiment input file.

**Dataset S24 (separate file)**

Hector v2.0 $CH_4$ emissions impulse experiment input file.

**Dataset S25 (separate file)**

Hector v2.0 $CO_2$ emissions impulse experiment input file.

**Dataset S26 (separate file)**

Hector v2.0 100Pg $CO_2$ emissions impulse experiment input file.

**Dataset S27 (separate file)**

Hector v2.0 emissions impulse experiment reference input file.

**Dataset S28 (separate file)**

Hector v2.0 emissions step experiment reference input file.

**Dataset S29 (separate file)**

MAGICC5.3 $CO_2$ concentration impulse experiment reference input file.

**Dataset S30 (separate file)**

MAGICC5.3 $CO_2$ concentration impulse experiment input file.

**Dataset S31 (separate file)**

MAGICC5.3 $4xCO_2$ concentration step experiment input file.

**Dataset S32 (separate file)**

MAGICC5.3 $4xCO_2$ concentration step experiment reference input file.

**Dataset S33 (separate file)**

MAGICC5.3 BC emissions impulse experiment input file.

**Dataset S34 (separate file)**

MAGICC5.3 BC emissions step experiment input file.

**Dataset S35 (separate file)**

MAGICC5.3 $CH_4$ emissions impulse experiment input file.

**Dataset S36 (separate file)**

MAGICC5.3 1% $CO_2$ emissions impulse experiment in 2010 input file.

**Dataset S37 (separate file)**

MAGICC5.3 1.01% $CO_2$ emissions impulse experiment in 2010 input file.

**Dataset S38 (separate file)**

MAGICC5.3 5% $CO_2$ emissions impulse experiment in 2010 input file.

**Dataset S39 (separate file)**

MAGICC5.3 10% $CO_2$ emissions impulse experiment in 2010 input file.

**Dataset S40 (separate file)**

MAGICC5.3 50% $CO_2$ emissions impulse experiment in 2010 input file.

**Dataset S41 (separate file)**

MAGICC5.3 100% $CO_2$ emissions impulse experiment in 2010 input file.

**Dataset S42 (separate file)**

MAGICC5.3 100% $CO_2$ emissions impulse experiment in 2015 input file.

**Dataset S43 (separate file)**

MAGICC5.3 100% $CO_2$ emissions impulse experiment in 2020 input file.

**Dataset S44 (separate file)**

MAGICC5.3 100% $CO_2$ emissions impulse experiment in 2030 input file.

**Dataset S45 (separate file)**

MAGICC5.3 100% $CO_2$ emissions impulse experiment in 2040 input file.

**Dataset S46 (separate file)**

MAGICC5.3 100% $CO_2$ emissions impulse experiment in 2050 input file.

**Dataset S47 (separate file)**

MAGICC5.3 100% $CO_2$ emissions impulse experiment in 2060 input file.

**Dataset S48 (separate file)**

MAGICC5.3 100% $CO_2$ emissions impulse experiment in 2070 input file.

**Dataset S49 (separate file)**

MAGICC5.3 100PgC $CO_2$ emissions impulse experiment in 2015 input file.

**Dataset S50 (separate file)**

MAGICC5.3 $CO_2$ emissions impulse experiment reference input file.

**Dataset S51 (separate file)**

MAGICC5.3 $CO_2$ emissions step experiment reference input file.

**Dataset S52 (separate file)**

MAGICC56.0 4x$CO_2$ concentration impulse experiment input file.

**Dataset S53 (separate file)**

MAGICC56.0 4x$CO_2$ concentration impulse experiment reference input file.

**Dataset S54 (separate file)**

MAGICC56.0 4x$CO_2$ concentration step experiment input file.

**Dataset S55 (separate file)**

MAGICC56.0 4xCO$_2$ concentration step experiment reference input file.

**Dataset S56 (separate file)**

MAGICC6.0 BC emissions impulse experiment input file.

**Dataset S57 (separate file)**

MAGICC6.0 CH$_4$ emissions impulse experiment input file.

**Dataset S58 (separate file)**

MAGICC6.0 100% CO$_2$ emissions impulse experiment input file.

**Dataset S59 (separate file)**

MAGICC6.0 100PgC CO$_2$ emissions impulse experiment input file.

**Dataset S60 (separate file)**

MAGICC6.0 emissions impulse experiment reference input file.

**Dataset S61 (separate file)**

MAGICC 6.0 MAGCFG_USER parameters.

**Dataset S62 (separate file)**

FAIRv1.0 model with general parameters.

**Dataset S63 (separate file)**

AR5-IR general parameters.

**Dataset S64 (separate file)**

Hector general parameters.

**Dataset S65 (separate file)**

MAGICC5.3 maggas_c parameters.

**Dataset S66 (separate file)**

MAGICC5.3 magice_c parameters.

**Dataset S67 (separate file)**

MAGICC5.3 magmod_c parameters.

**Dataset S68 (separate file)**

MAGICC5.3 magrun_c parameters.

**Dataset S69 (separate file)**

MAGICC5.3 maguser_c parameters.

**Dataset S70 (separate file)**

MAGICC5.3 magxtra_c parameters.

**S16<s>S15</s> References**

Bakker, Alexander M. R., Tony E. Wong, Kelsey L. Ruckert, and Klaus Keller. 2017. "Sea-Level Projections Representing the Deeply Uncertain Contribution of the West Antarctic Ice Sheet." *Scientific Reports* 7(1): 3880. http://www.nature.com/articles/s41598-017-04134-5 (November 27, 2017).

Bond, T. C. et al. 2013. "Bounding the Role of Black Carbon in the Climate System: A Scientific Assessment." *Journal of Geophysical Research Atmospheres* 118(11): 5380–5552.

Cicerone, R.J.; Oremland, R.S. 1988. "GLOBAL BIOGEOCHEMICAL Methane ( CH4 ) Is the Most Abundant Organic Gas in Measurement Data Has Been Reviewed by Ehhalt [ 1974 ] and Wofsy [ 1976 ] and Will Not Be Repeated Here . Since the Earth ' s Energy Balance ( See The Quasi-Steady State ( Defined In." 2(4): 299–327.

Clune, Thomas L, and R. B. Rood. 2011. "Software Testing and Verification in Climate Model Development." *IEEE Software* 28(6): 49–55. http://ieeexplore.ieee.org/lpdocs/epic03/wrapper.htm?arnumber=5999647.

Flato, Gregory et al. 2013. "Evaluation of Climate Models." *Climate Change 2013: The Physical Science Basis. Contribution of Working Group I to the Fifth Assessment Report of the Intergovernmental Panel on Climate Change*: 741–866. http://elib.dlr.de/95697/1/Alet-Eyring-WG1AR5_Chapter09_FINAL.pdf (October 17, 2017).

Garner, Gregory, Patrick Reed, and Klaus Keller. 2016. "Climate Risk Management Requires Explicit Representation of Societal Trade-Offs." *Climatic Change* 134(4): 713–23. http://link.springer.com/10.1007/s10584-016-1607-3 (November 27, 2017).

Gupta, Alexander Sen, Nicolas C. Jourdain, Jaclyn N. Brown, and Didier Monselesan. 2013. "Climate Drift in the CMIP5 Models." *Journal of Climate* 26(21): 8597–8615.

Harmsen, Mathijs J H M et al. 2015. "How Well Do Integrated Assessment Models Represent Non-CO 2 Radiative Forcing?" *Climatic Change* 133: 565–82. https://link.springer.com/content/pdf/10.1007%2Fs10584-015-1485-0.pdf (December 19, 2017).

Hartin, C. A. et al. 2015. "A Simple Object-Oriented and Open-Source Model for Scientific and Policy Analyses of the Global Climate System - Hector v1.0." *Geoscientific Model Development* 8(4): 939–55.

Hartin, Corinne A, Benjamin Bond-Lamberty, Pralit Patel, and Anupriya Mundra. 2016. "Ocean Acidification over the next Three Centuries Using a Simple Global Climate Carbon-Cycle Model: Projections and Sensitivities." *Biogeosciences* 13: 4329–42. www.biogeosciences.net/13/4329/2016/ (October 17, 2017).

Joos, F. et al. 2013. "Carbon Dioxide and Climate Impulse Response Functions for the Computation of Greenhouse Gas Metrics: A Multi-Model Analysis." *Atmospheric Chemistry and Physics* 13(5): 2793–2825.

Kriegler, Elmar. 2005a. "Imprecise Probability Analysis for Integrated Assessment of Climate Change." *Time*: 258. https://publishup.uni-potsdam.de/opus4-ubp/frontdoor/index/index/docId/497 (October 29, 2017).

———. 2005b. "Imprecise Probability Analysis for Integrated Assessment of Climate Change." *Time*: 258. https://www.pik-potsdam.de/members/edenh/theses/PhDKriegler.pdf (November 27, 2017).

Meinshausen, M., S. C B Raper, and T. M L Wigley. 2011. "Emulating Coupled Atmosphere-Ocean and Carbon Cycle Models with a Simpler Model, MAGICC6 - Part 1: Model Description and Calibration." *Atmospheric Chemistry and Physics* 11(4): 1417–56.

Millar, J. Richard, Zebedee R. Nicholls, Pierre Friedlingstein, and Myles R. Allen. 2017. "A Modified Impulse-Response Representation of the Global near-Surface Air Temperature and Atmospheric Concentration Response to Carbon Dioxide Emissions." *Atmospheric Chemistry and Physics* 17(11): 7213–28.

Myhre, Gunnar et al. 2013. "Anthropogenic and Natural Radiative Forcing." *Climate Change 2013: The Physical Science Basis. Contribution of Working Group I to the Fifth Assessment Report of the Intergovernmental Panel on Climate Change*: 659–740.

Raper, S C B, and U Cubasch. 1996. "Emulation of the Results from a Coupled General Circulation Model Using a Simple Climate Model." *Geophysical Research Letters* 23(10): 1107–10.

Sand, M. et al. 2016. "Response of Arctic Temperature to Changes in Emissions of Short-Lived Climate Forcers." *Nature Climate Change* 6(3): 286–89.

Shindell, Drew T. 2014. "Inhomogeneous Forcing and Transient Climate Sensitivity." *Nature Climate Change* 4(4): 18–21.

Smith, Christopher J et al. "FAIR v1.1: A Simple Emissions-Based Impulse Response and Carbon Cycle Model." https://www.geosci-model-dev-discuss.net/gmd-2017-266/gmd-2017-266.pdf? (December 19, 2017).

Smith, S J, and T C Bond. 2014. "Two Hundred Fifty Years of Aerosols and Climate: The End of the Age of Aerosols." *Atmospheric Chemistry and Physics* 14(2): 537–49. www.atmos-chem-phys.net/14/537/2014/ (October 12, 2017).

Taylor, Karl E., Ronald J. Stouffer, and Gerald A. Meehl. 2012. "An Overview of CMIP5 and the Experiment Design." *Bulletin of the American Meteorological Society* 93(4): 485–98.

Thomson, Allison M. et al. 2011. "RCP4.5: A Pathway for Stabilization of Radiative Forcing by 2100." *Climatic Change* 109(1): 77–94. http://link.springer.com/10.1007/s10584-011-0151-4 (October 12, 2017).

Urban, Nathan M. et al. 2014. "Historical and Future Learning about Climate Sensitivity." *Geophysical Research Letters* 41(7): 2543–52. http://doi.wiley.com/10.1002/2014GL059484 (November 27, 2017).

van Vuuren, Detlef P. et al. 2011. "How Well Do Integrated Assessment Models Simulate Climate Change?" *Climatic Change* 104(2): 255–85.

Wigley, T. M. L., and S. C. B. Raper. 1992. "Implications for Climate and Sea Level of Revised
 IPCC Emissions Scenarios." *Nature* 357(6376): 293–300.
 http://www.nature.com/articles/357293a0 (October 12, 2017).

Yang, Y., Smith, S. J., Wang, H., Mills, C. M. and Rasch, P. J.: Variability, timescales, and non-
linearity in climate responses to black carbon emissions, Atmos. Chem. Phys. Discuss., 1–44,
doi:10.5194/acp-2018-904, 2019.

---

## Referee Report (RR1)

**Second report from reviewer #1**

The essence of my critique of the the manuscript by Schwarber et al. is contained in he following comment from my first report:

**Testing of simple models against more complex ones is interesting and relevant to ESD, but the interpretation of results are difficult, since it is not obvious that a complex model represents specific aspects of reality more correctly than a simple model.**

The authors' response to this is:

*We appreciate that you agree this work is interesting and relevant to ESD. Comparing simplified models to more complex models is a technique often utilized in the literature (e.g., Joos et al., 2013) and we also employ this technique. We compare the responses of idealized SCMs to comprehensive SCMs and comprehensive SCMs to CMIP5-class models. In our paper, we do not necessarily expect individual models to represent reality, but instead rely on the multi-model mean to ground our comparisons. It is well established that the multi-model mean behavior of the complex models replicates a broad suite of observations better than any individual model (e.g., Figure 9.7, Flato et al. 2013). Our subsequent responses will also address this comment.*

Unfortunately, I do not think this justification is correct and rests on a flawed interpretation on results in the literature, including Fig. 9.7 in Flato et al. 2013 (Chapter 9 in the IPCC AR5 report). Below, I will present my arguments.

Figure 9.7 in Flato et al (2013) deals with the RMS-difference between space-time global seasonal-cycle climatology of models and observations. This means that in every grid cell the monthly climatology is computed based on the years 1980-2005 to produce a mean annual cycle for this period in the model and in the observation (reanalysis), and the RMS-difference is produced. There are two features to notice: (i) The metric for comparing model with observation is based on the full space time-field, not the global average as done in the present manuscript (MS). (ii) The metric measures the RMS-difference over the annual cycle in historical runs/observation over a 25-year period, while Schwarber et al. measure the percentage difference of the time-integrated response of pulsed forcing experiments over 100 or 20 years. Hence the data compared and the metric used in Fig. 9.7 and in Schwarber et al. have very little in common.

The feature of Fig. 9.7 which Schwarber et al. use as justification is that the RMS-difference seems to be smaller for the so-called mean model than for any of the individual models, and that – with respect to this specific metric – the mean model is the better representation of reality. This has been shown empirically to hold true for many other model fields, not only for the annual cycle, but I have never seen it demonstrated for the long-time response of the global mean temperature for an impulse or step forcing.

It would actually have been a groundbreaking result, if this could be shown to be true, because the metric used by Schwarber et al. applied to the 4xCO2 step-forcing experiments would effectively measure the equilibrium climate sensitivity (ECS). If it were true that the mean model (the ensemble mean of the individual model experiments) is closer to reality than any of the individual models in this metric, then we would know that the ECS of the mean model is very close to thetrue ECS, and all the problems we have with the uncertainty in the ECS-estimate would evaporate.

 A theoretical result explaining many observations like those in Fig. 9.7 was published by Annan and Hargreaves, J. Climate, 4537 (2011). It rests on the assumption that the observed reality and models are drawn from the same statistical distribution, but does not assume that this distribution is centered around the observation. They compute the probability that an ensemble member is closer to reality than the ensemble mean and show that it is generally small if the dimension of the data vector is large (se Figure 2 in that paper). For small effective dimension, however, this is no longer true. The metric used by Schwarber et al. measures only one number, the integrated response after 100 (or 20) years, so the data vector has dimension 1. This explains why one cannot use the ensemble mean of the complex models as  "the truth" when assessing the performance of the simple models.

It probably will not help much to use a higher-dimensional data set to characterize the model solutions, since the simplest models are completely determined by a rather small number of model parameters, which renders the effective dimension small.

I have a number of reservations also with other aspects of the manuscript and the authors' response, but the problem I have discussed above is so serious that I cannot recommend publication.

---

## Author Response (AR2)

Dear Editor,

Thank you for inviting us to submit a revised manuscript. Below we have summarized the important changes made to the manuscript in response to the comments provided:

1. We have clarified language concerning the use of the multi-model mean in our discussion.
2. We have clarified the description of the climate sensitivity parameters used in the models and more clearly directed readers to our Supplement, which contains a very thorough discussion of the model specifications.
3. We have added a brief paragraph on the theoretical underpinning of our work while recognizing the journal's goal to remain interdisciplinary.
4. We made minor language edits for clarity throughout.

**Response to Reviewer #1**

**Comment: Testing of simple models against more complex ones is interesting and relevant to ESD, but the interpretation of results are difficult, since it is not obvious that a complex model represents specific aspects of reality more correctly than a simple model. … I do not think this justification is correct and rests on a flawed interpretation of results in the literature, including Fig. 9.7 in Flato et al. 2013 (Chapter 9 in the IPCC AR5 report).**

**Comment reiterated by Dr. Michel Crucifix: "The reviewer #1 develops a valid point to criticize an element in the response of the author --- namely, the ensemble-mean of low-dimensional quantities (like the global mean temperature) cannot be expected to be closer to reality than that of individual models. Reviewer #1 appears to have based his advice of rejection mainly on that argument. Fortunately, this argument is not critical to the overall validity and relevance of the article."**

Response: We have clarified our use of the multi-model mean.

*Changes in Manuscript: We added the following text:*

*"For purposes of summarizing our results we compare the individual model responses to the comprehensive SCM multi-model mean for most of our experiments. We use this both for convenience and because the comprehensive SCMs can generally replicate the long-term results of general circulation models (GCMs; Meinshausen et al., 2011; Joos et al., 2013; Hartin et al., 2015, 2016). This is also, in a general philosophical sense, in line with the finding from GCMs that multi-model means compare better to observations than individual models (Flato et al., 2013), although we note that the Flato et al. finding was not specifically for global temperature. We, therefore, are not implying that the comprehensive SCM mean is necessarily the most accurate representation of the actual climate system response. It is instead simply a convenient metric for comparison. This metric illustrates both where the comprehensive SCMs are similar or different, and where the more idealized models differ from the comprehensive SCMs. Most of these latter differences are due to simplifications in the idealized models that bias their results, as discussed previously."*

**Response to Reviewer #2**

**Comment: As a minor comment, I must be missing something, but am surprised by the response of the AR5-IR model to a 4xCO2 scenario which seems to be trending to a value substantially higher than 6C. This seems odd in light of the stated sensitivity of 3C to a doubling of CO2.**

Response: We realized that our description of the climate sensitivity parameter in our paper has been misleading in places. In particular, in Figure 4 we previously noted that the climate sensitivity used in the SCMs was 3°C, however this was only the case for the comprehensive SCMs. Equilibrium climate sensitivity is a result of ocean parameters for both of the idealized SCMs, FAIR and AR5-IR. Therefore, the Reviewer is correct and the ECS value was 3.9°C in AR5-IR and 2.7°C in FAIR. More on this aspect of the models is available in S2, where we also explore the AR5-IR responses using other parameter values.

Additionally, as we noted in the main text, AR5-IR lacks a nonlinearity from concentration to forcing and is insensitive to changing background concentrations.

*Changes in Manuscript: We amended the text as mentioned above. In addition, we added language that clarifies the parameters used in each of the models in the Supplement. We also note throughout that the model parameters and model input files are available in our Supplementary Materials, which are also available for download on Github.*

**Response to Dr. Lucarini's comments**

**Comment: What is entirely missing is the theoretical framework for what they do. Indeed, the recent work of M. Ghil and myself goes in that direction and provides a pretty solid background for what they do. Testing response to forcings makes perfect sense if one wants to compare models; response can be (approximately) linear even if models are fully nonlinear; this is the essence of Ruelle's response theory in statistical mechanics (and variants thereof).**

Response: We agree that including background information on the theoretical underpinning of this work is important and have updated the manuscript accordingly.

*Changes in Manuscript: Citations and language were added to the introduction of the manuscript as follows:*

*"The impulse tests result in an impulse response function (IRF) for each model/species combination. IRFs characterize the dynamics of a linear system (Joos and Bruno, 1996; Ruelle, 2009) and, although climate models exhibit nonlinear responses, even some non-linear systems can be approximated by IRFs for small perturbations (Hooss et al., 2001; Lucarini and Sarno, 2011; Lucarini, 2018). The impulse responses examined here can be considered Green's functions, which form a key component of many simple climate models (Joos et al., 1999; van Vuuren, 2011a; Millar et al., 2015)."*

**Comment: Obviously, they need to provide full details on the model setups because results have to be reproducible by other people.**

Response: Our Supplement includes a very thorough discussion of the models used and their specifications, and we have added some additional references to that discussion (see S1 and S2) in the main text of the manuscript.

*Changes in Manuscript: Additional language citing the supplement was added to the main text.*

**Comment: Finally, I believe that the interpretation of the results should be expanded.**

Response: We added text to the discussion and conclusion section.

*Changes in Manuscript: Text has been moved from the supplement to the main text to add clarity. For example, we added:*

*"As a summary of our findings, we report the differences in time-integrated temperature response from the relevant multi-model mean in Table 1 for each of the experiments at selected time horizons. We chose the time horizons to report for each experiment by taking into consideration the atmospheric lifetime of the species and the ability to compare the experiments. For example, to compare the experiments exploring responses to $CO_2$ perturbations, we report the responses at 100 years after the pulse. For $CH_4$ and BC, we report at a time horizon of 20 years after the pulse reflecting the shorter lifetime of these species. Additional time-integrated temperature responses can be found in S9."*